# Quality control of protein synthesis in the early elongation stage

Asuteka Nagao [1] ✉, Yui Nakanishi[1], Yutaro Yamaguchi[1], Yoshifumi Mishina[1], Minami Karoji[1], Takafumi Toya[1], Tomoya Fujita[2], Shintaro Iwasaki [2,3], Kenjyo Miyauchi [1], Yuriko Sakaguchi[1] & Tsutomu Suzuki [1] ✉

In the early stage of bacterial translation, peptidyl-tRNAs frequently dissociate from the ribosome (pep-tRNA drop-off) and are recycled by peptidyl-tRNA hydrolase. Here, we establish a highly sensitive method for profiling of pep-tRNAs using mass spectrometry, and successfully detect a large number of nascent peptides from pep-tRNAs accumulated in *Escherichia coli pth*[ts] strain. Based on molecular mass analysis, we found about 20% of the peptides bear single amino-acid substitutions of the N-terminal sequences of *E. coli* ORFs. Detailed analysis of individual pep-tRNAs and reporter assay revealed that most of the substitutions take place at the C-terminal drop-off site and that the miscoded pep-tRNAs rarely participate in the next round of elongation but dissociate from the ribosome. These findings suggest that pep-tRNA drop-off is an active mechanism by which the ribosome rejects miscoded pep-tRNAs in the early elongation, thereby contributing to quality control of protein synthesis after peptide bond formation.

Accurate translation of the genetic information in mRNA into a functional protein is achieved by the concerted action of the translational apparatus, which consists of the ribosome, mRNA, tRNA, and translation factors. The overall rate of mistranslation ranges from $6 \times 10^{-4}$ to $5 \times 10^{-3}$ in bacteria[1–6] and is similar in eukaryotic cells[4]. Precise aminoacylation of tRNA is a crucial step in quality control of protein synthesis. Accordingly, aminoacyl-tRNA synthetases (aaRSs) strictly discriminate cognate tRNAs and amino-acids from non-cognate ones[7,8]. Even if misacylation takes place, some aaRSs use their editing activity to hydrolyze the incorrect aminoacyl-tRNA (aa-tRNA), thereby preventing mistranslation[9]. The aa-tRNA is recognized by the GTP-bound EF-Tu to form a ternary complex (EF-Tu·GTP·aa-tRNA), and accurate selection of cognate aa-tRNA by EF-Tu also plays a critical role in maintaining translational fidelity[10–13]. The ternary complex stochastically binds the A-site of the ribosome and determines whether the A-site codon is cognate by monitoring the codon–anticodon interaction[4,14]. The dissociation rate is much higher for non-cognate than for cognate aa-tRNAs, suggesting that non-cognate aa-tRNAs are

rejected at this step[15–17]; accordingly, this step is referred to as "initial selection"[4,14]. When the A-site codon is properly recognized by a cognate aa-tRNA, the decoding center in 16 S rRNA monitors the codon–anticodon interaction to induce a large conformational rearrangement of the 30 S subunit, triggering GTP hydrolysis of the ternary complex[18–20]. EF-Tu·GDP then dissociates from the ribosome, allowing the 3' terminus of the aa-tRNA to enter the A-site of the 50 S subunit. Peptide bond formation then takes place between the pre-existing peptidyl-tRNA (pep-tRNA) at the P-site and the aa-tRNA at the A-site, extending the peptide chain. After GTP hydrolysis but before peptide bond formation, near-cognate tRNAs are rejected by a mechanism called "proofreading"[4,14–16,21–24]. These two checkpoints occur before peptide bond formation and increase the accuracy of decoding during elongation.

Fidelity of translation is also maintained after peptide bond formation[25]. Under amino-acid starvation, mistranslation events frequently occur[8]. For instance, UUC or AAU codons are efficiently mistranslated by Leu-tRNA[Leu] or Lys-tRNA[Lys] when cells are cultured in

[1]Department of Chemistry and Biotechnology, Graduate School of Engineering, University of Tokyo, Bunkyo-ku, Tokyo 113-8656, Japan. [2]RNA Systems Biochemistry Laboratory, RIKEN Cluster for Pioneering Research, Wako, Saitama 351-0198, Japan. [3]Department of Computational Biology and Medical Sciences, Graduate School of Frontier Sciences, The University of Tokyo, Kashiwa, Chiba 277-8561, Japan. ✉e-mail: asuteka@chembio.t.u-tokyo.ac.jp; ts@chembio.t.u-tokyo.ac.jp

media lacking Phe or Asn, respectively[26–29]. In these situations, pep-tRNA unstably binds the P-site with the mismatched codon–anticodon interaction and does not efficiently react with incoming aa-tRNA in the A-site. Instead, release factor 2 (RF2) efficiently recognizes the miscoded ribosome and hydrolyzes the pep-tRNA to rescue the ribosome[25,30,31]. RF3 also participates in this mechanism by accelerating the release activity of RF2[32]. Genetic and biochemical analyses revealed that RF3 plays a primary role in quality control of protein synthesis, rather than in translation termination[32].

Even if the ribosome successfully initiates protein synthesis, not all translating ribosomes can reach the stop codon of the mRNA. Several mechanisms are involved in quality control of translation at the elongation stage[33]. If a ribosome translates a truncated mRNA generated by faulty transcription or nucleolytic cleavage, it will stall at the 3' end of the aberrant mRNA. Frameshifts also generate non-stop mRNA, and the ribosome can reach the 3' end of the mRNA. Because ribosomal stalling is detrimental to cell viability, bacteria have several mechanisms to rescue stalled ribosomes, including trans-translation mediated by the tmRNA/smpB complex and alternative systems involving ArfA and ArfB[34–37].

In the early stage of translation elongation, pep-tRNAs frequently dissociate from the ribosome (Fig. 1a), presumably because pep-tRNAs with short nascent chains do not form a sufficiently strong interaction with the ribosomal tunnel. The phenomenon is termed "pep-tRNA drop-off"[38]. The dissociated pep-tRNAs are hydrolyzed into peptide and tRNA by peptidyl-tRNA hydrolase (PTH)[39]. The deacylated tRNA is then recharged to participate in the next round of protein synthesis. PTH is an essential enzyme in *Escherichia coli* and other bacteria[38,40–42]. A temperature-sensitive mutant of PTH, *pth*[ts], was isolated in the 1970s[40]. When this strain is shifted to a non-permissive temperature, high levels of pep-tRNAs accumulate, rapidly depleting aa-tRNAs and leading to defective protein synthesis[40]. Thus, pep-tRNA drop-off is not accidental, but rather a normal event that occurs frequently during the elongation step of protein synthesis. A series of biochemical and genetic studies revealed that RF3, RRF, and EF-G in the GTP form catalyze the dissociation of pep-tRNA from the P-site in a manner similar to ribosome recycling[43–46]. If the ribosome stalls in the early elongation stage at a rare or hungry codon due to amino-acid starvation, pep-tRNA drop-off plays a critical role in rescuing the resultant premature termination complex[38,46–49]. Early studies proposed that this event preferentially occurs on miscoding ribosomes induced by amino-acid starvation or streptomycin treatment[50], leading to the "ribosome editor hypothesis"[50–52]. More recently, Verma et al.[53] investigated the impact of initial ORF sequences on protein synthesis comprehensively, and found that translation efficiency is strongly dependent on 3rd to 5th codon sequences in vitro and in vivo, regardless of their downstream sequence of mRNA and protein. Single-molecule FRET analysis revealed that ribosome stalls at early elongation of the lowly-expressing mRNA construct in a non-canonical state, which is probably induced by the nascent peptide interacting with the ribosome tunnel. It was speculated that the arrested ribosome might be resolved through peptidyl-tRNA drop-off and subsequent ribosome recycling.

If pep-tRNA drop-off occurs in early elongation to ensure translational fidelity, it might be possible to detect erroneous pep-tRNAs whose peptide sequences do not match those encoded by the N-terminal regions of template ORFs. Previously, however, no practical approach was available for characterizing peptide sequences of pep-tRNAs dissociated from the ribosome.

In this study, we establish a method for direct analysis of pep-tRNAs accumulated in *E. coli* cells using mass spectrometry. This technique enables us to profile and characterize the nascent peptide chains of the pep-tRNAs dissociated from the ribosome. Using this approach, we successfully detect more than 5000 molecular weights thought to be nascent peptides of 3–15 residues attached to pep-tRNAs from *pth*[ts] strain. Although about 80% of them matched molecular weights of peptides corresponding to the N-terminal sequences of *E. coli* ORFs, about 20% are derived from noncognate peptides bearing single amino-acid substitutions of the cognate peptides. Moreover, detailed analyses of individual pep-tRNAs isolated from *E. coli* cells reveal that most of the substitutions in the miscoded peptides take place at the C-terminal drop-off site. We further examine this observation using a reporter construct, and demonstrate that non-cognate pep-tRNAs produced by mistranslation rarely participate in the next round of elongation, but dissociate from the ribosome, suggesting that pep-tRNA drop-off is an active mechanism by which the ribosome rejects non-cognate pep-tRNAs in the early elongation. Based on our findings, we propose that pep-tRNA drop-off is a post-peptidyl transfer quality control mechanism that maintains translational fidelity at the early elongation stage. In addition, ribosome profiling of RRF-deficient cells reveals accumulation of ribosomes at the beginnings of mRNAs, providing direct evidence that RRF functions in the early elongation stage to maintain the fidelity of translation by promoting pep-tRNA drop-off.

## Results
### Mass spectrometric analyses of pep-tRNAs dissociated from the ribosome

To study pep-tRNA drop-off, we first constructed a temperature-sensitive *E. coli* strain of *pth* (*pth*[ts]) by introducing a Gly-to-Asp mutation at position 101[39] because of rapid deacylation of pep-tRNA by PTH in wild-type (WT) cells. The *pth*[ts] strain grew at 30 °C, but slower than WT (Fig. 1b). The growth of *pth*[ts] gradually decreased as the culture temperature increased, and the strain was unable to grow at 43 °C (Fig. 1b).

To characterize pep-tRNAs accumulated in the *pth*[ts] strain, we prepared total tRNA fractions under low-temperature acidic conditions from *pth*[ts] cells in logarithmic growth phase cultured at the permissive (30 °C) or incubated at non-permissive (43 °C) temperature for 30 min. To stabilize the acyl bonds of pep- and aa-tRNAs, their N-terminal amino groups were acetylated to yield acetyl-pep-tRNAs (Ac-pep-tRNAs) and acetyl-aa-tRNAs (Ac-aa-tRNAs), respectively (Fig. 1c). To analyze the peptides or amino-acids attached to tRNAs, we digested them with nuclease P1 and alkaline phosphatase to yield acetyl-peptidyl-adenosine (Ac-pep-Ado) and acetyl-aminoacyl-adenosine (Ac-aa-Ado), and then subjected the products to LC/MS (Fig. 1c). Conjugation of adenosines to the peptides ensured that the nascent peptide moieties were attached to the 3' termini of tRNAs (Fig. 1c).

At 30 °C, we clearly detected 18 species of Ac-aa-Ado (Fig. 1d), represented by Ac-Ala-Ado and formyl-Met-Ado (f-Met-Ado) in the mass chromatograms (Fig. 1e). These identifications were confirmed by collision-induced dissociation (CID) (Supplementary Fig. 1ab). Ac-Cys-Ado was not detected in our LC/MS system, probably due to its thiol group. Ac-Ile-Ado and Ac-Leu-Ado were not distinguished because they have identical mass. Although we also detected a tiny amount of acetyl-dipeptidyl-adenosine (Ac-dipep-Ado) (Fig. 1d), almost all tRNAs were aminoacylated under this condition (Fig. 1d). At 43 °C, we found remarkable enrichment of 18 species of Ac-dipep-Ado (Fig. 1f), exemplified by mass chromatograms (Fig. 1e) and CID (Supplementary Fig. 1c) of Ac-Met-Ala-Ado. We note that in this analyses, the abundances of Ac-aa-Ado and Ac-dipep-Ado were normalized against the level of f-Met-Ado, as f-Met-tRNA[iMet] is not a substrate of PTH, and its steady-state level should not be affected by a short period of PTH inactivation[54–56]. To examine whether cellular level of f-Met-Ado is unchanged during 30 min incubation at 43 °C, the peak intensities of f-Met-Ado, Ac-Ala-Ado and Ac-Met-Ala-Ado were normalized by those of dihydrouridine (D) which is a ubiquitous and stable tRNA modification, because it is unlikely that D is affected in this condition. As expected, Ac-Ala-Ado is drastically reduced, whereas Ac-Met-Ala-Ado is markedly accumulated. In contrast, f-Met-Ado does not change (Supplementary Fig. 2a).

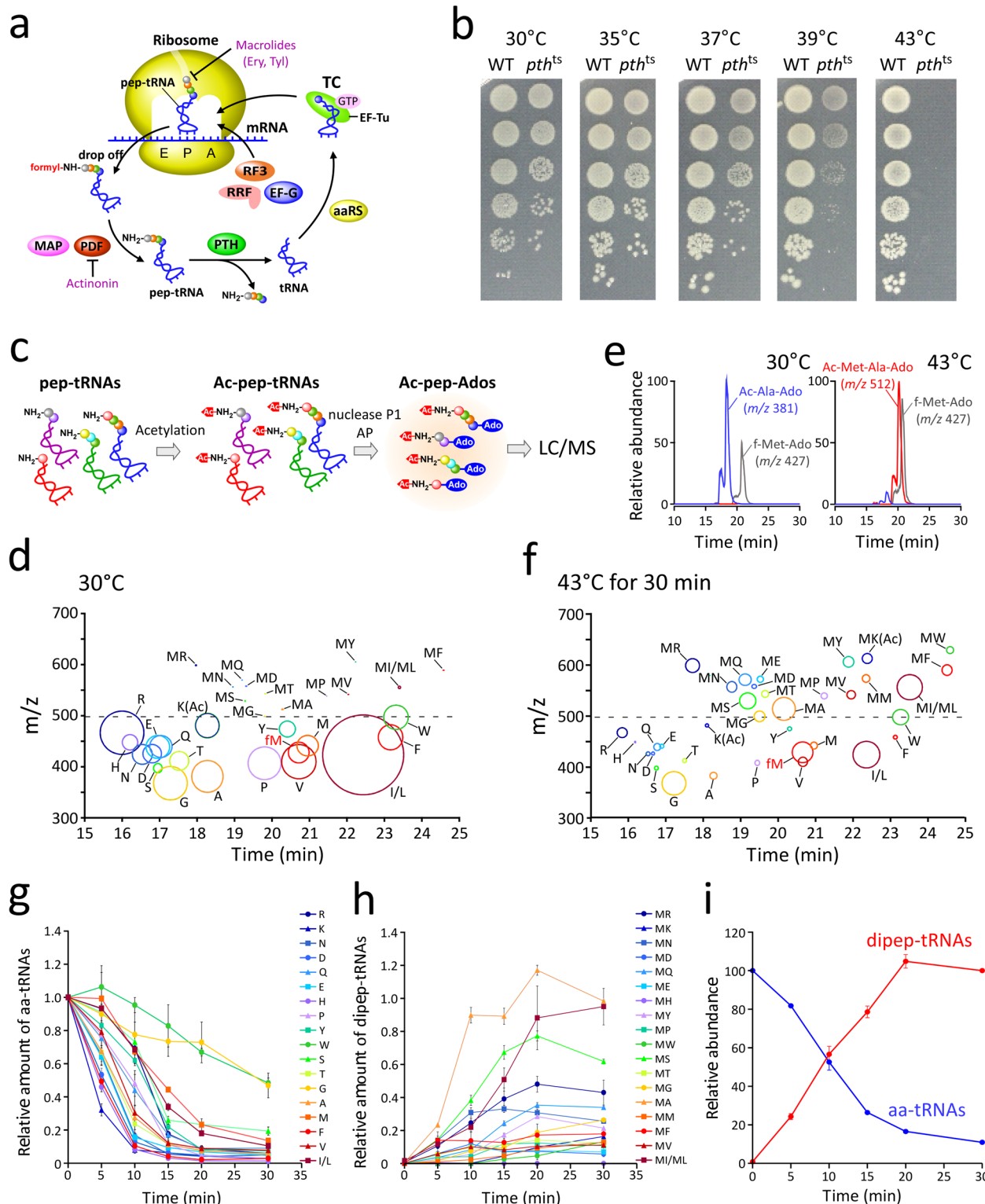

We carefully traced Ac-aa-Ado and Ac-dipep-Ado levels after *pth*^ts was shifted to 43 °C and found that the abundance of Ac-aa-Ado species decreased markedly (Fig. 1g, i); by contrast, the levels of Ac-dipep-Ado species increased drastically over 20 min (Fig. 1h, i). The rates of reduction of aa-tRNAs and accumulation of pep-tRNAs varied among amino-acid species (Fig. 1g, h). Upon PTH inactivation, the levels of most aa-tRNAs plunged sharply, whereas the levels of Ac-Gly-Ado and Ac-Trp-Ado decreased more gradually (Fig. 1g) and Ac-MA-Ado rapidly accumulated (Fig. 1h).

For longer nascent chains, although the detection of tri- and tetra-pep-tRNAs by mass spectrometry is more difficult than that of aa- and dipep-tRNAs because of the diversity in molecular weights of the longer pep-tRNAs, we successfully detected Ac-pep-Ado bearing tripeptides and tetrapeptides in the mass range 424–1000 (Fig. 2a). We observed the accumulation of these pep-tRNAs, as well as dipep-tRNAs, in *pth*^ts incubated at 43 °C, albeit with smaller intensities compared to dipep-Ado species (Fig. 2a). Even at 37 °C, we observed considerable accumulation of pep-tRNAs in *pth*^ts (Fig. 2a). Given that PTH

**Fig. 1 | Pep-tRNA drop-off in *E. coli* cells and mass spectrometric characterization. a** Translation elongation cycle and pep-tRNA drop-off. Ery and Tyl stand for erythromycin and tylosin, respectively. **b** Temperature sensitivity of the *E. coli pth*ts strain. *E. coli* WT and *pth*ts strains were serially diluted (1:10), spotted onto LB plates, and cultured overnight at the indicated temperatures (30, 35, 37, 39, and 43 °C). **c** Preparation scheme of Ac-pep-Ado for LC/MS analysis. AP stands for alkaline phosphatase. **d, f** Bubble chart plot of LC/MS analyses of Ac-aa-Ado and Ac-dipep-Ado species from the *pth*ts strain cultured at 30 °C (**d**) or incubated at 43 °C for 30 min (**f**). Amino-acids of Ac-aa-Ado and dipeptides of Ac-dipep-Ado are denoted by a single letter. The size of each bubble represents relative intensity of peak area in the MS chromatogram normalized against that of f-Met-Ado (fM). **e** LC/MS nucleoside analyses of total tRNAs containing Ac-aa-tRNA and Ac-pep-tRNA prepared from the *pth*ts strain cultured at 30 °C (left) or incubated at 43 °C

for 30 min (right). Mass chromatograms detected for Ac-Ala-Ado (*m/z* 381.15, blue), Ac-Met-Ala-Ado (*m/z* 512.19, red), and f-Met-Ado (*m/z* 427.14, gray). **g** Time-course analyses of aa-tRNAs in the *pth*ts strain after temperature shift to 43 °C. The calculation method of each relative value is described in Methods. Data are presented as means±s.d. of three independent experiments. **h** Time-course analyses of dipep-tRNAs in the *pth*ts strain after temperature shift to 43 °C. The calculation method of each relative value is described in Methods. Data are presented as means±s.d. of three independent experiments. **i** Overall change in aa-tRNAs (blue) and dipep-tRNAs (red) in *pth*ts cells after the temperature shift to 43 °C. The calculation method of each relative value is described in Methods. Data are presented as means± s. d. of three independent experiments. Source data are provided as a Source Data file.

activity persisted even at 37 °C in the *pth*ts strain, much higher levels of pep-tRNA than observed here should dissociate from the ribosome. These findings indicate that pep-tRNA drop-off is not an accidental event that arises under heat stress, but rather an inevitable event that takes place frequently under normal growth conditions, i.e., at 37 °C. Taken together, these observations demonstrated that aa-tRNAs are rapidly depleted, whereas pep-tRNAs accumulate, upon PTH inactivation.

### Effect of antibiotics on the pep-tRNA drop-off

Because most nascent peptides of the pep-tRNAs detected in this study were acetylated (Fig. 1e, f), the N-termini of these peptides must have been deformylated after dissociation from translating ribosomes. To confirm this, we treated *pth*ts cells with actinonin, an antibiotic that inhibits peptide deformylase (PDF) (Fig. 1a)[57]. As expected, formylated-dipep-Ado species (f-Met-Arg-Ado as a representative example), appeared following actinonin treatment, whereas Ac-Met-Arg-Ado disappeared (Supplementary Fig. 2b). The CID analyses confirmed their chemical structures (Supplementary Fig. 1d, e). This observation suggests that formylated pep-tRNAs dissociated from the ribosome are efficiently deformylated by PDF. In fact, a majority of the dipep-tRNAs detected in this study were acetylated at the N-terminus, suggesting that most of pep-tRNAs extracted from *pth*ts cells are ones dissociated from the ribosomes, not pep-tRNAs in elongating ribosomes.

Other chemical tools led us to investigate the nature of pep-tRNA drop-off. Macrolides are antibiotics that bind in the nascent peptide exit tunnel of the ribosome and inhibit protein synthesis[58]. The macrolide erythromycin stimulates the dissociation of pep-tRNA and increases the temperature sensitivity of *pth*ts cells[41,59]. Although overall translation is severely impaired by macrolides, it is known that certain nascent chains continue to elongate even in the presence of macrolides, and sometimes cause ribosome stalling, leading to translational regulation involved in acquisition of antibiotic resistance[60–65]. We examined the impact of macrolides on the pep-tRNA drop-off. Erythromycin (Ery) and tylosin (Tyl) are well-studied macrolides with 14- and 16-membered lactone rings, respectively. For this experiment, we knocked out *tolC* gene in the *pth*ts background (Δ*tolC*/*pth*ts) to increase sensitivity to these antibiotics[66]. In the presence of Ery, pep-tRNAs with tri- and tetrapeptides accumulated to high levels (Fig. 2a, b), whereas dipep-tRNAs were the major species by Tyl (Fig. 2a, b). These results are nicely explained by structural studies of ribosomes bound by these macrolides[67]. Ery blocks elongation of peptides with chain lengths up to six to eight amino-acids, whereas Tyl inhibits elongation near the peptidyl transferase center and produces shorter peptides ranging from dimers to tetramers[68], as Tyl has an elongated saccharide chain that reaches the peptidyl transferase center[67].

Next, we titrated Tyl to determine the minimum concentration that would dissociate as much of the elongating dipep-tRNAs as possible. The relative abundance of dipep-tRNAs was saturated when the Δ*tolC*/*pth*ts strain was treated with Tyl at concentrations higher than 0.1 mg/ml (Fig. 2c). Assuming that the amount of dipep-tRNA in

Tyl-treated samples represents the maximum level of dipep-tRNA dissociated from ribosome in the cells, we could roughly estimate the drop-off efficiency of dipep-tRNAs under normal growth conditions (in LB medium without drug at 37 °C). We measured each species of dipep-tRNAs accumulated in cells cultured at 37 °C in the absence or presence (0.1 mg/ml) of Tyl (Fig. 2d). Apparent drop-off frequency was calculated by dividing the relative abundance of dipep-tRNA without Tyl by that with Tyl. Frequencies of dipep-tRNA drop-off were quite divergent, ranging from 2.4% to 23.2% depending on the peptides (Fig. 2d). The average frequency was 10.5% (Fig. 2d), indicating that pep-tRNAs dissociate from the ribosome at a high frequency even under normal culture conditions (in LB medium at 37 °C). In biochemical study[69], drop-off efficiencies of a couple of dipep-tRNAs were measured, showing that fMet-Phe and fMet-Lys have rapid dissociation rate versus fMet-Ile. This is consistent with our data showing that the drop-off efficiencies of Met-Phe-tRNA and Met-Lys-tRNA were higher than that of Met-Ile-tRNA (Fig. 2d).

### Characterization of factors involved in pep-tRNA drop-off

We next characterized factors involved in the pep-tRNA drop-off. Early studies showed that RF3 and RRF play critical roles in the dissociation of pep-tRNA from the P-site (Fig. 1a)[43–46]. To confirm the involvement of RF3 in this process, we constructed a knockout strain of RF3 (encoded in *prfC*) in the *pth*ts background (Δ*prfC*/*pth*ts). The RF3 knockout alleviated the temperature sensitivity of the *pth*ts strain, enabling it to grow even at 39 °C (Fig. 3a). We next monitored Ac-aa-Ado and Ac-dipep-Ado species in the Δ*prfC*/*pth*ts strain incubated at 43 °C. Relative to the *pth*ts strain, Ac-aa-Ado species slowly decreased and Ac-dipep-Ado species accumulated to a less extent in the Δ*prfC*/*pth*ts strain (Fig. 3b), indicating that pep-tRNA drop-off was suppressed in the absence of RF3.

We were unable to knock out RRF (encoded in *frr*) because it is an essential factor in bacteria[70]. Instead, we overexpressed the *frr* gene via the arabinose promoter in the *pth*ts background; in the resultant strain, the level of RRF expression was controlled by arabinose concentration (Fig. 3c). The expression of RRF increased the temperature sensitivity of the *pth*ts strain. Growth of the strain decreased at 37 °C even in the presence of 0.1 mM arabinose, indicating that RRF expression induces pep-tRNA drop-off. We next analyzed Ac-aa-Ado and Ac-dipep-Ado species in this strain upon induction of RRF by arabinose. In the presence of arabinose, the levels of Ac-aa-Ado species decreased faster (Fig. 3d), and Ac-dipep-Ado species accumulated much faster to higher levels than in the absence of arabinose (Fig. 3d). The tendency was more pronounced at 1 mM than at 0.1 mM arabinose (Fig. 3d). Taken together, our mass spectrometric approach clearly demonstrated that both RF3 and RRF actively participate in pep-tRNA drop-off in the early stage of translation elongation.

If RRF actively participates in dissociation of the pep-tRNAs and recycles the ribosome in the early elongation, ribosome occupancy near the initiation sites in mRNAs should change upon acute inactivation of RRF. To explore this possibility, we constructed a temperature-sensitive mutant of RRF, *frr*ts[71] (Fig. 3e), and performed

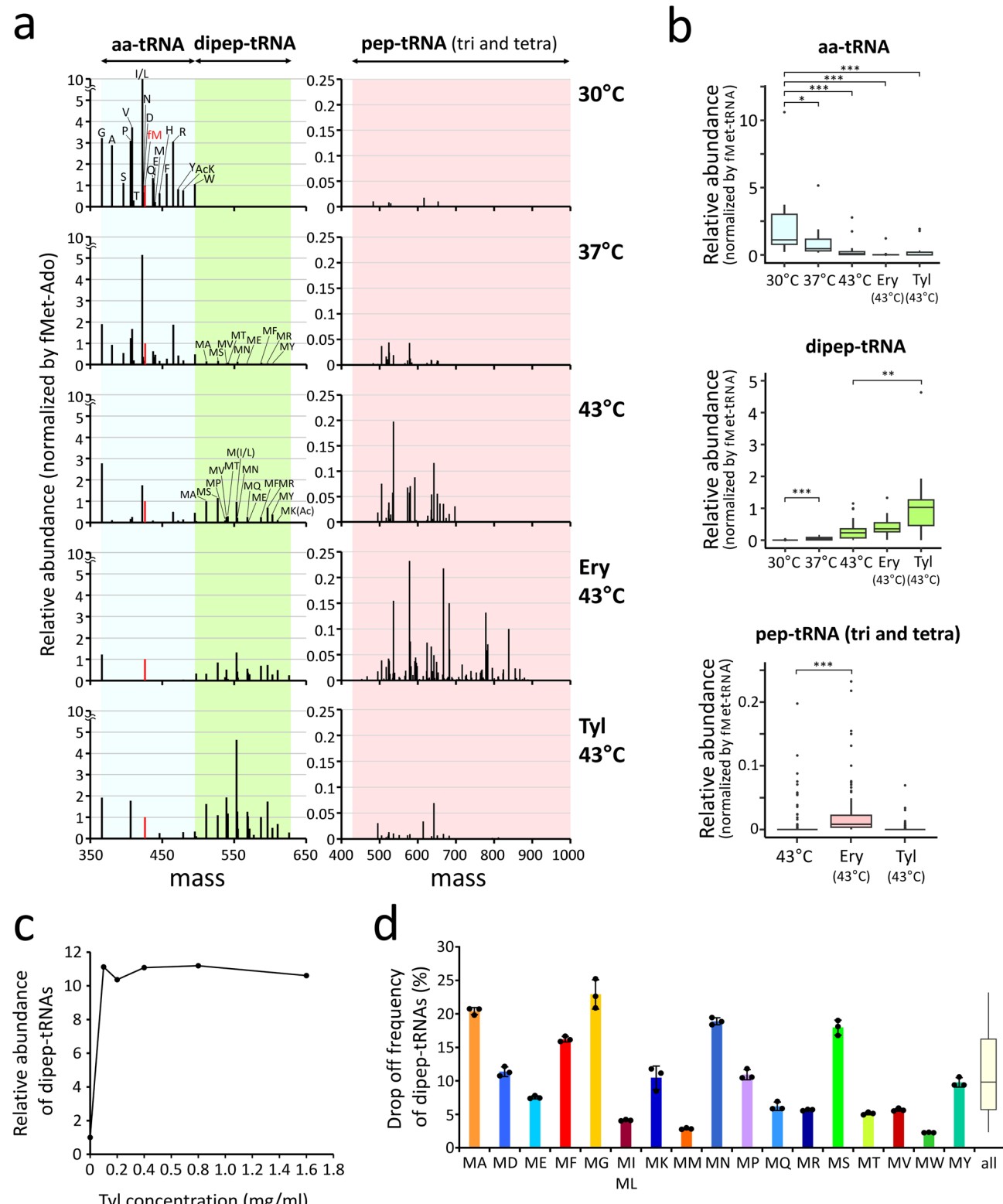

ribosome profiling of this strain after 30 min of incubation at 43 °C[72,73]. The distribution of read lengths of ribosome-protected mRNA fragments (RPFs) was reproducible in each duplicate sample (Supplementary Fig. 3a). Comparison of metagene analyses of RPFs around translation initiation sites between WT and *frr*^ts strains revealed that *frr*^ts cells exhibited a global increase in ribosome occupancy in the region where the P-site of the ribosome had proceeded to the second and fourth codons (Fig. 3f), namely, in the region where the nascent

chain had extended to di- to tetrapeptides. On the other hand, we observed no significant difference in the downstream region (Supplementary Fig. 3b). To confirm this result, we analyzed the ribosome-profiling data of *E. coli* cells in which RRF is rapidly depleted by the transcriptional shut-off combined with rapid degradation of RRF[74]. The metagene analyses revealed that the RPFs clearly accumulated in the beginning of translation, and that this accumulation increases over time of RRF depletion (Supplementary Fig. 3c). This is completely

**Fig. 2 | Effect of macrolides on pep-tRNA drop-off, and estimation of the drop-off frequency in *E. coli* cells. a** Mass profiles of mono- (aa-tRNA), di-, tri-, and tetra-pep-Ado in the *ΔtolC/pth*[ts] strain incubated for 30 min at 30 °C (top panels), 37 °C (second panels), 43 °C (third panels) with no drug, erythromycin (fourth panels), or tylosin (bottom panels). Mass ranges for mono- (366–495 Da), di- (497–626 Da), and tri- and tetra-pep-Ado (424–1000 Da) are indicated. The relative abundance of each peak for Ac-aa-Ado and Ac-pep-Ado species was normalized against the peak area of f-Met-Ado (red bar). **b** Box plots of relative abundance of aa-tRNAs (top, *n* = 18), dipep-tRNAs (middle, *n* = 18), and tri- and tetra-pep-tRNAs (bottom, *n* = 118) in the *ΔtolC/pth*[ts] strain cultured in the same conditions as described in Fig. 2a. Box plots show median (central line), upper and lower quartiles (box limits), maximum and minimum (whiskers). All data are normalized against the peak area of f-Met-Ado. *$p < 0.05$, **$p < 0.01$, and ***$p < 0.0001$ (two-sided Wilcoxon rank–sum test).

**c** Relative abundance of dipep-tRNAs accumulated in the *ΔtolC/pth*[ts] strain treated with various concentrations of Tyl at 37 °C. The relative intensity of each dipep-tRNA is normalized against that of f-Met-tRNA, and the normalized intensities of all dipep-tRNAs are summed for each Tyl concentration. **d** Drop-off frequency of dipep-tRNA in the *ΔtolC/pth*[ts] strain incubated at 37 °C for 30 min. For each species of dipep-tRNA, the drop-off frequency was calculated from the relative abundance of dipep-tRNA in the absence of Tyl, divided by the relative abundance in the presence of 0.1 mg/ml Tyl. The color code for each bar is the same as that in Fig. 1h. The average value and distribution of the drop-off frequencies of 17 dipep-tRNA species are shown in the box plot (*n* = 17) on the right. Box plot shows median (central line), upper and lower quartiles (box limits), maximum and minimum (whiskers). Data are presented as means ± s.d. of three independent experiments. Source data are provided as a Source Data file.

consistent with our observation. These findings fully support our speculation that ribosome accumulate near the initiation sites upon inactivation of RRF, suggesting that RRF actively dissociates pep-tRNAs and recycles ribosomes at the early elongation stage.

## Non-cognate nascent peptides detected by the pep-tRNA profiling

We next analyzed pep-tRNAs with longer peptides consisting of 3–14 amino-acid residues. Due to the large variation in their sequences and molecular masses, it was difficult to detect each peptide individually, as each was present in only a small quantity. To increase the detection sensitivity of each individual peptide as much as possible, we hydrolyzed the pep-tRNAs to release the peptide moieties and then performed a highly sensitive analysis based on capillary liquid chromatography nanoESI-MS (Fig. 4a). We successfully detected and profiled the molecular weights corresponding to those of the nascent peptides of the pep-tRNAs obtained from the *pth*[ts] strain (Fig. 4b), which clearly accumulated over time after shifting to 43 °C. Although most of the MS peaks detected here were not intense enough to perform MS/MS-based sequence analysis, we could determine the sequence of several abundant peptides with strong signals by assigning their CID spectra (Supplementary Fig. 1f). To determine whether the peptides are actually derived from the pep-tRNAs, we hydrolyzed the pep-tRNAs with or without [18O]-labeled water (50%) and performed MS analysis (Fig. 4a). In the presence of [18O] water, the nascent peptides had molecular masses 2 Da larger than their natural ones (Supplementary Fig. 4a, b, c), clearly demonstrating that the peptides analyzed in this method were actually derived from the pep-tRNAs. Thus, in this analysis, we detected the peptides derived from pep-tRNAs based on their accurate mass without CID data.

From the mass list obtained 30 min after the induction, we detected 14,144 individual peaks (Fig. 4c). We first searched for peptides of 3–14 residues that corresponded to the initial sequences of *E. coli* ORFs (Fig. 4c). Because the N-terminal Met of the pep-tRNAs can be removed by methionine aminopeptidase (MAP) (Fig. 1a), we added a series of N-terminal peptides missing the first Met to the reference list of *E. coli* ORFs harboring Gly, Ala, Ser, Thr, Cys, Pro, Val, Asn, Asp, Leu or Ile at the second position, according to the substrate recognition rule for MAP[75,76]. Among 14,144 peaks, we clearly detected 4,466 peaks that exactly matched the molecular weights of the initial peptides (3–14 aa.) of *E. coli* ORFs within a mass error range of 5 ppm (Fig. 4c). We refer to these as "cognate peptides". In the remainder of the peaks, we noticed the presence of non-cognate peptides resulting from amino-acid substitutions. Hence, we prepared another list of references consisting of the initial peptides (3–14 amino-acids) of *E. coli* ORFs bearing single amino-acid substitution. By matching to this reference, we detected 1171 peaks as the molecular weights of non-cognate peptides with single amino-acid substitution (Fig. 4c). Of the remaining 8507 peaks, 8450 peaks were not assigned to peptides because they did not match any mass constituted of proteogenic amino-acids; 57 peaks appeared to be derived from peptides with multiple substitutions and/or truncated peptides (Fig. 4c). We were surprised to observe that about one-fifths (20.8%) of the detected peptides were non-cognate with single amino-acid substitutions of cognate peptides (Fig. 4d).

Next, we compared the molecular mass distributions between cognate and non-cognate peptides (Fig. 4e). The median masses of cognate and non-cognate peptides were 745.4 and 646.2, respectively; the non-cognate peptides were significantly shorter than the cognate peptides (Fig. 4f). We confirmed the reproducibility of this result by performing pep-tRNA profiling of triplicate samples (Supplementary Fig. 5). The median values of molecular weights for the cognate and non-cognate peptides were 750.9 ± 6.4 and 646.0 ± 9.5, respectively, suggesting that non-cognate peptides with single substitution tend to have a smaller molecular weight than the cognate peptides. These observations indicate that mistranslation occurs frequently in early elongation, and that the resultant erroneous pep-tRNAs can be actively excluded through pep-tRNA drop-off.

## Deep analyses of individual pep-tRNAs

In the pep-tRNA profiling, we characterized nascent peptides of pep-tRNAs based on their accurate masses without CID data, but it is necessary to investigate peptide sequences of pep-tRNAs to obtain more detailed information about pep-tRNA drop-off. To this end, we isolated each tRNA species with peptide from *pth*[ts] cells in logarithmic growth phase incubated at 43 °C for 30 min using reciprocal circulating chromatography (RCC)[77] and precisely analyzed their nascent peptides (Fig. 5a). Because RCC requires heat denaturation, we found it challenging to isolate individual pep-tRNAs without deacylation. Hence, we optimized the RCC procedure to minimize the deacylation of pep-tRNAs (see Methods) and then evaluated the deacylation frequency of each species of pep-tRNAs under the optimized RCC conditions (Supplementary Fig. 6a). Each species of Ac-dipep-Ado and Ac-aa-Ado was quantified by LC/MS. Most dipep-tRNAs retained more than 80% of their peptides after the heat treatment for optimized RCC (Supplementary Fig. 6a); only Ac-Met-Asn-tRNA was slightly more deacylated, although 55% still remained (Supplementary Fig. 6a). Because Ac-Asn-tRNA behaves similarly to the Ac-Met-Asn-tRNA (Supplementary Fig. 6a), the side chain of Asn might accelerate the deacylation reaction during RCC procedure.

Using optimized RCC, we successfully isolated 42 species of tRNAs conjugated with peptides (Fig. 5a and Supplementary Fig. 6b); this number represents the set of all elongator tRNAs that can be separated by RCC (see Methods). To confirm the attachment of acetylated peptides to the 3′ termini of tRNAs, we digested the isolated pep-tRNAs with RNase T₁ and subjected the digested samples to RNA-MS (Fig. 5a). In the case of tRNA[Ala2], its 3′-terminal fragment (CUC-CACCA) with an acetylated Met-Ala dipeptide was clearly detected by CID (Supplementary Fig. 6c). We also detected the same fragment for acetyl-Ala, as well as its deacylated form (Supplementary Fig. 6c).

The fact that the bulk pep-tRNA profiling indicated a number of non-cognate peptides with single amino-acid substitution (Fig. 4d) led

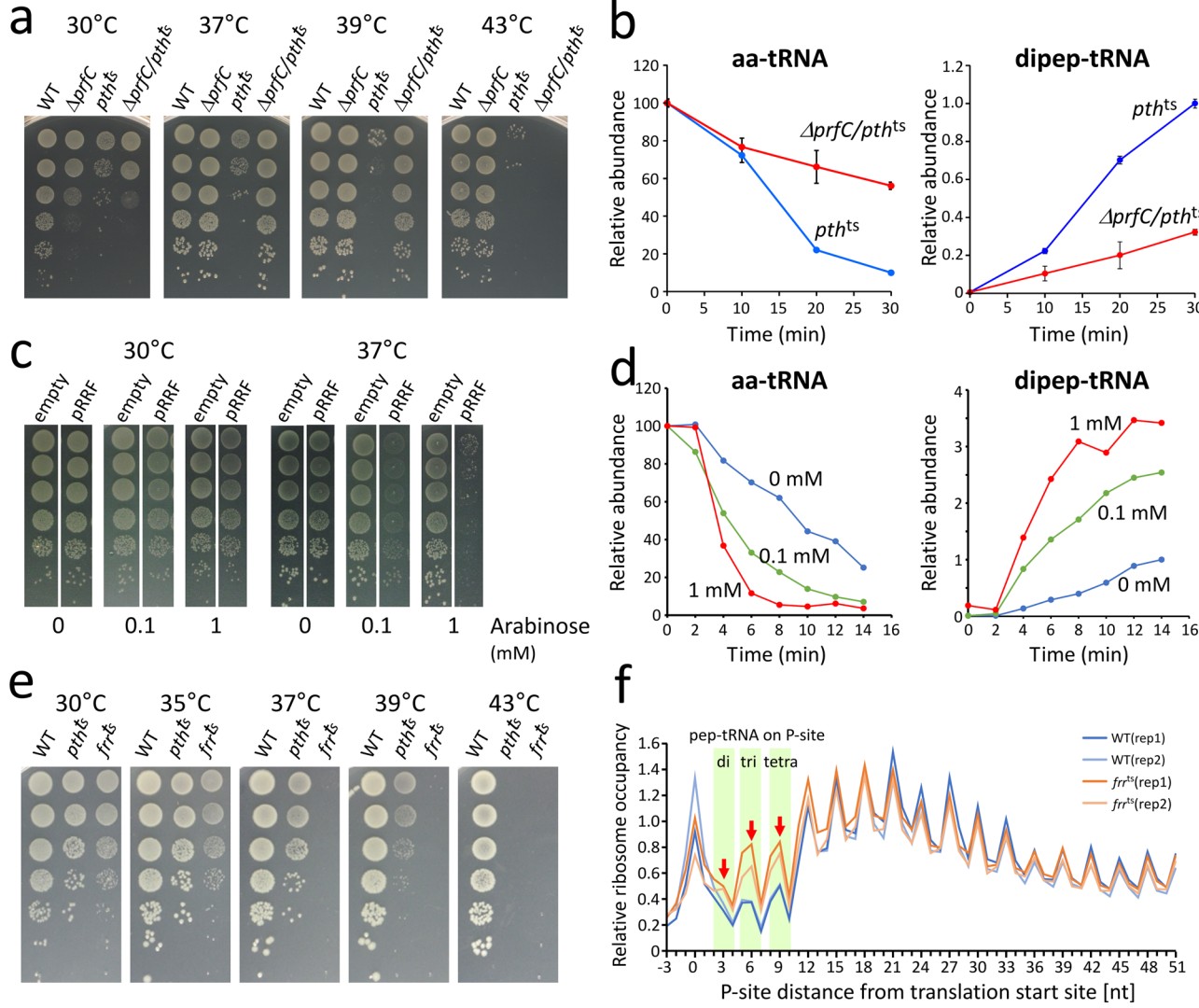

**Fig. 3 | Characterization of factors involved in pep-tRNA drop-off in the early elongation stage. a** Depletion of RF3 (encoded by *prfC*) alleviates the temperature sensitivity of the *pth*ts strain. WT, *ΔprfC*, *pth*ts, and *ΔprfC/pth*ts strains were serially diluted (1:10), spotted onto LB plates, and cultivated overnight at the indicated temperatures (30, 37, 39, and 43 °C). **b** Overall change in aa-tRNAs (left panel) and dipep-tRNAs (right panel) in *pth*ts (blue) and *ΔprfC/pth*ts cells (red) after the temperature shift to 43 °C. The calculation method of each relative value is described in Methods. Data are presented as means±s.d. of three independent experiments. **c** Overexpression of RRF aggravates the temperature sensitivity of the *pth*ts strain. *pth*ts cells transformed with empty vector (empty) or expression vector for *frr* (pRRF) were serially diluted (1:10), spotted onto LB plates containing the indicated concentrations of arabinose (0, 0.1, and 1 mM), and cultivated overnight at the indicated temperatures. **d** Overall change in aa-tRNAs (left panel) and dipep-tRNAs

(right panel) in the *pth*ts strain that expresses different levels of RRF. The cells were cultured in the presence of 0 mM (blue), 0.1 mM (green), and 1 mM (red) arabinose at 30 °C, and then shifted to 43 °C and sampled every 2 min for 14 min. The calculation method of each relative value is described in Methods. **e** Temperature sensitivity of the *frr*ts strain. WT, *pth*ts, and *frr*ts strains were serially diluted (1:10), spotted onto LB plates, and cultivated overnight (the time is longer than that of (**a**)) at the indicated temperatures (30, 35, 37, 39, and 43 °C). The pictures of the spots of WT and *pth*ts strains are shared with Fig. 1b. **f** Metagene analyses of RPFs mapped around the translation initiation sites of *E. coli* ORFs for WT (replicate 1, blue and replicate 2, pale blue) and *frr*ts (replicate 1, orange and replicate 2, pale orange) strains incubated at 43 °C for 30 min. The positions of ribosome containing di-, tri-, and tetra-pep-tRNAs on P-site are indicated based on the distance between the P-site and the translation start site. Source data are provided as a Source Data file.

to the following two possibilities for how these non-cognate pep-tRNAs were produced. One was the possibility that a misaminoacylated tRNA read the corresponding codon. The other was that a correct aminoacyl-tRNA miscoded non-cognate codon. To examine these possibilities, we first surveyed a 3′-terminal fragment with misacylation of a non-cognate amino-acid (Fig. 5a and Supplementary Fig. 7). However, we obtained no evidence of misacylation of any isolated tRNA (Supplementary Fig. 7), implying that non-cognate pep-tRNAs are produced by miscoding rather than misacylation.

Next, each isolated pep-tRNA was hydrolyzed to release its nascent peptides, which were then analyzed by LC/MS/MS using capillary LC/nanoESI-MS (Fig. 5a). The peptides were assigned by

database search using peptide mass fingerprinting (MASCOT). We prepared an original database based on the initial sequences (4−17 aa.) of *E. coli* ORFs, including a series of peptides without the first Met, by taking account of the substrate specificity of MAP[75,76]. In the case of pep-tRNA[Gln1] (Fig. 5b), we detected 55 peptides derived from the initial sequences of 48 genes (Supplementary Data 1). The C-terminal Gln was confirmed to be present in all detected peptides which were unequivocally determined by assignment of the product ions of their CID spectra (Supplementary Fig. 8). Strikingly, about half of the detected peptides (25 peptides) were assigned to non-cognate peptides with single amino-acid substitutions (Fig. 5b and

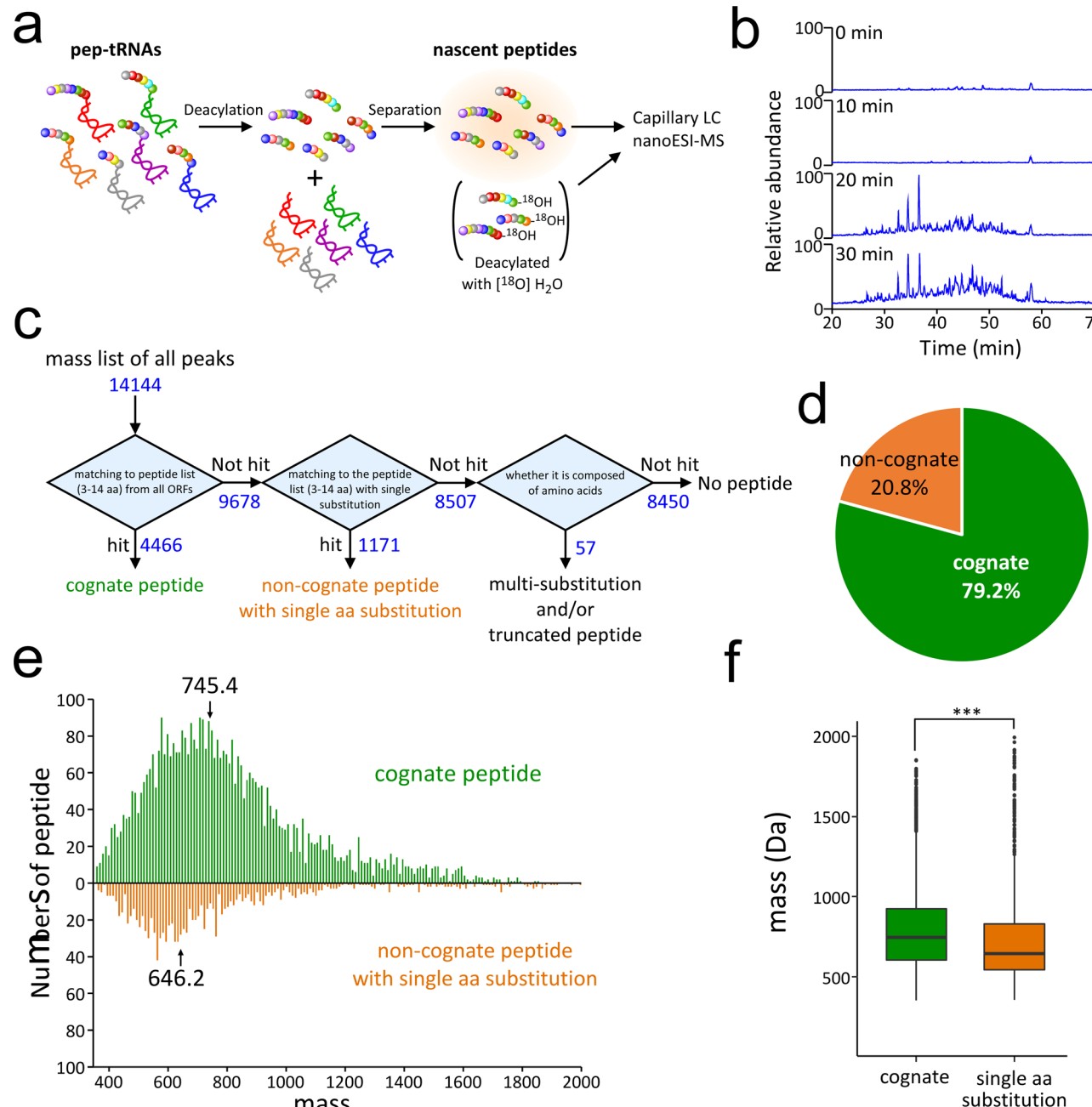

**Fig. 4 | Profiling of nascent peptides of pep-tRNAs dissociated from the ribosome in the *pth*^ts strain. a** Preparation scheme for nascent peptides attached to pep-tRNAs for capillary LC nanoESI-MS analysis. The pep-tRNAs extracted from the cell are hydrolyzed into nascent peptides and tRNAs (deacylation), followed by separation of tRNA using anion exchange resin (separation). The nascent peptides are subjected to LC/MS. To label the C-termini of the peptides, pep-tRNAs are hydrolyzed with [$^{18}$O]-labeled water in the deacylation step. **b** Total ion chromatograms (TICs) of nascent peptides from pep-tRNAs accumulated in *pth*^ts cells incubated at 43 °C for the indicated times. **c** Workflow for the pep-tRNA profiling. The mass list of 14,144 peaks detected by the LC/MS analyses are matched to the peptides (4,466 peaks) and non-cognate peptides (1,171 peaks) with single amino-acid substitution. **d** Proportion of cognate and non-cognate pep-tRNAs identified by pep-tRNA profiling. **e** Histograms of peptide masses assigned to cognate (green) and non-cognate (orange) pep-tRNAs by pep-tRNA profiling. Median masses of cognate and non-cognate peptides were 745.4 and 646.2, respectively. **f** Boxplots of peptide masses assigned to cognate ($n = 4466$, green) and non-cognate ($n = 1171$, orange) pep-tRNAs by pep-tRNA profiling. Boxplots show median (central line), upper and lower quartiles (box limits), maximum and minimum (whiskers). ***$p < 0.0001$ (two-sided Wilcoxon rank–sum test). Source data are provided as a Source Data file.

Supplementary Data 1), while the remaining half were assigned to cognate peptides (Fig. 5b and Supplementary Data 1). Curiously, all single amino-acid substitutions in the 25 non-cognate peptides lay at the C-termini, drop-off sites of the respective genes (Supplementary Data 1).

We extended the investigation to peptides attached to other tRNA species. Up to 72 peptides from the individual pep-tRNAs were assigned to the N-terminal sequence of *E. coli* ORFs with or without a

single amino-acid substitution (Fig. 5c: the nomenclatures, C$_{-3-0}$X, for the non-cognate pep-tRNA types are described in the next section). As observed in the bulk profiling of pep-tRNA (Fig. 4d), more than one-fifths of the nascent peptides were non-cognate peptides with single amino-acid substitution (Fig. 5c, d). All peptides derived from isolated pep-tRNAs were assigned by their CID spectra. Among the 42 tRNA species, we identified 713 nascent peptides, 409 (57.4%) cognate and 304 (42.6%) non-cognate ones (Fig. 5d). At the N-termini, 322 of the

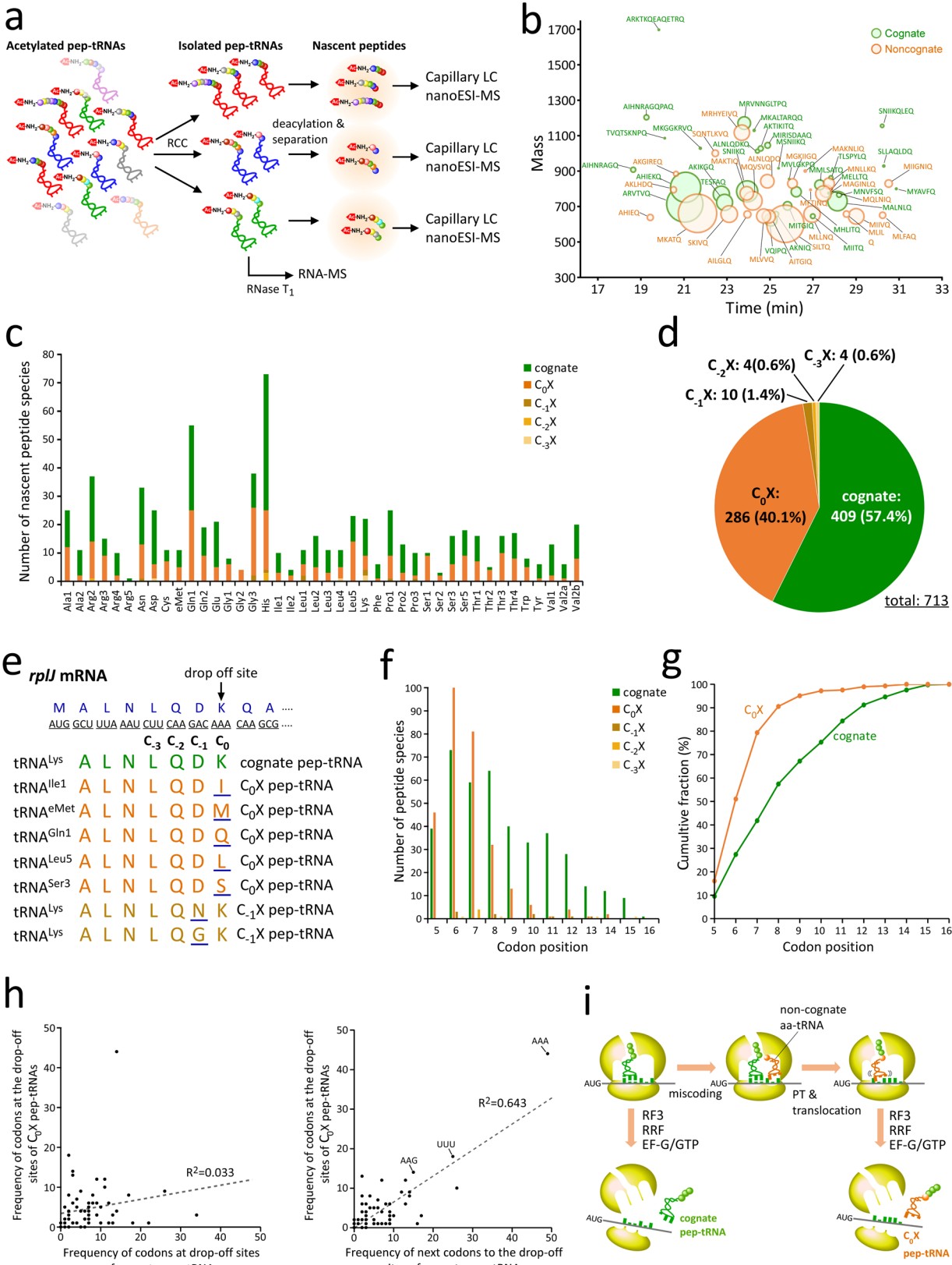

peptides had Met, whereas the remaining 391 lacked Met due to the action of MAP (Supplementary Data 1).

To confirm our assignment of the peptides, we constructed several strains in the *pth*^ts background lacking the genes that serve as template mRNAs for the detected pep-tRNAs. We then isolated several pep-tRNA species from each strain by RCC and

analyzed their nascent peptides. In the case of the *lpp* gene, the cognate peptide MKATKL attached to tRNA^Leu1 was clearly detected in *pth*^ts but completely disappeared in Δ*lpp* (Supplementary Fig. 9a). We also detected two non-cognate peptides attached to tRNAs for Ile2 and Thr1 in *pth*^ts, but failed to detect them in the corresponding tRNAs isolated from Δ*lpp*

**Fig. 5 | Characterization of individual pep-tRNAs and mechanism of pep-tRNA drop-off in the early elongation stage. a** Preparation scheme for nascent peptides attached to individual tRNA species for LC/MS analyses. **b** Bubble chart plot of LC/MS analyses of nascent peptides from pep-tRNA$^{Gln1}$ isolated from the *pth*$^{ts}$ strain incubated at 43 °C for 30 min. Peptide sequences assigned to cognate (green) and non-cognate (orange) pep-tRNAs are indicated. Size of each bubble indicates the peak area of the mass chromatogram for the corresponding peptide. **c** Number of nascent peptides for each individual pep-tRNA determined by LC/MS/MS analyses. Cognate, $C_0X$, $C_{-1}X$, $C_{-2}X$, and $C_{-3}X$ pep-tRNAs are shown in green, orange, brown, gold, and beige, respectively, as indicated in the panel. **d** Proportion of cognate and non-cognate pep-tRNAs determined by LC/MS/MS analyses. Color code is the same as in Fig. 5c. The number and percentile of each type are shown in the pie chart. **e** Nascent peptides from isolated pep-tRNAs

assigned to *rplJ* mRNA. The eight heptamer peptides are dissociated from the eighth codon (AAA) of *rplJ* mRNA. tRNA species and type of pep-tRNAs are shown on the left and right sides, respectively. Misincorporated amino-acids are underlined. Color code is the same as in Fig. 5c. **f** Type-specific distribution of pep-tRNAs at the codon position of the drop-off site. Color code is the same as in Fig. 5c. **g** Cumulative fraction of cognate (green) and non-cognate (orange) pep-tRNAs in codon positions from the initiation site. **h** Scattered plots for frequency of codons at drop-off sites between cognate and $C_0X$ pep-tRNAs (left panel), and between the codons next to the drop-off sites of cognate pep-tRNAs and $C_0X$ pep-tRNAs (right panel). The coefficient of determination ($R^2$) is indicated in each panel. $R^2$ in the case of excluding AAA codon is 0.332. **i** Mechanistic insight into pep-tRNA drop-off. Source data are provided as a Source Data file.

(Supplementary Fig. 9a). Likewise, we also ensured the absence of any cognate and non-cognate peptides in the respective knockout strains (Supplementary Fig. 9b–e). These data clearly validated our assignment of the nascent peptides derived from the isolated pep-tRNAs.

## Characterization of the nascent peptides

We investigated the characteristic features of peptides from the pep-tRNAs. The detected nascent peptides were sorted by mRNAs which serve as templates for the pep-tRNAs (Supplementary Data 1). From the *rplJ* mRNA, we detected 20 nascent peptides ranging from 5 to 13 residues (Supplementary Data 1). Fourteen of them were non-cognate peptides with a single amino-acid substitution, whereas the remaining six were cognate peptides. Eight heptamer peptides dissociated from the same 8th position (AAA codon) of *rplJ* mRNA (Fig. 5e). All of these peptide sequences were confirmed by CID (Supplementary Fig. 10). The cognate peptide ALNLQDK detected from pep-tRNA$^{Lys}$ (Fig. 5e) corresponds to the N-terminal sequence of *rplJ*, whose N-terminal Met is removed by MAP. In addition, we detected seven non-cognate peptides with single amino-acid substitutions (Fig. 5e). In the five tRNAs for Ile1, eMet, Gln1, Leu5, and Ser3, the single amino-acid substitution took place at the C-terminal drop-off site (position 0) from the original Lys residue to its respective amino-acid (Fig. 5e). We refer to these as $C_0X$ type (Fig. 5c, d, e and Supplementary Data 1). In tRNA$^{Lys}$, the D-to-N or D-to-G substitution was detected at the penultimate site (position −1) from the C-terminus of the peptide (Fig. 5e). We call them $C_{-1}X$ type (Fig. 5c, d, e and Supplementary Data 1). We also detected small numbers of non-cognate pep-tRNAs with single amino-acid substitution at positions −2 and −3 from the drop-off site; we refer to these as the $C_{-2}X$ and $C_{-3}X$ types, respectively (Fig. 5c, d and Supplementary Data 1). The $C_0X$ pep-tRNAs dissociate from the ribosome following a miscoding event that happens at the drop-off site, whereas the other types are generated by mistranslation in the previous rounds of translation. We classified all the other non-cognate pep-tRNAs based on the substituted sites (Fig. 5c, d). Thus, a total of 713 peptides from 42 tRNA species were classified into five groups: 409 cognate, 286 $C_0X$, 10 $C_{-1}X$, four $C_{-2}X$, and four $C_{-3}X$ (Fig. 5d). The $C_0X$ type occupies the vast majority of the non-cognate peptides (94%), indicating that dissociation of non-cognate pep-tRNAs is more likely to occur immediately after the miscoding event.

We then compiled each type of peptide according to the codon positions of the drop-off site (Fig. 5f). The cognate peptides were widely distributed among codon positions up to the 16th, whereas $C_0X$-type non-cognate peptides were narrowly distributed in the early codon positions. Over 95% of $C_0X$ peptides were detected between the fifth and ninth positions, demonstrating that non-cognate pep-tRNAs dissociate from the ribosome in the early elongation stage. This tendency was confirmed by the cumulative fractions of cognate

versus $C_0X$ peptides in codon positions (Fig. 5g). This result is consistent with the bulk pep-tRNA profiling data showing that the non-cognate peptides were significantly shorter than the cognate peptides (Fig. 4e, f). These observations suggest that fidelity of translation is actively maintained by excluding non-cognate pep-tRNAs via the drop-off mechanism in the early stage of elongation.

To characterize codon frequency at drop-off sites, we next counted codons at the drop-off sites for both cognate (Supplementary Fig. 11a and Supplementary Data 1) and $C_0X$ pep-tRNAs (Supplementary Fig. 11b and Supplementary Data 1). Frequencies of codons used at the drop-off sites were distributed unevenly in both cognate and $C_0X$ pep-tRNAs, probably due to a strong bias in codon usage in the transcriptome. We then carried out RNA-seq to obtain the transcriptome profile of the *pth*$^{ts}$ strain and compiled actual codon usage for the 5th to 17th positions of all expressed mRNAs based on transcripts per million (TPM) from RNA-seq. We plotted the frequencies of codons at the drop-off sites against codon usage in the early positions in the transcriptome for cognate (Supplementary Fig. 11c) and $C_0X$ pep-tRNAs (Supplementary Fig. 11d). Several C-starting codons (CAU, CAA, CCG etc.) were enriched for the cognate pep-tRNAs regardless of their infrequent usage in the transcriptome (Supplementary Fig. 11c), whereas AU-rich codons (AAA, UUU, AAG etc.) were enriched for the $C_0X$ pep-tRNAs (Supplementary Fig. 11d). We next examined the relationship between the frequency and GC content of codons at the drop-off sites (Supplementary Fig. 11e, f). All sense codons were classified into four groups based on the number of G or C bases in the three letters of codons. In each codon group, the frequencies of codons at the drop-off sites were graphed in box plots and compared between codon groups. Although no tendency was observed in the four codon groups for the cognate pep-tRNAs (Supplementary Fig. 11e), the drop-off sites for the $C_0X$ pep-tRNAs were significantly enriched in AU-rich codons (Supplementary Fig. 11f). Given that aa-tRNAs for Lys and Phe are rapidly depleted upon PTH inactivation (Fig. 1g), AAR and UUY codons at the A-site may be hungry, promoting miscoding events that result in the generation of $C_0X$ pep-tRNAs.

Given that the miscoding event takes place at near-cognate pairs of codon–anticodon interactions, we asked whether near-cognate patterns were actually observed at the drop-off sites for $C_0X$ pep-tRNAs. We classified codon–anticodon mismatches at the drop-off sites for 286 $C_0X$ pep-tRNAs, and found that 74 sites were pairings with single mismatches and the remaining 212 sites bore multiple mismatches (Supplementary Fig. 11g). Only 26% of the drop-off sites were near-cognate pairings that yielded $C_0X$ pep-tRNAs. Thus, 74% of the $C_0X$ pep-tRNAs were generated by random pairs of codon–anticodon interactions.

Next, we compared the frequency of codons at drop-off sites for cognate vs $C_0X$ pep-tRNAs, but observed little correlation ($R^2 = 0.033$) between them (Fig. 5h). On the other hand, we noticed a clear correlation ($R^2 = 0.643$) between the codons next to the drop-off sites of the cognate pep-tRNAs and those of $C_0X$ pep-tRNAs (Fig. 5h).

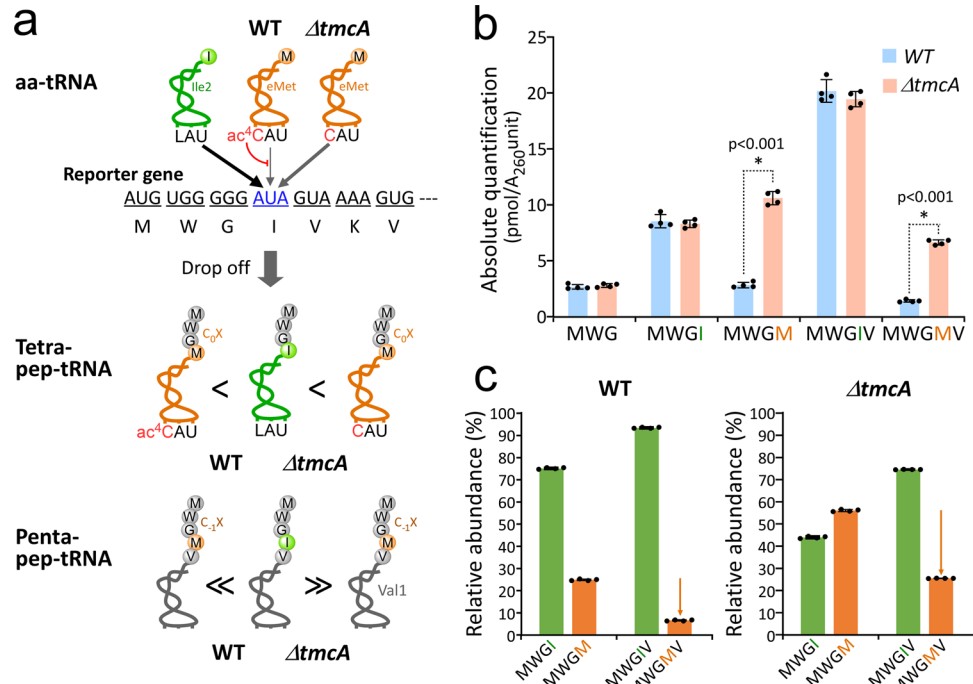

**Fig. 6 | Miscoding induces pep-tRNA drop-off and quality control of translation. a** A reporter system for examining the effect of miscoding on pep-tRNA drop-off. The reporter gene has a unique N-terminal sequence that does not exist in any *E. coli* genes. AUA codon is set as a test codon at the fourth position. If the AUA codon is decoded normally by tRNA^Ile2 with L34, the cognate pep-tRNA (MWGI-tRNA^Ile2) is produced. If the AUA codon is misread by tRNA^eMet, $C_0X$ tetra-pep-tRNA (MWGM-tRNA^eMet) is generated. In the WT strain, ac⁴C34 of tRNA^eMet prevents misreading of the AUA codon, whereas in the *ΔtmcA* strain, tRNA^eMet with unmodified C34 frequently misreads the fourth codon (AUA) in the reporter gene. If a miscoding event induces the pep-tRNA drop-off, $C_0X$ tetra-pep-tRNA should accumulate in the *ΔtmcA* strain. On the other hand, if the $C_0X$ tetra-pep-tRNA is excluded from the translation machinery by the pep-tRNA drop-off, $C_{-1}X$ penta-pep-tRNA (MWGMV-tRNA^Val1) should accumulate at lower levels in both the *ΔtmcA* and WT strains. **b** Quantification of peptides of tri-, tetra-, and pentapeptides derived from pep-tRNAs translating the reporter gene expressed in the WT (pale blue) and *ΔtmcA* (pale red) strains. Each peptide was absolutely quantified by an external standard method using synthetic authentic peptides (Supplementary Fig. 12a). Data are presented as means±s.d. of four independent experiments. *p < 0.001 (two-tailed *t*-test). Exact *p*-values for MWGM and MWGMV are $3.1 \times 10^{-7}$ and $1.2 \times 10^{-8}$, respectively. **c** Quality control of translation by pep-tRNA drop-off. Relative abundances of cognate (green) and non-cognate (orange) peptides derived from tetra-pep-tRNAs (MWGI and MWGM) and penta-pep-tRNAs (MWGIV and MWGMV) were calculated from absolute quantification of the peptides (Fig. 6b) in WT (left) and *ΔtmcA* (right) strains. Source data are provided as a Source Data file.

This observation implies a mechanism for drop-off (Fig. 5i): once a hungry codon enters the A-site, a cognate pep-tRNA stays for a considerable length of time at the P-site, thereby promoting pep-tRNA drop-off. At the same time, if the hungry codon is miscoded by non-cognate aa-tRNA, followed by peptide bond formation and translocation, the non-cognate pep-tRNA is unstably bound to the P-site, yielding the generation of $C_0X$ pep-tRNA mediated by ribosome dissociation factors.

## Miscoding induces the pep-tRNA drop-off

Given that $C_0X$ type non-cognate peptides were abundant in the *pth*^ts strain, we asked whether non-cognate pep-tRNAs generated by miscoding were actively dissociated from the ribosome during translation elongation. To explore this possibility, we analyzed miscoding of the AUA codon. The AUA codon specifies Ile, but misreading of AUA codon as Met is induced by hypomodification of tRNA^eMet[78]. To measure the miscoding activity of the AUA codon, we made a reporter construct with an artificial ORF starting from an unnatural sequence, MWGIVK, which is not present in any ORFs in *E. coli*, in order to detect pep-tRNAs derived from this artificial template (Fig. 6a). The Ile at the fourth position of the reporter was encoded by an AUA codon. We expressed the reporter gene in the *pth*^ts strain. Upon inactivation of PTH by incubating the cells at 43 °C for 30 min, pep-tRNAs were prepared from the cells, followed by deacylation, and the nascent peptides were subjected to LC/MS to detect the peptides derived from the reporter mRNA. We clearly detected cognate and non-cognate peptides derived from the reporter, from

trimers to pentamers. Each peptide was absolutely quantified based on an external standard method using synthetic authentic peptides (Fig. 6b and Supplementary Fig. 12a). In the *pth*^ts strain (denoted as WT), a series of cognate peptides, MWG, MWGI, and MWGIV, were detected. In addition, we clearly detected the $C_0X$ tetrapeptide MWGM and the $C_{-1}X$ pentapeptide MWGMV, both of which were produced by miscoding the AUA codon at the fourth position as Met. The relative abundance of tetra- and penta-peptides was calculated by absolute quantification (Fig. 6c and Supplementary Fig. 12a), revealing that the MWGM non-cognate peptide ($C_0X$ type) constituted about 25% of the tetrapeptides, whereas the MWGMV non-cognate peptide ($C_{-1}X$ type) constituted only about 6.5% of pentapeptides. We observed a clear reduction of non-cognate peptides from $C_0X$ to $C_{-1}X$ (Fig. 6a).

Next, we induced miscoding of the AUA codon by knocking out *tmcA*, which encodes a tRNA acetyltransferase responsible for $N^4$-acetylcytidine (ac⁴C) modification in the anticodon of tRNA^eMet[79]; ac⁴C plays a critical role in preventing miscoding of the AUA codon[78,80]. In the *ΔtmcA* strain (*pth*^ts background), we detected and quantified cognate and non-cognate peptides derived from the reporter gene (Fig. 6b and Supplementary Fig. 12a). As expected, the level of the non-cognate peptide MWGM ($C_0X$ type) increased markedly, constituting up to 56% of the tetrapeptides (Fig. 6c), whereas the remaining 44% of tetrapeptides were cognate, indicating that loss of the ac⁴C modification significantly induced miscoding of the AUA codon in the reporter mRNA and dissociation of the non-cognate pep-tRNA from ribosome at the codon.

Regarding pentapeptides, we observed a clear reduction in the level of the non-cognate peptide MWGMV ($C_{-1}X$ type), which constituted 25% of the pentapeptides (Fig. 6c).

The abundance of the $C_{-1}X$ MWGMV peptide relative to that of the cognate peptide was markedly lower than that of the $C_0X$ MWGM peptide in both WT and Δ*tmcA* strains (*pth*ts background) (Fig. 6a, b, c). Furthermore, we calculated the drop-off rates of MWGI and MWGM pep-tRNA by dividing the amount of each tetrapeptide by the sum of amounts of tetra and pentapeptides (see Methods). The results show that the drop-off rate of MWGM pep-tRNA is approximately 2-fold larger than that of MWGI pep-tRNA in both WT and Δ*tmcA* strains (Supplementary Fig. 12b). These observations strongly suggest that the $C_0X$ MWGM peptide is actively eliminated by the drop-off mechanism and does not efficiently participate in the next round of translation.

## Discussion

Early studies hypothesized that the pep-tRNA drop-off might be involved in accuracy of protein synthesis[50,81]. However, technical difficulties in characterizing peptides of drop-off pep-tRNAs have hindered the understanding of the physiological function of this event. We here established a highly sensitive method for profiling pep-tRNAs using mass spectrometry and successfully analyzed nascent peptides of pep-tRNAs accumulated in *E. coli pth*ts strains. This method can be applied to various *E. coli* strains, and other organisms including yeast and mammalian cells to generalize our observation in the future studies.

We showed that the N-terminal formyl group is missing from the peptide chain of the pep-tRNA. PDF binds to the ribosome tunnel exit and waits to deformylate the nascent peptide chains during protein synthesis[82]. Our observation indicates that PDF also acts on pep-tRNAs dissociated from the ribosome. After deformylation, the first Met is removed from the peptides longer than trimers by MAP, especially when the second residues are small amino-acids[75,76]. These findings contribute to our deep understanding of how the peptides of dissociated pep-tRNAs are metabolized.

We clearly detected the accumulation of pep-tRNAs and the depletion of aa-tRNAs over time for each species of peptide and amino-acid, showing that the drop-off efficiency varies depending on the species of pep-tRNA (Fig. 1g, h). The large variation in accumulation rate of pep-tRNAs and depletion rate of aa-tRNAs means that each tRNA has a different recycling rate during protein synthesis. This is an important finding in light of the usage of each tRNA and its aminoacylation efficiency in the cell. It is also interesting to consider which factors in pep-tRNAs determine drop-off efficiency. One factor might be the strength of the interaction between the peptide moiety and the inner surface of the ribosome tunnel, as discussed in the next paragraph. Another factor might be the difference in the codon–anticodon pairing of each pep-tRNA.

According to the comprehensive study of protein production examining different initial ORF sequences[53], translation efficiency is strongly dependent on 3rd to 5th codon positions. We here investigated initial sequences of *E. coli* ORF in association with the drop-off efficiency and the protein expression. We first chose 678 ORFs that are highly expressed in *E. coli* with top 20% TPM based on our RNA-seq data (Supplementary Data 2). In 678 ORFs, pep-tRNAs were detected from 161 ORFs (Groups 1 and 2), but not detected from 517 ORFs (Group 3) (Supplementary Fig. 13a). The 161 ORFs are further divided into two groups, 73 ORFs (Group 1) in which pep-tRNAs are dissociated from 5th and 6th codon positions (Supplementary Fig. 13a), and 88 ORFs (Group 2) in which pep-tRNAs are dissociated from 7th codon or later positions (Supplementary Fig. 13a). The Group 1 pep-tRNAs are efficiently dissociated at earlier stage (5 and 6 positions) of elongation, whereas the pep-tRNAs of Groups 2 and 3 are less efficiently dissociated at the same positions. For each group, we obtained GFP scores[53] corresponding to 3rd-5th codons nucleotide sequences of

ORFs, and then compared the distribution of their GFP scores between 3 groups. As shown in Supplementary Fig. 13b, the GFP scores of Group 1 show significantly lower than those of Groups 2 and 3. No significant difference between Groups 2 and 3 is found. This observation strongly suggests that pep-tRNAs are efficiently dissociated from *E. coli* ORFs having initial sequences with low translational efficiency. In addition, it is also suggested that the presence of bulky amino acids such as F, Y and W in the nascent chain stabilize the association of ribosome during translation in the early stage[83]. Comparing the frequency of each amino acid in the identified pep-tRNA in this study and in the N-terminal region (2nd-10th codon region) of *E. coli* ORFs, F, Y and W appear less frequently in the pep-tRNAs dissociated from ribosome than the other amino acids (Supplementary Data 3), indicating that translating ribosome in the early elongation stage is stabilized by bulky amino acids in pep-tRNAs.

Genetic and biochemical studies have suggested that RF3 and RRF are actively involved in pep-tRNA drop-off[38,43–46,84]. In general, RF3 interacts with the post-termination complex of ribosome after peptide release at the stop codon mediated by RF1 or RF2, and accelerates the dissociation of the release factor from the ribosome[85–87]. RF3 binding induces the structural rearrangement of the release factor-bound ribosome from the classical state to the rotated state, moving P-site tRNA to the P/E state[88–90]. Biophysical studies showed that RF3 in the GTP form is able to interact with the ribosome regardless of the presence or absence of the release factor, or the ribosome with pep-tRNA or deacylated-tRNA at the P-site[85,91]. In addition, it is known that RF3 could bind to the ribosome with pep-tRNA bearing short peptide at P-site, and alter its conformation from the classical state to the rotated state[91,92]. These facts imply that RF3 potentially binds to an elongating ribosome with pep-tRNA at the P-site, and facilitates its conformational change to the rotated state with pep-tRNA in the P/E hybrid state, where RRF can interact. In the early stage of translation elongation, the nascent peptide of pep-tRNA is short, and the peptide weakly interacts with the inner surface of the tunnel, allowing RF3 to translocate pep-tRNAs to the P/E hybrid state. RF3 then hydrolyzes GTP and changes its conformation to the GDP-bound form, leaving the ribosome. Subsequently, RRF and EF-G disassemble the ribosome, releasing pep-tRNA via the same mechanism as disassembly of the post-termination complex[93]. According to a crystal structure of 70 S ribosome in complex with two tRNAs, RRF, and EF-G[94], the tip of domain I of RRF interacts with the P-loop of 23 S rRNA and dislocates the CCA terminus of P-site tRNA to the p/R-site, a previously unidentified site between the P and E-sites. In this process, the nascent peptide attached to the CCA terminus should detach from the tunnel and move to the p/R-site. Because of the short nascent peptide, the pep-tRNA might readily adopt the p/R state, facilitating ribosome disassembly. Curiously, in the p/R state, the codon–anticodon pairing is destabilized. The distance between the third base of the codon and the first base of the anticodon widens to 5 Å[94], indicating that weak codon–anticodon interaction strongly influences pep-tRNA drop-off. Due to taking miscoding state at the P-site, it is likely that $C_0X$ type non-cognate pep-tRNA readily takes the p/R state. According to our analysis, AU-rich codons are enriched at drop-off sites for the $C_0X$ pep-tRNAs (Supplementary Fig. 11f), implying that the AU-rich codons are prone to be miscoded and allow the $C_0X$ pep-tRNAs to take P/E and p/R state. Then, the $C_0X$ pep-tRNAs are dissociated from the ribosome by the concerted action of RF3 and RRF/EF-G. In addition, if pep-tRNA takes P/E or p/R states, there is no room in the 50 S subunit to accommodate the peptide attached to CCA terminus[94,95]. The nascent peptide might weaken the interaction between CCA-terminus of pep-tRNA and E-site of 50 S subunit, facilitating pep-tRNA drop-off.

Translational fidelity and pep-tRNA drop-off were implicated by genetic studies[44,84,96]. More recently, the mutants of uS12, a ribosomal protein involved in the decoding accuracy, have been reported to have genetic interactions with RRF and PTH[97,98]. An earlier study

showed that streptomycin, a miscoding-inducing antibiotic, increased pep-tRNA drop-off and the temperature sensitivity of $pth^{ts}$ strain and expected the preferential dissociation of erroneous pep-tRNAs from ribosome[99]. Pep-tRNA profiling and detailed analyses of isolated pep-tRNAs allows us to observe these erroneous peptides harboring single amino-acid substitutions in addition to cognate peptides (Figs. 4 and 5). For isolated pep-tRNAs, we also identified the gene and the drop-off site from which each pep-tRNA was derived and found that 94% of the non-cognate peptides were of the $C_0X$ type (only 6% are of $C_{-1}X$, $C_{-2}X$, and $C_{-3}X$ types), strongly suggesting that miscoding events induce pep-tRNA drop-off, as the previous study expected. This was also supported by the results from the experiments using a reporter construct (Fig. 6). Moreover, the analysis of the patterns of the codon-anticodon pairing on non-cognate pep-tRNAs revealed that the near-cognate patterns with a single mismatch are minor cases, whereas 74% of the $C_0X$ pep-tRNAs were generated by non-cognate patterns of miscoding, with little complementarity in codon–anticodon pairing (Supplementary Fig. 11g). From these observations, we conclude that aminoacyl-tRNAs can misread non-cognate codons and accept peptides through peptidyl transfer reactions in the initial stage of translation elongation, even though initial selection and proofreading contribute to the quality control of translation, and that thus-generated non-cognate pep-tRNAs are actively excluded from translation by pep-tRNA drop-off. The drop-off sites for cognate pep-tRNAs were widely distributed in the codon positions up to 16th from the start codon, whereas the drop-off sites for non-cognate pep-tRNAs were narrowly distributed in the early codon positions up to 10th (Fig. 5f). This tendency is consistent with the finding that non-cognate peptides are lighter than cognate peptides (Fig. 4e, f). These observations imply that non-cognate pep-tRNAs likely to be produced for some reason in the earlier stage of translation, but are frequently eliminated from the translation system by pep-tRNA drop-off, thereby maintaining the overall fidelity of protein synthesis.

Why does miscoding frequently happen in the early elongation? In general, overall codon usage of ORFs correlates well with abundance of tRNA species. Namely, highly used codons are read by high abundant tRNAs, whereas rare codons are read by less abundant tRNAs[100–102]. Because the competition between cognate and non-cognate tRNAs for A-site codon affects translation rate and fidelity, optimal codons are decoded more quickly and accurately than non-optimal ones[103–105]. Therefore, the optimal codon usage contributes to efficient translation[100,106–108], proper protein folding[109–111] and regulation of mRNA stability[112–115]. However, in many bacteria, the codon usage in the beginning of ORFs markedly deviates from the overall codon usage for two reasons[116–123]. One is that AU-rich codons are enriched in the beginning of ORFs to avoid RNA folding around the start codon for efficient translation initiation[120], the other is that non-optimal codons in the initiation region are read by less abundant tRNAs to slow down the elongation rate, thereby reducing ribosome traffic jams in the downstream of genes[122]. As shown in Supplementary Fig. 14a, we compared codon usage distribution in the beginning of E. coli ORFs. The codon usage in the 2–8th positions is quite different from total codon usage ($R^2 = 0.367$). Curiously, the codon usage in the 9–15th positions shows a strong similarity to the total codon usage ($R^2 = 0.864$). The deviation becomes smaller as the codon positions depart from the start codon, and almost disappears around the 50th position (58–64th codons, $R^2 = 0.985$). Then, we calculated ratio of each codon usage in the early positions versus total positions of ORFs, and compared their distributions by box plots with standard deviations (Supplementary Fig. 14b). We clearly show that the distribution in the 2–8th positions ($\sigma = 0.745$) is quite different from that in the 9–15th positions ($\sigma = 0.297$) as well as those in the later positions. A large deviation of the codon usage bias in the beginning of ORF might be one reason to explain high error rate in the early elongation stage. Supporting this, we detected a large amount of erroneous pep-tRNAs

with short peptides accumulated in E. coli cells. Therefore, the pep-tRNA drop-off plays a role in quality control of protein synthesis in the early elongation. In addition, the 5′-region-specific codon usage bias is more pronounced in bacteria with higher GC content[120], implying that pep-tRNA drop-off occurs more frequently and the role of PTH is more important in such GC-rich bacteria. Further studies are necessary to generalize the quality control of protein synthesis in the early elongation mediated by pep-tRNA drop-off in other organisms.

In this study, we revealed a novel mechanism of quality control of protein synthesis mediated by pep-tRNA dissociation from the ribosome in the early stage of translation elongation (Fig. 7). Since the codon usage in the beginning of ORFs deviates from the overall codon usage, non-cognate aa-tRNA frequently passes through two quality control mechanisms (initial selection and proofreading) due to low codon optimality and accommodates to the A-site, followed by peptide bond formation and translocation, producing a non-cognate pep-tRNA at the P-site. The miscoded ribosome is efficiently disassembled by the concerted action of RF3 and RRF/EF-G, releasing the non-cognate pep-tRNA which is then hydrolyzed and recycled by PTH. The disassembled ribosome also participates in the new round of translation. In the middle-late region of mRNAs, the codon usage bias becomes smaller as the codon positions depart from the start codon, and the protein synthesis smoothly proceeds due to high codon optimality. Because the nascent peptides of pep-tRNAs are long enough to form strong interactions with the ribosome tunnel, pep-tRNAs do not dissociate from the ribosome. If translation accuracy is compromised under some stress conditions in which codon optimality is disrupted, such as amino-acid starvation, RF2 senses the miscoding state and triggers release of the peptide of the pep-tRNA on the ribosome[25,30–32], followed by ribosome disassembly mediated by RF3, RRF, and EF-G via the same mechanism as ribosome recycling.

## Methods

### Construction of strains and vectors

The temperature-sensitive strain $pth^{ts}$ was constructed as described below. The $pth$ gene with a kanamycin-resistance cassette ($Km^R$) was PCR-amplified from genomic DNA of strain JW1196 ($ychH$::FRT-$Km^R$-FRT), digested with EcoRI and HindIII, and cloned into pQE60 (QIAGEN) to construct pAN100. The temperature-sensitive mutation (302 G > A; G101D)[39] was introduced into the $pth$ gene in pAN100 via QuikChange site-directed mutagenesis (Stratagene), yielding pAN101. Finally, $pth^{ts}$ (G101D) with $Km^R$ was PCR-amplified from pAN101 and electroporated into strain BW25113 harboring pKD46, followed by selection with kanamycin (50 μg/ml) to yield the $pth^{ts}$ strain (TS101). All primers used are listed in Supplementary Data 4. A series of knockout strains in the $pth^{ts}$ background was constructed as follows. The gene-deletion strains $\Delta tolC$, $\Delta lpp$, $\Delta grcA$, $\Delta tabA$, $\Delta tatA$, $\Delta tig$, and $\Delta tmcA$, obtained from the E. coli KEIO collection resource[124], were transformed with pCP20 to remove the $Km^R$ cassette from the genome in each strain. The temperature sensitivity of $pth$ was transferred to each knockout strain using P1 phage containing the $pth^{ts}$ (G101D) gene derived from the $pth^{ts}$ strain (TS101), yielding $\Delta tolC/pth^{ts}$, $\Delta lpp/pth^{ts}$, $\Delta grcA/pth^{ts}$, $\Delta tabA/pth^{ts}$, $\Delta tatA/pth^{ts}$, $\Delta tig/pth^{ts}$, and $\Delta tmcA/pth^{ts}$ strains.

The temperature-sensitive strain for RRF (encoded by the $frr$ gene) was constructed as follows. The $frr$ gene and the $Km^R$ cassette were PCR-amplified from the genomic DNA of strain JW1196 ($ychH$::FRT-$Km^R$-FRT) and digested with NcoI/EcoRI and EcoRI/HindIII, respectively. They were inserted into vector pBAD/Myc-His A (Invitrogen) at the appropriate points in the multiple cloning site, yielding pAN102. The temperature-sensitive mutation (350 T > A; V117D) gene was introduced into $frr$ in pAN102 via QuikChange site-directed mutagenesis, yielding pAN103. The $frr^{ts}$ gene with the $Km^R$ cassette was PCR-amplified from pAN103 and electroporated into strain BW25113

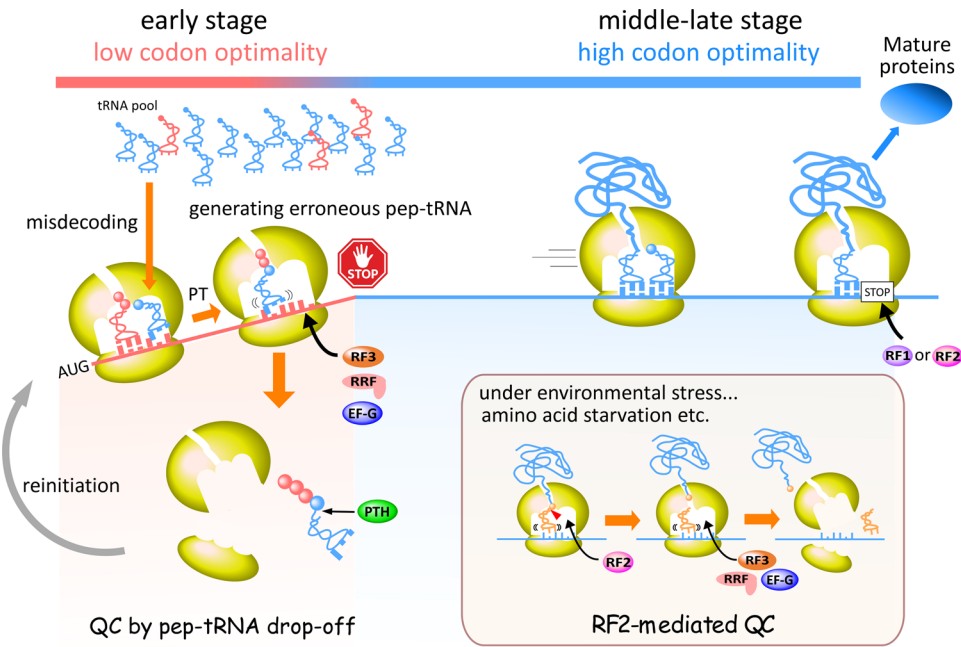

**Fig. 7 | Quality control mechanism in the early translation elongation by pep-tRNA drop-off.** In many bacteria, low codon optimality at the beginning of ORFs reduces the fidelity of translation and leads to produce a large amount of erroneous pep-tRNAs. However, the miscoded ribosome is efficiently disassembled by the concerted action of RF3 and RRF/EF-G (pep-tRNA drop-off), maintaining the accuracy of overall translation. In the middle and late region of mRNAs, translation smoothly proceeds due to high codon optimality to make a functional protein. If translation accuracy is compromised under some stress conditions in which codon optimality is disrupted, such as amino-acid starvation, RF2 senses the miscoding state and triggers release of the peptide of the pep-tRNA on the ribosome, followed by ribosome disassembly mediated by RF3, RRF, and EF-G via the same mechanism as ribosome recycling.

harboring pKD46, followed by selection with kanamycin (50 µg/ml) to obtain the $frr^{ts}$ strain (TS109). All primers used were listed in Supplementary Data 4.

To construct the reporter gene used in Fig. 6, a GFP gene starting with a sequence encoding MWGIVK was PCR-amplified and cloned into pBAD/Myc-His A (Invitrogen) by the SLICE method[125,126] to yield pAN104. All primers are listed in Supplementary Data 4.

### Preparation of aa- and pep-tRNAs from *E. coli* cells

*E. coli* strains in the $pth^{ts}$ background were cultured at 30 °C in LB medium. In mid-log phase ($OD_{600} \approx 0.5$), the temperature was shifted from 30 °C to 43 °C; the culture was incubated for 30 min and then harvested. The cells were suspended in a buffer containing 120 mM NaOAc (pH 5.2) and 10 mM DTT, followed by addition of a 3.5-fold volume of TriPure Isolation Reagent (Roche) and vigorous mixing for 20 min at 4 °C. After addition of 1/5 volume of chloroform, the lysate was centrifuged at 8,000 *g* at 4 °C for 15 min to separate the phases, and the aqueous phase was collected. Total RNA was recovered by 2-propanol precipitation and dissolved in 300 mM NaOAc (pH 6.0). 2-iodeacetamide (f.c. 10 mM) was added to the solution and incubated at 0 °C for 1 h in the dark to alkylate the thiol groups of cysteine resides in aa-/pep-tRNAs, and then the sample was subjected to ethanol precipitation. To stabilize the acyl bonds, α-amino groups of aa-/pep-tRNAs were acetylated with acetic anhydride at 0 °C as described[127], followed by 2-propanol precipitation.

For further purification, the total RNA was dissolved in QA buffer [50 mM NaOAc (pH 5.2) and 100 mM NaCl], mixed with Q-Sepharose Fast Flow resin (GE Healthcare), and incubated at 4 °C for 20 min. The resin was washed twice with QA buffer, and total RNA was eluted with QB buffer [50 mM NaOAc (pH 5.2) and 1 M NaCl] and recovered by 2-propanol precipitation.

For analysis of dipep-tRNAs from $\Delta tolC/pth^{ts}$ cells treated with tylosin, cells in log phase were treated with various concentrations of tylosin for 5 min at 30 °C and incubated at 37 °C for 30 min. The dipep-tRNAs were extracted and analyzed as described above.

### Isolation of pep-tRNAs

Individual pep-tRNA species analyzed in this study were isolated from total RNA of the *E. coli* strains by RCC[77]. DNA probes with a 5′-ethylcarbamate amino linker (Sigma-Aldrich) were covalently immobilized on NHS-activated Sepharose 4 Fast Flow (GE Healthcare) and packed into tip columns for the RCC instrument. All probe sequences are listed in Supplementary Data 4. To minimize the dissociation of peptide from the tRNA and prevent the oxidation of methionine, we optimized the binding buffer consisting of 1.2 M NaCl, 30 mM MES-NaOH (pH 6.0), 15 mM EDTA, and 2.5 mM DTT. The number of pipetting steps was set to 17. In this study, we isolated 42 out of a total of 46 elongator tRNA species in *E. coli*. Only a few bases differ among pairs of isodecoders in three tRNAs for Leu1, Ile2, and Tyr, so these were not isolated separately. In addition, tRNA^Sec was not isolated.

### Nucleoside and RNA mass-spectrometric analyses

Liquid chromatography–mass spectrometry (LC/MS) analyses of Ac-(aa/pep)-Ado and Ac-(aa/pep)-tRNA fragments were performed as described[78,128]. The acetylated tRNA fraction was digested into nucleosides by incubation for 1 h at 37 °C with nuclease P1 (Wako Pure Chemical Industries) and alkaline phosphatase (E. coli C75, Takara). Nucleosides containing Ac-aa-Ado and Ac-pep-Ado were analyzed on a reverse-phase chromatograph (RPC, Sunshell C18, ChromaNik Technologies Inc)-MS equipped with a quadrupole-orbitrap hybrid mass spectrometer (Q Exactive, Thermo Fisher Scientific). Xcalibur4.5 SP1(Thermo Fisher Science) was used for the analysis of LC/MS data.

In the analyses of Ac-aa-tRNAs, MS intensities of Ac-aa-Ado species are normalized against that of f-Met-Ado. For each Ac-aa-Ado, the relative intensity of MS peak area at each time point is normalized against the intensity at 0 min. For the analyses of Ac-dipep-Ado, the

relative intensity of MS peak area at each time point is normalized against that of f-Met-Ado. To evaluate overall change in aa-tRNAs and dipep-tRNAs in *pth*ts cells after the temperature shift to 43 °C, relative intensity of MS peak area of each aa-tRNA and dipep-tRNA is normalized against that of f-Met-tRNA. The normalized intensities of all aa-tRNAs and all dipep-tRNAs are summed at each time point. Percentiles of summed intensities for aa-tRNAs and dipep-tRNAs are plotted over time as relative abundance.

To analyze 3′-terminal RNA fragments, isolated tRNAs were digested with RNase $T_1$ (Epicentre) in 20 mM $NH_4OAc$ (pH 5.3) and analyzed on the capillary LC/nanoESI MS system (LTQ Orbitrap XL, Thermo Fisher Scientific)[128].

## Ribosome profiling

WT BW25113 and *frr*ts strains cultured in log phase were incubated at 43 °C for 30 min, harvested by centrifugation, and frozen immediately. Chloramphenicol (f.c. 100 µg/ml) was added to the cultures just before harvesting the cells. Whole-cell lysates were prepared using SK-100 (Tokken) in a buffer containing 20 mM Tris-HCl (pH 7.5), 100 mM $NH_4Cl$, 10 mM $Mg(OAc)_2$, 5 mM $CaCl_2$, 1 mM DTT, and 100 µg/ml chloramphenicol, and then treated with 7.5 U of TURBO DNase (Invitrogen). Lysate containing 40 µg RNA was digested with 17 U of MNase (Roche). The subsequent library preparation was conducted as described previously[129,130]. Ribo-Zero rRNA Removal Kit for Bacteria (Illumina) was used to deplete rRNA contamination. The libraries were sequenced on a HiSeq 3000 (Illumina) (paired ends for 100 nt). The ribosome profiling data is deposited in DDBJ (accession number: DRA012800).

The reads 1 were corrected by reads 2, using Fastp[131]. Reads were then demultiplexed according to barcode sequences, listed in Supplementary Data 4. Linker sequence-trimmed reads were aligned to rRNAs and tRNAs by bowtie2[132] and the unaligned reads were mapped to *E. coli* genome (NC_000913.2) by bowtie2. The A-site offset of reads was estimated as previously described[133,134]. mRNAs yielding 1 or more mean reads per codon in all replicates were analyzed. Reads on the given codon were normalized to the mean reads per codon for the mRNA.

## Reference lists for pep-tRNA profiling

For pep-tRNA profiling, MS-DIAL software[135] was used for peak detection of the MS data for pep-tRNA peptides and deconvolution of MS peaks to obtain the mass list. To determine whether a detected molecular mass in the list corresponded to a nascent peptide of *E. coli* ORF, we made two kinds of reference list: one for the initial sequences (3–14 aa.) of *E. coli* ORFs, and the other for the initial sequences (3–14 aa.) with all combinations of single substitutions, based on the peptide sequences from the *E. coli* K-12 MG1655 U00096.3 Database table from EcoGene (http://ecogene.org/). Given that MAP removes the first Met of pep-tRNA, we included a set of N-terminal peptides missing the first Met in the reference list of *E. coli* ORFs bearing Gly, Ala, Ser, Thr, Cys, Pro, Val, Asn, Asp, Leu or Ile at the second position based on the substrate recognition rule of MAP[75,76]. We calculated the molecular mass of each peptide from the accurate masses of amino-acids: Ala: 71.03697, Arg: 156.10078, Asn: 114.04274, Asp: 115.02677, Cys* (carbamidomethyl cysteine): 160.03041, Gln: 128.05834, Glu: 129.04237, Gly: 57.02137, His: 137.05871, Ile: 113.08377, Leu: 113.08377, Lys: 128.09464, Met: 131.04024, Phe: 147.06817, Pro: 97.05257, Ser: 87.03187, Thr: 101.04747, Trp: 186.07904, Tyr: 163.06307, and Val: 99.06817.

## MS analyses for nascent peptides of pep-tRNAs

MS analyses of the peptides of pep-tRNAs were performed as follows. Total tRNA or isolated pep-tRNAs were hydrolyzed into peptides and tRNA by incubation in 0.3% ammonia water at 50 °C for 30 min, and then the ammonia was completely evaporated. To verify the peptides derived from pep-tRNAs with isotope labeling, 0.3% ammonia with 50%

[18O] water (Otsuka Pharmaceutical) was used to hydrolyze pep-tRNA. The hydrolysate dissolved in a buffer containing 10 mM ammonium bicarbonate (pH 7.5) and 250 mM NaCl was mixed with Q-Sepharose Fast Flow resin to capture tRNAs. Under this condition, all deacyl tRNAs are removed by the Q-Sepharose resin, and most of the peptides are not removed, whereas some of anionic peptides may be trapped by the resin. The peptides in the flowthrough fraction were purified using C18 stage tip (Thermo Fisher Scientific) according to the manual instructions, dried and dissolved in 0.1% (v/v) formic acid. However, highly hydrophobic peptides may not be recovered in this process. Xcalibur4.5 SP1(Thermo Fisher Science) was used for the analysis of LC/MS data.

To profile pep-tRNAs and analyze nascent peptides from the isolated pep-tRNAs, peptides prepared as described above were subjected to the capillary LC/nanoESI MS system (LTQ Orbitrap XL or Orbitrap Eclipse, Thermo Fisher Scientific). Peptides were separated at a flow rate of 300 nL/min by capillary LC using a linear gradient from 2–80% solvent B (v/v) in a solvent system consisting of 0.1% (v/v) formic acid (solvent A) and 70% (v/v) acetonitrile (solvent B). For pep-tRNA profiling, molecular mass detected in the mass list of the sample from *pth*ts cells cultured at 30 °C (at 43 °C for 0 min) were removed from the mass list of the sample at 43 °C for 30 min as chemical noise. For peptide sequencing of the isolated pep-tRNAs, the ionized peptide with proton adduct was decomposed by CID in the instrument. Peptide sequences were determined using the MASCOT (Matrix Science) search engine with custom-made databases (Nascent peptide databases), as described below. It was concerned that alkylated Cys (here shown as Cys*), Thr and Ser in the peptides were β-eliminated during alkaline treatment for deacylation of pep-tRNAs. To test this possibility, we analyzed tryptic digests of alkylated BSA treated with 0.3% ammonia water at 50 °C for 30 min (the same condition as deacylation). We found no evidence of β-elimination in any tryptic peptides.

For analyses of pep-tRNAs derived from the reporter gene, nascent peptides of pep-tRNAs extracted from the cells overexpressing the reporter gene were prepared in the same way as described above, and subjected to RPC-MS (Q Exactive, Thermo Fisher Scientific). Xcalibur4.5 SP1(Thermo Fisher Science) was used for the analysis of LC/MS data. To perform absolute quantification of each peptide, *N*-acetylated oligo peptides for MWG, MWGI, MWGM, MWGIV, and MWGMV were chemically synthesized (Eurofins Genomics) and subjected to RPC-MS at various concentrations. A calibration curve for the amount of each peptide vs. its MS peak area was prepared by a double-logarithmic plot for each peptide (Supplementary Fig. 12a). Each peptide in 1 OD of tRNA fraction was quantified from the MS peak area using the calibration curve.

Since the drop-off efficiency strongly depends on the codon-anticodon pairing at a given codon position, the drop-off and read-through efficiencies at the next codon and subsequent positions should be the same regardless of the miscoding event at the given position. Thus, to calculate the drop-off rates for the 4-mer cognate (MWGI-tRNA) and noncognate pep-tRNAs (MWGM-tRNA), we use the 5-mer pep-tRNA (MWGIV-tRNA and MWGMV-tRNA) as products of readthrough at the fourth AUA codon. The formula, 4-mer pep-tRNA/ (4-mer pep-tRNA + 5-mer pep-tRNA), was applied to calculate the drop-off rate in this study.

## Nascent peptide sequence by MS

To determine peptide sequences of the isolated pep-tRNAs, we made two reference lists for initial sequences of *E. coli* ORFs. Nascent Peptide Databases 1 and 2 are lists of nascent peptides covering 4–10 aa. and 10–16 aa., respectively, from the initiation sites of *E. coli* ORFs. For $C_0X$ non-cognate peptides, the C-terminus of each peptide was substituted with the "X" residue, which represents any of the amino-acids. To search for $C_{-1}X$, $C_{-2}X$, and $C_{-3}X$ peptides, the corresponding residues in the peptide listed in the reference are substituted with "X".

The MS data for the peptides of the isolated pep-tRNAs were submitted to MASCOT MS/MS ions search (Matrix Science, ver2.6.0) against Nascent Peptide Databases 1 and 2. Search parameters used in this analysis were set as follows: no enzyme, fixed modifications to Lys acetylation, N-terminal acetylation and Cys carbamidomethylation with a variable modification set for Met oxidation, and fixing fragment monoisotopic mass with peptide mass tolerances of ±5 ppm and fragment mass tolerances of ±0.8 Da.

The MS data are also searched against a decoy database of the original databases. The significance threshold was set at $p < 0.05$ with a 5% false discovery rate (FDR). For each isolated pep-tRNA, a mass difference of the nascent peptide relative to its theoretical value ($\Delta$ppm) <5, peptide score > 30, and pepExpect <0.05 were extracted from the MASCOT search (MASCOT Daemon (2.6.0)). Assignment of each peptide to a specific *E. coli* ORF was carried out as follows. If there was only one candidate gene, the peptide is assigned. If there were multiple candidate genes, the list of candidates was narrowed down based on the following selection criteria: higher peptide score, preference for cognate pep-tRNA over non-cognate pep-tRNA, and higher TPM from the RNA-seq analysis. If there were still multiple candidate genes after this process, the peptide was not assigned to any genes and was not included in the list.

The MS proteomics data have been deposited to the ProteomeXchange Consortium via the PRIDE[136] partner repository with the dataset identifiers PXD030807 and 10.6019/PXD030807 (Nascent Peptide Database2), PXD030808 and 10.6019/PXD030808 (Nascent Peptide Database1) for the search of cognate and $C_0X$ non-cognate peptides, and with the dataset identifiers PXD030805 and 10.6019/PXD030805 (Nascent Peptide Database2 CX), PXD030806 and 10.6019/PXD030806 (Nascent Peptide Database1 CX) for the search of $C_{-1}X$, $C_{-2}X$, and $C_{-3}X$ peptides, respectively.

### RNA-seq analysis

$pth^{ts}$ cells were cultured at 30 °C in LB medium. In mid-log phase (OD$_{600}$ ≈ 0.5), the culture temperature was shifted from 30 °C to 43 °C; the cells were incubated for an additional 30 min and then harvested. Total RNA was extracted using a simple acidic phenol method described in[137]. Ten micrograms of total RNA was treated with RNase-free DNase (RQ1 DNase, Promega) and purified using the RNeasy Mini kit (QIAGEN). rRNAs were removed from the total RNA using the RiboMinus Transcriptome Isolation Kit, bacteria (Invitrogen), and then 100 ng of the purified total RNA was subjected to the construction of the cDNA library using the NEBNext Ultra II Directional RNA Library Prep Kit for Illumina (NEB). Deep sequencing was performed for a single-read (50 bp) using the HiSeq4000 (Illumina) at the Vincent J. Coates Genomics Sequencing Laboratory (UC Berkeley). The reads were processed with fastp[131] and mapped to the *E. coli* reference genome (NC_000913.2) using bowtie2[138]. The number of reads were counted using Ribo-Seq tools (https://github.com/ingolia-lab/RiboSeq). TPM was calculated based on the read number for each gene. RNA-seq data is deposited in DDBJ with the accession number DRA012753.

### Characterization of codon frequency at the drop-off sites

The number of codons at the drop-off sites for each cognate (Supplementary Fig. 11a) and $C_0X$ pep-tRNAs (Supplementary Fig. 11b) was counted as the frequency of codons at the drop-off site. We then plotted the frequency of codons at the drop-off sites for cognate versus $C_0X$ pep-tRNAs (Fig. 5h). In addition, we counted the codons 3' adjacent to the drop-off site and plotted them against the frequency of codons at the drop-off site for $C_0X$ pep-tRNAs (Fig. 5h).

The actual codon usage for the 5th through to 17th positions in all expressed mRNAs was calculated as follows. First, the number of sense codons in the 5th through to 17th positions of each mRNA was counted for each codon and multiplied by the ratio of TPM for each mRNA obtained from RNA-seq analysis. For each codon, the actual codon usage was calculated by summing up the calculated values for all transcripts. The frequencies of codons at the drop-off sites were plotted against the actual codon usage in the 5th through to 17th positions in transcriptome for the cognate (Supplementary Fig. 11c) and the $C_0X$ pep-tRNAs (Supplementary Fig. 11d). For further characterization, the drop-off site codons of cognate or $C_0X$ pep-tRNAs were classified into four groups based on the number of G or C in the three bases of codons, and the codon frequency in each group was calculated (Supplementary Fig. 11e, f). In addition, we analyzed codon–anticodon mismatches at the drop-off sites for 286 $C_0X$ pep-tRNAs, and found that 74 sites were pairings with a single mismatch and the remaining 212 were pairings with multiple mismatches (Supplementary Fig. 11g).

We used Canvas11, R(4.0.2), Microsoft Excel(2016), GraphPad prism9, ACD/ChemSketch(Freeware, 2018.2.1) and ChemDrow20.1 to draw figures and analyze statistical data. FLA-7000(Fujifilm) and MultiGauge V3.0 were used for gel images.

### Reporting summary

Further information on research design is available in the Nature Portfolio Reporting Summary linked to this article.

### Data availability

All data supporting the findings of this study are available within the paper, its Supplementary Information or Source data file. Ribosome profiling and RNA-seq data are deposited in DNA Data Bank of Japan (DDBJ) with the accession numbers DRA012800 and DRA012753. The mass spectrometry proteomics data have been deposited to the ProteomeXchange Consortium via the PRIDE partner repository with the dataset identifiers PXD030805, PXD030806, PXD030807 and PXD030808. *E. coli* genome reference used for RNA-seq and Ribosome profiling analyses is NC_000913.2. Source data are provided with this paper.

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

## Acknowledgements

We are grateful to the Suzuki lab members for technical assistance and fruitful discussion. This work was supported by Grants-in-Aid for Scientific Research on Priority Areas from the Ministry of Education, Culture, Sports, Science, and Technology of Japan (MEXT); Japan Society for the Promotion of Science (JSPS) (23112705, 26113003, 26220205, and 18H05272 to T.S., 26116003 and 25660053 to A.N.), and Exploratory Research for Advanced Technology (ERATO, JPMJER2002 to T.S.) from Japan Science and Technology Agency (JST). This work was also supported by JSPS KAKENHI Grant 16H06279 (PAGS) and the Vincent J. Coates Genomics Sequencing Laboratory at the UC Berkeley (NIH S10 OD018174), and Bioinformatics Analysis Environment Service on RIKEN Cloud at RIKEN Advanced Center for Computing and Communications.

## Author contributions

A.N., Y.N., Y.Y., Y.M., M.K., and T.T. performed all biochemical and genetic works assisted by K.M. and Y.S. T.F. and S.I. contributed to ribosome profiling and data analysis. A.N. and T.S. wrote this paper. All authors discussed the results and revised this paper. A.N. and T.S. designed the studies. T.S. supervised all the project.

## Competing interests

The authors declare no competing interests.
