## [Peer Review File · Nature Communications]

Quality control of protein synthesis in the early elongation stageREVIEWER COMMENTS

Reviewer #1 (Remarks to the Author):

This manuscript by Nagao et al. characterizes the peptidyl-tRNA species that drop off of the ribosome and accumulate in *E. coli* lacking the enzyme that hydrolyses the linkage between peptidyl-tRNA in solution, peptidyl hydrolase (PTH). Their careful MS analyses of these peptides greatly increase our understanding of drop off and the mechanism behind it. Surprisingly, they observe that nearly a third of the peptidyl-tRNAs that drop off the ribosome have the wrong amino acid at the C-terminus – that is, a miscoding event led to a mismatch between the tRNA and mRNA codon in the ribosomal P site, increasing the odds of drop off. Previous work by Zaher and Green showed that P-site codon anticodon mismatches can lead to pre-mature termination as release factor 2 is recruited (to a sense codon). The findings here expand the scope of this phenomenon, showing that miscoding leads to drop off early in elongation (in the first fifteen or so codons). Nagao et al. show that RF3 and RRF both play a role in promoting these drop off events and reveal sites where they are more likely and sites where they are less likely. The data are compelling and high quality, and their findings make an important contribution to our understanding of protein synthesis.

I have a few critical concerns about the way the data are presented and one or two conclusions are stated (major points):

1. The antibiotics erythromycin and tylosin increase peptidyl-tRNA drop off. The authors titrate tylosin to a concentration where they see the maximum level of peptidyl-tRNA drop off, and use this to normalize how much drop off is happening even without tylosin treatment. They then argue that 1 in 10 elongation cycles fails (early in the gene). The assumption that they make is that at the high concentration of tylosin “all elongating peptRNAs were dissociated under this condition” (page 8). I am not convinced that this is a good assumption. Even if all the ribosomes were bound to tylosin, many of them may keep translating without the peptidyl tRNA dropping off. Indeed, these antibiotics have some sequence selectivity, and the excellent work of Mankin and co-workers has shown that many ribosomes do in fact read through the early phase of elongation (without drop off) and stall later in the ORF (see Kannan et al in both *Cell* 2012 and in *PNAS* 2014). I think the safest course of action is to use the data in Fig 2C to talk about the role of sequence on drop-off, but not to make claims about the absolute fraction of nascent peptides that drop off, or the absolute rate at which this occurs (e.g. 1 in 10).
2. The ribosome profiling data in Fig 3F appear to show that in the absence of RRF, more ribosome density appears at the second, third, and fourth codons in genes on average, consistent with RRF promoting drop off early in the gene. Given the other results in the paper, where drop off occurs up to 15 codons or so into the ORF, this window seems more narrow than I would expect. The absolute height of start codon peaks and the dip following the start codon varies quite a bit in different ribosome profiling samples. I would like to see the data from the replicate that they describe. I would also recommend analyzing the publicly available RRF knockout ribosome profiling data of Saito et al (*eLife* 2020) and the RF3 knockout data of Baggett et al (*PLOS Genetics* 2017) to see if these trends hold true.
3. The first four figures and the corresponding text are very clear, but the presentation of the data in Figure 5 and the nomenclature are very hard to follow. First, the transition to the section “Deep analyses of individual pep-tRNAs” is surprising because the authors have not yet told us that the mistakes in the peptides are primarily at the C-terminus. The point of isolating individual tRNAs is to ask whether it was tRNA mischarging (by aaRS) or miscoding that led to the mistake. But that experiment doesn't make sense unless we know that it's the C-terminal amino acid which is incorrect. Second, the authors use the C0X and C-1X nomenclature to refer to the site where mistakes were made (in Fig 5C and 5D) long before they explain in the text what this nomenclature means. Please rewrite this section to flow more smoothly. Where is the mutation? How did it get there?

4. On page 18 the authors state “we conclude that decoding accuracy is not high enough in the initial stage of translation elongation.” To explain this observation, they later argue that poor codon usage leads to higher error rates early in the gene (in the model, Fig. 7). It is true that the authors see a very high rate of errors and drop off early in genes. But they can only see drop off of peptidyl-tRNA up to 15 codons or so, because after that the peptide is so strongly engaged with the nascent peptide exit tunnel that drop off becomes impossible. This means that the authors are blind to error rates later in ORFs. So it is too strong to conclude that decoding accuracy is lower in the initial stages of elongation vs the later stages, when they can’t observe error rates in later stages. The codon usage argument is not necessary – it is trying to explain an observation that may well not even be true. Note that in Fig 5F and 5G, that the mismatched peptidyl-tRNAs drop off when they are short, whereas the cognate peptidyl-tRNAs drop off even when they are longer. It is hard to explain that result with codon usage, which is bad even further downstream than the first 5-10 codons.

5. What is RF3 doing to promote drop-off? The paragraph in the discussion rightfully suggests that it promotes rotation which would be more possible with short peptides (like they were deacylated tRNA). But they only cite a single structural paper when there is additional biochemical evidence to support this idea. And many readers will wonder what a release factor is doing in this situation, in the absence of other release factors. I suggest adding the following points to the discussion: a) RF3 binds to ribosomes whether or not RF1 or RF2 is there and regardless of whether termination has occurred (Koutmou et al RNA 2014 and Adio et al eLife 2018) and b) RF3 causes rotation (Gao et al Cell 2007 and Sternberg et al NSMB 2009). Finally, I’ll note that RF3 does not seem to be essential to drop off, because the delta-prfC cells with PTHts alleles still die at 43 degrees (Fig. 3A) and recycling is still occurring. Perhaps rotation is still possible even without RF3 – it is a natural motion of the ribosome, less inhibited by short peptides perhaps.

6. The authors state “therefore, the pep-tRNA drop-off plays a major role in quality control of protein synthesis in the early elongation.” That is a pretty strong statement. It’s not clear how the importance of this mechanism compares to other mechanisms, initial selection, proofreading, post-PT quality control, etc. To get a sense of how big the effect is, they could perhaps monitor the expression level of their reporter protein with and without RF3, look at other RF3 phenotypes, or use a Farabaugh reporter where the misincorporation of a single amino acid leads to enzyme activity that can be measured. But it might be easier to just tone down the language a little.

Minor points:

I can’t distinguish the colors in Fig 1G and 1H, for which trace is which amino acid. And what order are they in top to bottom in the legend?

The small differences in Fig 2B might be clearer with log₂ y-axis.

Please add the culture conditions for Fig 5 and 6 to the main text (e.g. 43 degrees for 30 min).

The authors often use the term “misdecoding” or the verb, “misdecode.” This doesn’t sound right to my ear. While I recognize that they are trying to be specific about the step where the problem occurs, I would strongly prefer “miscoding” instead.

The model in Figure 5I suggests that at a hungry codon, drop off rates increase, and so do miscoding events, that then lead to drop off one codon downstream (with a non-cognate peptide). The data in 5E show miscoding on the Lys AAA codon. I would also expect to see drop off on the previous Asp codon (prior to mistakes being made). Is this peptide also observed?

Fig 3B and 3D show that RF3 and RRF have a role on dipeptides dropping off. Do they affect the longer nascent chain tRNAs as well?

Do RRF or RF3 specifically influence non-cognate drop off vs cognate drop off?

There is a prior paper on drop off, showing that the amino acid sequence in the first 5 codons affects protein yield (and I wished that our current authors would look at yield) including some lovely biochemical work. I was surprised not to see this work cited here and their results compared and contrasted. See Verma et al Nature Communications 2019.

Reviewer #2 (Remarks to the Author):

The manuscript of Nagao et al. describes the development of a methodology that allows for accurate identification in *E. coli* cells of species of peptidyl-tRNA that presumably drop-off from the ribosome at the early stages of elongation. With this methodology in hand, the authors perform a series of experiments that brings them to suggest that the bacterial cell is equipped with specific mechanisms to discard peptidyl-tRNAs from early elongation events where non-cognate amino acids are wrongly incorporated. The work is extremely thorough, well-written, and well-presented. The findings contribute to our understanding of the different strategies that cells use for translation quality control and therefore will be of interest to many readers of the journal.

Following are comments and suggestions aimed for the authors to clarify and strengthen specific points:

- The aa- and peptidyl-tRNAs isolated by the described procedure are likely present in the cytosol at the moment of cell harvesting but they could also have been associated with ribosomes and drop-off could have occurred during the procedure. Authors should discuss this possibility.

- Because some of the characterized peptidyl-tRNAs were obtained from the pth-ts cells grown at 37C (instead of the non-permissive temperature of 43C), authors state several times that peptidyl-tRNA drop-off occurs during normal non-stressed conditions. However, the growth of the pth-ts cells is compromised even at 30 C. Therefore, for these cells, 37 C can be seen as a stress condition. Authors should tone down the statements where this is mentioned and simply suggest that peptidyl-tRNA drop-off may occur during normal growth.

- Saito et al. (eLife 2020;9:e59974) also performed ribosome profiling in *E. coli* cells depleted of RRF but, in contrast to the results presented in this work, they did not report a genome-wide accumulation of ribosomes in the early codons of genes. Could authors comment why the different results if this is really the case?

- The authors imply in several places in the manuscript that peptidyl-tRNAs drop off from the P site even though, from my understanding, they provide no evidence that this is the case. Drop-off could also occur while transitioning to the A/P state during translocation.

- When discussing the mechanism of drop-off, the authors suggest that RF3 is able to translocate peptidyl-tRNA to the P/E hybrid state. However, even for peptidyl-tRNAs with short nascent chains unable to establish multiple contacts in the tunnel, this is highly unlikely since there is no room in the E site to accommodate an amino acid.

- I'd suggest for the authors to shorten the discussion by no re-stating in this section many of the results. Also, it is unnecessary to present the model described in Figure 7 in both, the legend to the figure and in the discussion.

Reviewer #3 (Remarks to the Author):

In the early stages of translation elongation ribosomes are susceptible to peptidyl-tRNA drop-off. Stability of these complexes depends on the nascent peptide interacting with the ribosomal tunnel (Chadani et al. EMBO J 2021), and it dramatically increases after 6 amino acids are incorporated (Heurgué-Hamard et al. EMBO J 2000), and the date of drop-off is heavily influenced by the nature of the tRNA / tRNA isoacceptor (Dinçbas et al. JMB 1999) and peptide (Chadani et al. EMBO J 2021, Chadani et al. Mol Cell. 2017). The process of drop-off is kinetically competing with translation termination, and bacterial termination factor RF3 plays a crucial role in this (Zaher and Green, Cell 2011), with mismatched P-site tRNA stimulating premature peptide release (Zaher and Green, Nature 2009). Studying the process of peptidyl-tRNA drop-off in the cell on the global scale is challenging due to the lack of appropriate techniques that would allow global detection and quantification cellular peptidyl-tRNA species.

The manuscript by Nagao and colleagues is a real tour de force, both technologically and in terms of the amount of performed experiments. Importantly, the authors set up, validate and apply an MS-based approach for global detection and quantification cellular peptidyl-tRNA species. I believe this method has a great potential for discovering new biology in the future, e.g. for identifying new ORFs that generate high levels of drop-off similarly to lambda-phage-encoded bar minigenes that mediate translational shutoff (Ontiveros et al. JMB 1997). I have only minor comments / suggestions.

1. P. 7: When it comes to detection of longer peptidyl-tRNA species, maybe the authors could more explicitly spell out what part of the signal (i.e. lack of it) is that these are not as abundant as the less stably ribosome-associated short peptidyl-tRNA species and what part is due to potential complications associated with detection of these species by mass spectrometry.
2. P. 8: Regarding the effect of macrolide tylosin on drop-off: biochemical evidence suggests that – at least in some cases – polypeptides longer than 2-3 amino acids are made (see (Yakhnin et al. mBio 2019)). Could you please comment on that / refine the statement?
3. P. 14: Maybe one could use plating assays in the presence of sub-MIC concentration of miscoding-inducing antibiotics to bridge the data shown on Figure 3 with the section 'Misdecoding induces the pep-tRNA drop-off'?
4. P. 16: When discussing frequencies of drop-off as a function of peptide composition, maybe one could connect to earlier biochemical data better.

Reviewer #4 (Remarks to the Author):

This work from Tom Suzuki's group systematically investigates how peptidyl-tRNA drop-off occurs (from the translating ribosomes) in *Escherichia coli*, and proposes the phenomenon to be a ribosomal/cellular mechanism of quality control operating during the early stages of translation elongation. The initial discovery of peptidyl-tRNA hydrolase (Pth) by the groups of Chapeville and RajBhandary in the late 60s set the stage for the investigations on the role of Pth (an essential enzyme) in translation. The genetic studies of peptidyl-tRNA drop-off were then greatly facilitated by the isolation of temperature sensitive mutants in peptidyl-tRNA hydrolase (Pth) by the Menninger group in early 70s. While many other groups have since then contributed to the understanding of the role of RRF, RF3 and other components of the translation machinery in peptidyl-tRNA drop-off, a deep understanding of the mechanism of peptidyl-tRNA drop-off from the ribosome remained wanting.

This investigation begins with the validation of the earlier key genetic findings of the peptidyl-tRNA drop-off showing the role of RRF and RF3 in peptidyl-tRNA drop-off and the crucial role played by Pth in recycling of tRNAs. A deficiency of Pth quickly depletes aminoacyl-tRNA pool to accumulate as

peptidyl-tRNAs. Majority of the peptidyl-tRNAs have short peptides of 2 to 5 residue length but the ones having 15 residues were also detected. RRF deficiency in the *frr* ts strains reveals ribosome stalling in the early stages of elongation (and thus a role of RRF in fidelity of translation soon after initiation). The authors have developed robust mass spectrometry (MS) pipeline to sequence the peptides attached to tRNAs. Importantly, the authors have also optimised their method of separating tRNAs by reciprocal circulating chromatography (RCC) to now separate the tRNAs with peptides attached to them (into 42 species). Given that the ester bond that connects the peptides with the tRNAs is highly susceptible to heat/pH, this is a rather challenging task, and is certainly a 'work of art' in science! And, it is these achievements of the authors that have allowed them to sequence the variety of the peptides attached to each of the individual tRNA species and map them to the starting regions of the open reading frames (ORF) in *E. coli* genome to decipher the mechanistic details peptidyl-tRNA drop-off. The authors categorised the dropped-off peptidyl-tRNAs into the cognate and noncognate categories. The cognate ones do not show any mismatches with the ORFs of the genes. However, the noncognate ones show single mismatches at 0, -1, -2, or -3 positions (C0X, C-1X, C-2X and C-3X, respectively). Weak/mismatched pairing between codon/anticodon appears to be a major reason for peptidyl-tRNA drop-off. To validate the hypothesis, the authors first deleted, from the *E. coli* genome, some of the genes (*lpp*, *rplJ*) predicted from the MS analysis of the dropped-off peptidyl-tRNAs to show that the corresponding peptidyl-tRNAs were no longer present in the pool. The authors then designed a clever reporter system having a novel sequence (not found in any other ORF in *E. coli*) in the early part of the ORF and exploited the property of the elongator tRNA(Met) to occasionally decode Ile codon (AUA). Not satisfied with just this, the authors then deleted *tmcA* gene encoding tRNA acetyltransferase that modifies the C34 in the anticodon to N4-acetylcytidine (ac4C). The modification is important in preventing misreading of AUA codon as Met. A deficiency of *TmcA* led to increased misreading of AUA in the reporter (and drop-off of the peptidyl-tRNAs of C0X and C-1X class). The reporter has allowed the authors to carry out a good quantitative analysis.

The authors have made several other important finds. For example, based on the observation that the initiator Met (fMet) is missing in most of the peptidyl-tRNAs the authors suggest that peptide deformylase (PDF) and Met aminopeptidase (MAP) work even on the free peptidyl-tRNAs having as short as tripeptide chain. The study also allows the authors to comment on the species of aminoacyl-tRNAs that deplete fast upon Pth deficiency.

Overall, I find that the present work is extensive, thorough and the manuscript is well written for the most parts. The experiments have been designed in a competent manner and demonstrate authors' indulgence to uncover finer details of the phenomenon. I have following points for the authors to clarify/consider.

1. In the abstract, delete, "we happened to find" and rephrase the statement (I am sure this was not a serendipitous finding).
2. On page 4, last line: change 'enables' to 'enabled'.
3. The latter half of the last long paragraph of Introduction section is pretty much a reproduction of the Abstract of the manuscript. This repetition is unnecessary and should be replaced with just one or two statements.
4. In many places in the Results section the authors give an impression that the Pth ts strain was grown at 43 °C, which is a nonpermissive temperature (for example, see the last paragraph on page 6; and then again in the next paragraph on page 7). The authors must have grown the culture at the permissive temperature and then shifted the culture to the nonpermissive temperature. This should be carefully worded.
5. On page 9, second paragraph, last statement: the authors state 'our' hypothesis. The models for drop-off of peptidyl-tRNAs have been proposed earlier (Singh et al.; doi: 10.1016/j.jmb.2008.05.033).

6. Likewise, the role of RRF in the fidelity of translation has been proposed on the basis of genetic analyses in the above referred paper and also in another paper (Seshadri et al.; doi:10.1111/j.1365-2958.2009.06685.x)

7. On page 11, penultimate paragraph, last statement: change, 'lied' to 'laid'.

8. On page 13, second paragraph, first line: change, 'them' to 'then' (there is at least one more place in the manuscript where a similar correction is required).

9. Page 16, first paragraph: may be better to say, 'growth' rather than 'living'.

10. Page 17, second paragraph: 'deacylated-tRNA' not 'deacyl-tRNA'

11. Page 19, second paragraph: 'avoid' not 'loosen'

12. While the authors have appropriately discussed shifting of the peptidyl-tRNA into the P/E (and p/R) site for drop-off, the authors may also discuss on the recruitment of incorrect aminoacyl-tRNA in the A site to begin with, and the role of uS12 in this. A couple of recent studies (Datta et al. doi: 10.1111/mmi.14861; doi: 10.1111/mmi.14675) have shown genetic interaction between uS12 and RRF and have also proposed a model that the authors would find relevant.

13. The authors do not comment anything on the effect of proline on the -1 or 0 site in peptidyl-tRNA drop-off. This may be discussed.

14. In Table S1, for the non-cognate peptides (of COX type) of *higA*, *rplO*, and *sspA* the last amino acid is the same as coded for by the original sequence (ecocyc.org). The reviewer has not checked all of the sequences, and the authors should look at it once again. While this is unlikely to change any of the major findings, the data in the Table need to be reviewed carefully.

15. Both the Figs. 3a and 3e show growth of *pthts* strain at 39 °C. The dilutions are similar in both and the pictures are taken after 24 h. Why then the *pthts* strain is growing better in 3e than 3a? If the growth times are different, they need to be corrected.

Reviewer #5 (Remarks to the Author):

In this paper, the authors attempted to reveal a new quality control mechanism for protein synthesis via the dissociation of peptide-linked tRNAs from the ribosome at the early stage of translation elongation. The authors established a highly sensitive method for direct profiling of nascent peptide-binding tRNAs using mass spectrometry, and identified thousands of peptides from those accumulated in *E. coli pthts*. They found that one-third of the peptides were miscoded peptides, and most of them had the wrong amino acid residues introduced by mismatched tRNAs at their C-termini. The authors have also confirmed the validity of this observation using a reporter construct.

Although the authors' method for detecting peptide-linked tRNAs is novel and impressive, I believe that this paper is not appropriate for *Nat Commun* for the following three reasons.

1. The novelty of this paper is low.

The following facts, which constitute the important framework of this paper, have already been reported:

(i) In *E. coli*, the error frequency of codons close to the initiation codon has been found to be significantly low (Parker J. *Biochem Biophys Res Commun.* 1984; Precup J. *Mol Gen Genet.* 1989),

probably due to the rapid dissociation of erroneous short peptidyl tRNAs are easily dissociated before or during translation (Rodnina MV. *Annu Rev Biochem.* 2001).

(ii) Codon-anticodon mismatch on the ribosome of mRNA and tRNA causes peptide drop-off (Zaher HS. *Nature.* 2009).

(iii) It has already been shown that RF3 prematurely terminates peptides with mistakes during protein synthesis on bacterial ribosomes (Zaher HS. *Cell.* 2011).

The authors have proved the same facts by using MS, but there is nothing novel in these. Also, these reports are cited in the text, but the reviewer could not understand what novelty this paper added to this field in addition to the previous reports.

2. Strange handling of peptides

After purifying pep-tRNA from *E. coli*, the authors recovered the peptide by treating it with 0.3% ammonia (according to my calculations, this is a solution of about pH 11) at 50°C for 30 minutes. Under these conditions, ammonia-induced beta-elimination must be occurring. Cysteine is the most susceptible to beta-elimination, and data supporting the occurrence of beta-elimination in the peptides can be found in Table S1. The peptides listed in Table S1 have a total of 5115 amino acid residues, of which only three are cysteine. Since proteins usually contain one cysteine residue for every 200 amino acid residues, the proportion shown in Table S1 is very small. Therefore, this is evidence that beta-elimination is occurring. The residues after beta-elimination are subjected to addition reactions by various substances in the solution. Therefore, these peptides would either be misidentified as other sequences or recognized as no-cognate peptides. Under these conditions, in addition to cysteine, serine and threonine may have also been subjected to beta-elimination as well. The resulting peptide sequences are likely to contain many errors and serious biases. It would be inappropriate to make conclusions based on these peptide sequences.

In order to prove that there is little mistake or bias in the peptide sequence, a control experiment is needed to show that there is no beta-elimination in many peptides under these conditions.

Alternatively, more mild methods could be applied, such as the use of peptidyl-tRNA hydrolase to cleave peptides from pep-tRNA or the use of nuclease P1 to analyze the sequence of peptides that still have Ado bound. The reviewer believes that the authors should conduct such experiments and prepare a paper based on the new data.

The authors focused their analysis on acetylated pep-tRNA. In order to acetylate pep-tRNA in their method, the formyl group of newly synthesized pep-tRNA must be removed by the peptide deformylase after dissociation from the ribosome, and the amino group must be exposed at the peptide amino terminus. This amino group-exposed pep-tRNA should generally be an unstable material that exists only for a very short time in the course of its degradation. The reason is that these are intermediates that are generated during the degradation process after dissociation from the ribosome, in which the formyl group and methionine are removed by peptide deformylase and methionine aminopeptidase, respectively. In general, reproducibility is required for experiments on unstable substances. Therefore, all of the peptide sequences submitted as data must be either validated by multiple similar analyses, or by sequence analysis of samples of *E. coli* dysfunctional for peptide deformylase by actinonin inhibition, or by obtaining similar peptide sequences in the absence of acetylation.

3. We don't know if the same phenomenon occurs in wild-type *E. coli*.

In this paper, the authors performed almost all of their experiments using mutants of the pthts as a background strain. Therefore, it remains to be verified whether the same happens in wild-type *E. coli*. The scale of the data obtained from wild-type *E. coli* may be smaller than that obtained from pthts strains, but its pep-tRNA data is necessary, and its principle must be the same as that given by pthts strains. We need to know about universal phenomena, not pthts strain-specific phenomena. If such observations cannot be made, then mass spectrometry is still not sensitive enough to be used for analysis in this field.

Others

What is the origin of aa-tRNA?

The authors have analyzed aa-tRNA as well as pep-tRNA (p6, line 11-). Is this aa-tRNA observing a newly aminoacylated tRNA, or is it observing a substance in the process of degradation of pep-tRNA? There is no mention of this, and it was difficult to read the paper because I could not predict where the story would go at the beginning of the paper. The reviewer thinks that this should be clearly stated in the paper.

First, we thank the reviewers for their careful and helpful review of our manuscript, and for providing valuable suggestions for its improvement. Our point-by-point responses to each of the reviewers' comments are shown below. Changes to the main text are marked in yellow.

Response to Reviewer #1's comments

This manuscript by Nagao et al. characterizes the peptidyl-tRNA species that drop off of the ribosome and accumulate in *E. coli* lacking the enzyme that hydrolyses the linkage between peptidyl-tRNA in solution, peptidyl hydrolase (PTH). Their careful MS analyses of these peptides greatly increase our understanding of drop off and the mechanism behind it. Surprisingly, they observe that nearly a third of the peptidyl-tRNAs that drop off the ribosome have the wrong amino acid at the C-terminus – that is, a miscoding event led to a mismatch between the tRNA and mRNA codon in the ribosomal P site, increasing the odds of drop off. Previous work by Zaher and Green showed that P-site codon anticodon mismatches can lead to pre-mature termination as release factor 2 is recruited (to a sense codon). The findings here expand the scope of this phenomenon, showing that miscoding leads to drop off early in elongation (in the first fifteen or so codons). Nagao et al. show that RF3 and RRF both play a role in promoting these drop off events and reveal sites where they are more likely and sites where they are less likely. The data are compelling and high quality, and their findings make an important contribution to our understanding of protein synthesis.

Response: We really appreciate deep understanding this study and many positive words.

I have a few critical concerns about the way the data are presented and one or two conclusions are stated (major points):

1. The antibiotics erythromycin and tylosin increase peptidyl-tRNA drop off. The authors titrate tylosin to a concentration where they see the maximum level of peptidyl-tRNA drop off, and use this to normalize how much drop off is happening even without tylosin treatment. They then argue that 1 in 10 elongation cycles fails (early in the gene). The assumption that they make is

that at the high concentration of tylosin “all elongating peptRNAs were dissociated under this condition” (page 8). I am not convinced that this is a good assumption. Even if all the ribosomes were bound to tylosin, many of them may keep translating without the peptidyl tRNA dropping off. (see Indeed, these antibiotics have some sequence selectivity, and the excellent work of Mankin and co-workers has shown that many ribosomes do in fact read through the early phase of elongation (without drop off) and stall later in the ORF Kannan et al in both Cell 2012 and in PNAS 2014). I think the safest course of action is to use the data in Fig 2C to talk about the role of sequence on drop-off, but not to make claims about the absolute fraction of nascent peptides that drop off, or the absolute rate at which this occurs (e.g. 1 in 10).

Response: Thank you very much for the critical comment. As shown by Shura Mankin’s pioneer works, macrolides do not dissociate all elongating pep-tRNAs, and certain peptides can bypass the macrolides bound to the exit tunnel. In this revision, we mentioned this important fact with appropriate references. Thus, we don’t know if all elongating pep-tRNA are dissociated even in the presence of high concentration of Tyl. We just intended to show that pep-tRNAs are dissociated highly frequently even under normal culture conditions, and compare the amount of pep-tRNAs dissociated from the ribosome with and without Tyl. So, we deleted a series of statements for absolute rate of pep-tRNA drop off in elongation cycles. However, comparison of the drop-off frequency of each pep-tRNA with and without Tyl would be important to know different frequency of pep-tRNA drop-off due to dipeptide composition.

2. The ribosome profiling data in Fig 3F appear to show that in the absence of RRF, more ribosome density appears at the second, third, and fourth codons in genes on average, consistent with RRF promoting drop off early in the gene. Given the other results in the paper, where drop off occurs up to 15 codons or so into the ORF, this window seems more narrow than I would expect. The absolute height of start codon peaks and the dip following the start codon varies quite a bit in different ribosome profiling samples. I would like to see the data from the replicate that they describe. I would also recommend analyzing the publicly available RRF knockout ribosome profiling data of Saito et al (eLife 2020) and the RF3 knockout data of Baggett et al (PLOS Genetics 2017) to see if these trends hold true.

Response: We really appreciate this comment and suggestion to confirm our observation by analyzing publicly available datasets for RRF knockdown and RF3 KO. Regarding peptide length of pep-tRNAs dissociated from the elongating ribosome, we actually detected pep-tRNAs having longer peptides up to 16 mer in the analysis of the isolated pep-tRNAs (Figure 5f). However, a large population of the cognate and non-cognate peptides is distributed in the mass range from 200 to 600, suggesting that short peptides are main fractions of the dissociated peptides. Thus, it is quite reasonable to observe more ribosome density at the 2nd-4th codons in the absence of RRF. Shorter pep-tRNAs might more readily allow the ribosome to form a P/E or intermediate states, like p/R or pe/E state. As requested, the metagene analyses of RPFs from the replicates of WT and *frr*^{ΔS} (in Figure A) showed the high reproducibility of this molecular phenotype. Figure 3f has been replaced with this new figure.

According to the reviewer's suggestion, we analyzed the data of Saito et al. (Figure B) and Baggett et al. (Figure C). The results from Saito et al. show that RRF knockdown results in the ribosome accumulation in the early stage of translation, and that this accumulation increases over time of knockdown (Figure B). This is completely consistent with our observation. On the other hand, the data from RF3 knockout strain in Baggett et al. (Figure C) show no clear accumulation of ribosomes in

the early stage of translation, indicating that RF3 may have a smaller contribution to pep-tRNA drop-off than RRF does. However, we here notice a large variation in the footprint length in the data of Baggett et al., when compared to our dataset and Saito et al. The metagene analysis of Baggett et al. might be affected by variable length of the footprints. In this revision, we added the data from Saito et al. in **Figure S3c**. Similar tendency of the ribosome profiling data from two independent groups provided solid evidence that RRF is significantly involved in pep-tRNA drop-off in the early elongation stage.

3. The first four figures and the corresponding text are very clear, but the presentation of the data in Figure 5 and the nomenclature are very hard to follow. First, the transition to the section “Deep analyses of individual pep-tRNAs” is surprising because the authors have not yet told us that the mistakes in the peptides are primarily at the C-terminus. The point of isolating individual tRNAs is to ask whether it was tRNA mischarging (by aaRS) or miscoding that led to the mistake. But that experiment doesn't make sense unless we know that it's the C-terminal amino acid which is incorrect. Second, the authors use the C0X and C-1X nomenclature to refer to the site where mistakes were made (in Fig 5C and 5D) long before they explain in the text what this nomenclature means. Please rewrite this section to flow more smoothly. Where is the mutation? How did it get there?

Response : Indeed, we realize that the transition to the section "Deep analyses of individual pep-tRNAs" was a bit confusing. First, let us explain the history of the analyses performed in this section. The pep-tRNA profiling in the previous section reported single substitutions occurring in the drop-off peptides. Therefore, we raised two possibilities to generate those non-cognate peptides. One is misaminoacylation of some tRNAs, the other is miscoding at A-site of ribosome. First, we carefully examined the possibility of misaminoacylation by analyzing amino acids attached to isolated tRNAs, and found no evidence of misaminoacylation in each tRNA (**Figure S6**). Therefore, we next sequenced the nascent peptides attached to the isolated pep-tRNAs and compared them with N-terminal sequences of *E. coli* ORFs (**Figures 5b and S7**). It turned out that a majority of single substitutions took place at the C-termini of the peptides, prompting us to come up with the idea that non-cognate aa-tRNA miscodes the A-site codon, followed by peptide bond formation to generate non-cognate pep-

tRNA which is dissociated from the ribosome. We rewrote this part so that readers can understand the flow of the story.

Regarding the second point, as pointed out, Figure 5c appeared before explaining the nomenclatures $C_{0-3}X$ for non-cognate pep-tRNAs. We added some explanation for this.

4. On page 18 the authors state “we conclude that decoding accuracy is not high enough in the initial stage of translation elongation.” To explain this observation, they later argue that poor codon usage leads to higher error rates early in the gene (in the model, Fig. 7). It is true that the authors see a very high rate of errors and drop off early in genes. But they can only see drop off of peptidyl-tRNA up to 15 codons or so, because after that the peptide is so strongly engaged with the nascent peptide exit tunnel that drop off becomes impossible. This means that the authors are blind to error rates later in ORFs. So it is too strong to conclude that decoding accuracy is lower in the initial stages of elongation vs the later stages, when they can’t observe error rates in later stages. The codon usage argument is not necessary – it is trying to explain an observation that may well not even be true. Note that in Fig 5F and 5G, that the mismatched peptidyl-tRNAs drop off when they are short, whereas the cognate peptidyl-tRNAs drop off even when they are longer. It is hard to explain that result with codon usage, which is bad even further downstream than the first 5-10 codons.

Response: As this reviewer pointed out, we cannot conclude that decoding accuracy is lower in the early stages of elongation than the later stages, because we did not measure the error rates in later stages. However, in the early stage, we detected a highly abundant non-cognate pep-tRNAs bearing peptides shorter than 7 mer, whereas non-cognate pep-tRNAs with peptides longer than 10 mer are barely detectable (Figure 5f). Having this observation, we can safely claim that early translation up to codon position 10 are not accurate, and pep-tRNA drop off ensures precise translation. So, we don’t compare decoding accuracy between early and later stages of elongation, but just would like to emphasize high error rate of translation at very beginning of ORF. Regarding codon usage argument, we here revise and compare codon usage distributions between 2-8th and 9-15th positions (Figure S13a), and found that the codon usage distribution in the 2-8th positions ($\sigma=0.745$) is quite different from that in the 9-15th positions ($\sigma=0.297$) (Figure S13b). Thus, a large deviation of the codon usage bias in the beginning of ORF

might be one reason to explain high error rate in the early elongation stage. To take this reviewer's comment into account, we revised and toned down this argument in this revision.

5. What is RF3 doing to promote drop-off? The paragraph in the discussion rightfully suggests that it promotes rotation which would be more possible with short peptides (like they were deacylated tRNA). But they only cite a single structural paper when there is additional biochemical evidence to support this idea. And many readers will wonder what a release factor is doing in this situation, in the absence of other release factors. I suggest adding the following points to the discussion: a) RF3 binds to ribosomes whether or not RF1 or RF2 is there and regardless of whether termination has occurred (Koutmou et al RNA 2014 and Adio et al eLife 2018) and b) RF3 causes rotation (Gao et al Cell 2007 and Sternberg et al NSMB 2009). Finally, I'll note that RF3 does not seem to be essential to drop off, because the delta-prfC cells with PTHts alleles still die at 43 degrees (Fig. 3A) and recycling is still occurring. Perhaps rotation is still possible even without RF3 – it is a natural motion of the ribosome, less inhibited by short peptides perhaps.

Response: We really appreciate these constructive comments and letting us to know related findings with appropriate citations. Spontaneous rotation without RF3 is quite possible. We fully revised a series of descriptions on the roles of RF3 in pep-tRNA drop off with appropriate references.

6. The authors state “therefore, the pep-tRNA drop-off plays a major role in quality control of protein synthesis in the early elongation.” That is a pretty strong statement. It's not clear how the importance of this mechanism compares to other mechanisms, initial selection, proofreading, post-PT quality control, etc. To get a sense of how big the effect is, they could perhaps monitor the expression level of their reporter protein with and without RF3, look at other RF3 phenotypes, or use a Farabaugh reporter where the misincorporation of a single amino acid leads to enzyme activity that can be measured. But it might be easier to just tone down the language a little.

Response: As pointed out, we do not know how much pep-tRNA drop-off actually contributes to the quality control in the early elongation stage, when compared to the

other mechanisms such as initial selection and proofreading. So, we toned down the expression from “plays a major role” to “plays a role”.

Minor points:

I can't distinguish the colors in Fig 1G and 1H, for which trace is which amino acid. And what order are they in top to bottom in the legend?

Response: As suggested, we reconsidered the trace colors in Figure 1gh. The color code for each amino acid has been determined based on the hydrophobicity of amino acids; warm and cool colors are given to hydrophobic and hydrophilic amino acids, respectively. The point markers are also revised to make them easy to distinguish. The orders of amino acids and dipeptides are based on hydrophobicity of amino acids reported by Kyte and Doolittle, J Mol. Biol. (1982).

The small differences in Fig 2B might be clearer with log₂ y-axis.

Response: Thank you for the advice. But here we normalized the peak intensity of each aa-tRNA and pep-tRNA by that of fMet-tRNA in each condition. So, we cannot plot aa/pep-tRNA species not detected by mass spec in the log₂ y-axis.

Please add the culture conditions for Fig 5 and 6 to the main text (e.g. 43 degrees for 30 min).

Response: We added the culture conditions for Figures 5 and 6 in the main text.

The authors often use the term “misdecoding” or the verb, “misdecode.” This doesn't sound right to my ear. While I recognize that they are trying to be specific about the step where the problem occurs, I would strongly prefer “miscoding” instead.

Response: As suggested, we unified the usage of term “miscoding” throughout the manuscript.

The model in Figure 5I suggests that at a hungry codon, drop off rates increase, and so do miscoding events, that then lead to drop off one codon downstream

(with a non-cognate peptide). The data in 5E show miscoding on the Lys AAA codon. I would also expect to see drop off on the previous Asp codon (prior to mistakes being made). Is this peptide also observed?

Response: Thank you for asking us. The answer is YES. We clearly observed the peptide, ALNLQD, in the data from isolated tRNA^{Asp} (see Table S1), suggesting that ALNLQD-tRNA^{Asp} dissociated from ribosome at 7th codon (Asp codon) of *rplJ* mRNA. As shown in Figure 5h, we found a clear correlation between the drop-off codons for C₀X pep-tRNAs and their 5' adjacent codons for the cognate pep-tRNAs among all pep-tRNA identified in this study.

Fig 3B and 3D show that RF3 and RRF have a role on dipeptides dropping off. Do they affect the longer nascent chain tRNAs as well? Do RRF or RF3 specifically influence non-cognate drop off vs cognate drop off?

Response: In the experiments in Figure 3b and 3d, we only analyzed dipep-tRNAs but not longer pep-tRNAs. Regarding non-cognate versus cognate drop off, it is impossible to discriminate these two species from dipep-tRNAs. Given the mechanism of pep-tRNA drop-off, it is likely that RRF and RF3 also promote the dissociation of longer pep-tRNAs as well as both cognate and non-cognate pep-tRNAs.

There is a prior paper on drop off, showing that the amino acid sequence in the first 5 codons affects protein yield (and I wished that our current authors would look at yield) including some lovely biochemical work. I was surprised not to see this work cited here and their results compared and contrasted. See Verma et al Nature Communications 2019.

Response: We appreciate this suggestion. Indeed, Verma et al (Nat Comm 2019) is an excellent study that comprehensively examined the impact of initial ORF sequences on their translation efficacy. They found that GFP expression is strongly dependent on codon positions at 3 to 5, regardless of their downstream mRNA and protein sequence *in vitro* and *in vivo*. They observed ribosome stalling at early elongation of the lowly-expressing mRNA construct. The majority of the arrested ribosomes exhibited a non-canonical state which is probably induced by the nascent peptide interacting with the ribosome tunnel. The authors speculated that the arrested ribosome might be resolved through pep-tRNA drop-off and subsequent ribosome recycling. We have properly mentioned this study by

citing this paper in introduction and discussion.

In addition, we here investigated initial ORF sequences in association with the drop off efficiency and the protein expression. We first took 678 ORFs that are highly expressed in *E. coli* with top 20% TPM based on our RNA-seq data (Table S2). In 678 ORFs, pep-tRNAs were detected from 161 ORFs (Groups 1 and 2), but not detected from 517 ORFs (Group 3) (Figure S12a). The 161 ORFs are further divided into two groups, 73 ORFs (Group 1) in which pep-tRNAs are dissociated from 5th and 6th codon positions (Figure S12a), and 88 ORFs (Group 2) in which pep-tRNAs are dissociated from 7th codon position or later positions (Figure S12a). The Group 1 pep-tRNAs are efficiently dissociated at earlier stage (5 and 6 positions) of elongation, whereas the pep-tRNAs of Groups 2 and 3 are less efficiently dissociated at the same stage. For each group, we obtained GFP scores (Verma et al., Nat Comm 2019) corresponding to 3rd-5th codons nucleotide sequences of ORFs, and then compared distribution of their GFP scores between 3 groups. As shown in Figure S12b, the GFP scores of Group 1 show significantly lower than those of Groups 2 and 3. No significant difference between Groups 2 and 3 is found. This observation strongly suggests that pep-tRNAs are efficiently dissociated from *E. coli* ORFs having initial sequences with low translational efficiency.

Reviewer #2 (Remarks to the Author):

The manuscript of Nagao et al. describes the development of a methodology that allows for accurate identification in *E. coli* cells of species of peptidyl-tRNA that presumably drop-off from the ribosome at the early stages of elongation. With this methodology in hand, the authors perform a series of experiments that brings them to suggest that the bacterial cell is equipped with specific mechanisms to discard peptidyl-tRNAs from early elongation events where non-cognate amino acids are wrongly incorporated. The work is extremely thorough, well-written, and well-presented. The findings contribute to our understanding of the different strategies that cells use for translation quality control and therefore will be of interest to many readers of the journal.

Response: We really appreciate these positive words.

Following are comments and suggestions aimed for the authors to clarify and strengthen specific points:

- The aa- and peptidyl-tRNAs isolated by the described procedure are likely present in the cytosol at the moment of cell harvesting but they could also have been associated with ribosomes and drop-off could have occurred during the procedure. Authors should discuss this possibility.

Response: Thank you for this comment. Pep-tRNAs extending on the ribosome should have formyl-Met at the N-terminus. However, most of pep-tRNAs detected in this study had acetylated N-termini, suggesting that they have free amino group. Considering that the formyl group of the pep-tRNAs should be enzymatically removed by PDF after dissociation from the ribosome, most of pep-tRNAs detected in this study are not elongating pep-tRNAs on the ribosome. We added a description for this point in the 2nd section of the result in the main text.

- Because some of the characterized peptidyl-tRNAs were obtained from the pth-ts cells grown at 37C (instead of the non-permissive temperature of 43C), authors state several times that peptidyl-tRNA drop-off occurs during normal non-stressed conditions. However, the growth of the pth-ts cells is compromised even at 30 C. Therefore, for these cells, 37 C can be seen as a stress condition. Authors should tone down the statements where this is mentioned and simply suggest that peptidyl-tRNA drop-off may occur during normal growth.

Response: As suggested, we tone down the statement.

-Saito et al. (eLife 2020;9:e59974) also performed ribosome profiling in E. coli cells depleted of RRF but, in contrast to the results presented in this work, they did not report a genome-wide accumulation of ribosomes in the early codons of genes. Could authors comment why the different results if this is really the case?

Response: Thank you for this critical comment. The reviewer #1 also provided a similar suggestion. We analyzed the data of Saito et al. (Figure B) and found that RRF knockdown results in the ribosome accumulation in the early stage of translation, and that this accumulation increases over time of knockdown. This is completely consistent with our observation. Similar tendency of the ribosome profiling data from two independent groups provided solid evidence that RRF is significantly involved in pep-tRNA drop-off in the early elongation stage. In this revision, we added the data from Saito et al. in Figure S3c.

-The authors imply in several places in the manuscript that peptidyl-tRNAs drop off from the P site even though, from my understanding, they provide no evidence that this is the case. Drop-off could also occur while transitioning to the A/P state during translocation.

Response: As pointed out, it is possible to consider that pep-tRNA dissociates from A/P hybrid state. However, according to previous studies, it is known that pep-tRNA drop-off is enhanced by RRF. Thus, it is quite natural to consider that pep-tRNA primarily dissociates from P-site or P/E hybrid state (including chimeric states), because RRF acts on the ribosome with vacant A-site.

-When discussing the mechanism of drop-off, the authors suggest that RF3 is able to translocate peptidyl-tRNA to the P/E hybrid state. However, even for peptidyl-tRNAs with short nascent chains unable to establish multiple contacts in the tunnel, this is highly unlikely since there is no room in the E site to accommodate an amino acid.

Response: Thank you for raising this critical point. When pep-tRNA with short chain dissociates from the ribosome, CCA-terminus with nascent peptide needs to detach

from the tunnel. Then, the pep-tRNA might move to P/E site or its relevant state. If deacyl-tRNA binds to P/E state, its CCA-terminus specifically interacts with E site on the large subunit. As suggested, there is no room in the E site to accommodate the nascent peptide. So, the nascent peptide might weaken the interaction between CCA-terminus of pep-tRNA and E-site of 50S subunit, facilitating pep-tRNA drop-off. We added this point in discussion.

-I'd suggest for the authors to shorten the discussion by no re-stating in this section many of the results. Also, it is unnecessary to present the model described in Figure 7 in both, the legend to the figure and in the discussion.

Response: Thank you for the suggestion. We shortened the discussion as much as possible by removing redundant descriptions. In addition, we also shortened the legend for Figure 7.

Reviewer #3 (Remarks to the Author):

In the early stages of translation elongation ribosomes are susceptible to peptidyl-tRNA drop-off. Stability of these complexes depends on the nascent peptide interacting with the ribosomal tunnel (Chadani et al. EMBO J 2021), and it dramatically increases after 6 amino acids are incorporated (Heurgué-Hamard et al. EMBO J 2000), and the date of drop-off is heavily influenced by the nature of the tRNA / tRNA isoacceptor (Dinçbas et al. JMB 1999) and peptide (Chadani et al. EMBO J 2021, Chadani et al. Mol Cell. 2017). The process of drop-off is kinetically competing with translation termination, and bacterial termination factor RF3 plays a crucial role in this (Zaher and Green, Cell 2011), with mismatched P-site tRNA stimulating premature peptide release (Zaher and Green, Nature 2009). Studying the process of peptidyl-tRNA drop-off in the cell on the global scale is challenging due to the lack of appropriate techniques that would allow global detection and quantification cellular peptidyl-tRNA species.

The manuscript by Nagao and colleagues is a real tour de force, both technologically and in terms of the amount of performed experiments. Importantly, the authors set up, validate and apply an MS-based approach for global detection and quantification cellular peptidyl-tRNA species. I believe this

method has a great potential for discovering new biology in the future, e.g. for identifying new ORFs that generate high levels of drop-off similarly to lambda-phage-encoded bar minigenes that mediate translational shutoff (Ontiveros et al. JMB 1997). I have only minor comments / suggestions.

Response: We really appreciate your high evaluation of our work and many positive words that encourage us.

1. P. 7: When it comes to detection of longer peptidyl-tRNA species, maybe the authors could more explicitly spell out what part of the signal (i.e. lack of it) is that these are not as abundant as the less stably ribosome-associated short peptidyl-tRNA species and what part is due to potential complications associated with detection of these species by mass spectrometry.

Response: Thank you for the suggestion. We revised the text to explain some technical difficulties of mass spec in the detection of longer pep-tRNA compared to aa- or dipep-tRNAs.

2. P. 8: Regarding the effect of macrolide tylosin on drop-off: biochemical evidence suggests that – at least in some cases – polypeptides longer than 2-3 amino acids are made (see (Yakhnin et al. mBio 2019)). Could you please comment on that / refine the statement?

Response: Thank you for commenting on this. As the reviewer pointed out, it is known that certain nascent peptides can bypass the macrolides bound to the exit tunnel. In fact, Tyr-dependent ribosome stalling induces the expression of rRNA methyltransferase responsible for macrolide resistance. In this revision, we mentioned this important fact with appropriate references.

3. P. 14: Maybe one could use plating assays in the presence of sub-MIC concentration of miscoding-inducing antibiotics to bridge the data shown on Figure 3 with the section 'Misdecoding induces the pep-tRNA drop-off'?

Response: Thank you for this suggestion. It is an important experiment showing the relationship between miscoding and pep-tRNA drop-off. Previously studies by John Menninger's group reported that streptomycin, a miscoding-inducing antibiotic,

decreases viability of *pth*^{ts} cells at non-permissive temperature with accumulation of pep-tRNAs (Caplan and Menninger, 1984), indicating that miscoding induces pep-tRNA drop-off. We added this information in introduction and discussion.

4. P. 16: When discussing frequencies of drop-off as a function of peptide composition, maybe one could connect to earlier biochemical data better.

Response: Thank you for this suggestion. We've been trying to provide a logical explanation for the drop-off efficiency with peptide composition. In earlier studies, Mans Ehrenberg's group compared drop-off efficiencies of a couple of dipep-tRNAs *in vitro*, and reported that fMet-Phe and fMet-Lys have rapid dissociation rate versus fMet-Ile (Dincbas et al., JMB, 1999). This is consistent with our data showing that Met-Phe and Met-Lys were detected more abundant than Met-Ile (Figure 2d). We put this information in the 2nd section of the result. However, we don't find any biochemical data to measure drop-off efficiencies of longer pep-tRNAs.

Reviewer #4 (Remarks to the Author):

This work from Tom Suzuki's group systematically investigates how peptidyl-tRNA drop-off occurs (from the translating ribosomes) in Escherichia coli, and proposes the phenomenon to be a ribosomal/cellular mechanism of quality control operating during the early stages of translation elongation. The initial discovery of peptidyl-tRNA hydrolase (Pth) by the groups of Chapeville and RajBhandary in the late 60s set the stage for the investigations on the role of Pth (an essential enzyme) in translation. The genetic studies of peptidyl-tRNA drop-off were then greatly facilitated by the isolation of temperature sensitive mutants in peptidyl-tRNA hydrolase (Pth) by the Menninger group in early 70s. While many other groups have since then contributed to the understanding of the role of RRF, RF3 and other components of the translation machinery in peptidyl-tRNA drop-off, a deep understanding of the mechanism of peptidyl-tRNA drop-off from the ribosome remained wanting.

Response: Thank you very much for letting us know the earlier studies on Pth. We added some original papers in introduction.

This investigation begins with the validation of the earlier key genetic findings of

the peptidyl-tRNA drop-off showing the role of RRF and RF3 in peptidyl-tRNA drop-off and the crucial role played by Pth in recycling of tRNAs. A deficiency of Pth quickly depletes aminoacyl-tRNA pool to accumulate as peptidyl-tRNAs. Majority of the peptidyl-tRNAs have short peptides of 2 to 5 residue length but the ones having 15 residues were also detected. RRF deficiency in the *frr* ts strains reveals ribosome stalling in the early stages of elongation (and thus a role of RRF in fidelity of translation soon after initiation). The authors have developed robust mass spectrometry (MS) pipeline to sequence the peptides attached to tRNAs. Importantly, the authors have also optimised their method of separating tRNAs by reciprocal circulating chromatography (RCC) to now separate the tRNAs with peptides attached to them (into 42 species). Given that the ester bond that connects the peptides with the tRNAs is highly susceptible to heat/pH, this is a rather challenging task, and is certainly a 'work of art' in science! And, it is these achievements of the authors that have allowed them to sequence the variety of the peptides attached to each of the individual tRNA species and map them to the starting regions of the open reading frames (ORF) in *E. coli* genome to decipher the mechanistic details peptidyl-tRNA drop-off. The authors categorised the dropped-off peptidyl-tRNAs into the cognate and noncognate categories. The cognate ones do not show any mismatches with the ORFs of the genes. However, the noncognate ones show single mismatches at 0, -1, -2, or -3 positions (C0X, C-1X, C-2X and C-3X, respectively). Weak/mismatched pairing between codon/anticodon appears to be a major reason for peptidyl-tRNA drop-off. To validate the hypothesis, the authors first deleted, from the *E. coli* genome, some of the genes (*lpp*, *rplJ*) predicted from the MS analysis of the dropped-off peptidyl-tRNAs to show that the corresponding peptidyl-tRNAs were no longer present in the pool. The authors then designed a clever reporter system having a novel sequence (not found in any other ORF in *E. coli*) in the early part of the ORF and exploited the property of the elongator tRNA(Met) to occasionally decode Ile codon (AUA). Not satisfied with just this, the authors then deleted *tmcA* gene encoding tRNA acetyltransferase that modifies the C34 in the anticodon to N4-acetylcytidine (ac4C). The modification is important in preventing misreading of AUA codon as Met. A deficiency of TmcA led to increased misreading of AUA in the reporter (and drop-off of the peptidyl-tRNAs of C0X and C-1X class). The reporter has allowed the authors to carry out a good quantitative analysis.

The authors have made several other important finds. For example, based on the observation that the initiator Met (fMet) is missing in most of the peptidyl-tRNAs the authors suggest that peptide deformylase (PDF) and Met aminopeptidase (MAP) work even on the free peptidyl-tRNAs having as short as tripeptide chain. The study also allows the authors to comment on the species of aminoacyl-tRNAs that deplete fast upon Pth deficiency.

Overall, I find that the present work is extensive, thorough and the manuscript is well written for the most parts. The experiments have been designed in a competent manner and demonstrate authors' indulgence to uncover finer details of the phenomenon. I have following points for the authors to clarify/consider.

Response: We really appreciate deep understanding of our work, and many positive words that encourage us.

1. In the abstract, delete, "we happened to find" and rephrase the statement (I am sure this was not a serendipitous finding).

Response: As suggested, "we happened to find" has been changed to "we found".

2. On page 4, last line: change 'enables' to 'enabled'.

Response: Corrected.

3. The latter half of the last long paragraph of Introduction section is pretty much a reproduction of the Abstract of the manuscript. This repetition is unnecessary and should be replaced with just one or two statements.

Response: Thank you for pointing out. According to "Formatting Instruction" of the journal, the final paragraph of Introduction must contain a brief summary of the major results and conclusions of the current work, written in the present tense. We revised this part as concisely as possible to reduce the redundancy.

4. In many places in the Results section the authors give an impression that the Pth ts strain was grown at 43 °C, which is a nonpermissive temperature (for example, see the last paragraph on page 6; and then again in the next

paragraph on page 7). The authors must have grown the culture at the permissive temperature and then shifted the culture to the nonpermissive temperature. This should be carefully worded.

Response: Thank you for your suggestion. We rewrote the relevant expressions in the text pointed here using “incubated” and “shifting to”.

5. On page 9, second paragraph, last statement: the authors state ‘our’ hypothesis. The models for drop-off of peptidyl-tRNAs have been proposed earlier (Singh et al.; doi: 10.1016/j.jmb.2008.05.033).

Response: As pointed out, this statement was misleading. We revised this sentence as follows, “These findings fully support our speculation that ribosome accumulate near the initiation sites upon inactivation of RRF, suggesting that RRF actively dissociates pep-tRNAs and recycles ribosomes at the early elongation stage.”

6. Likewise, the role of RRF in the fidelity of translation has been proposed on the basis of genetic analyses in the above referred paper and also in another paper (Seshadri et al.; doi:10.1111/j.1365-2958.2009.06685.x)

Response: As suggested, we appropriately cited relevant papers in discussion.

7. On page 11, penultimate paragraph, last statement: change, ‘lied’ to ‘laid’.

Response: Corrected.

8. On page 13, second paragraph, first line: change, ‘them’ to ‘then’ (there is at least one more place in the manuscript where a similar correction is required).

Response: Thank you very much for finding typos. We carefully checked the manuscript and corrected them.

9. Page 16, first paragraph: may be better to say, ‘growth’ rather than ‘living’.

Response: Corrected.

10. Page 17, second paragraph: 'deacylated-tRNA' not 'deacyl-tRNA'

Response: Corrected.

11. Page 19, second paragraph: 'avoid' not 'loosen'

Response: Corrected.

12. While the authors have appropriately discussed shifting of the peptidyl-tRNA into the P/E (and p/R) site for drop-off, the authors may also discuss on the recruitment of incorrect aminoacyl-tRNA in the A site to begin with, and the role of uS12 in this. A couple of recent studies (Datta et al. doi: 10.1111/mmi.14861; doi: 10.1111/mmi.14675) have shown genetic interaction between uS12 and RRF and have also proposed a model that the authors would find relevant.

Response: This is very important information. It is intriguing for us that uS12 mutants have genetic interactions with RRF and PTH, strongly indicating quality control mechanism of translation mediated by pep-tRNA drop-off. By citing this paper, we added this information in discussion.

13. The authors do not comment anything on the effect of proline on the -1 or 0 site in peptidyl-tRNA drop-off. This may be discussed.

Response: It is known that Pro is a poor substrate as donor and acceptor for peptide bond formation, leading to ribosome stalling. We also thought that Pro codons might be involved in pep-tRNA drop-off. However, we don't see any bias on Pro codons compared to other codons in terms of drop-off efficiency.

14. In Table S1, for the non-cognate peptides (of COX type) of *higA*, *rplO*, and *sspA* the last amino acid is the same as coded for by the original sequence (ecocyc.org). The reviewer has not checked all of the sequences, and the authors should look at it once again. While this is unlikely to change any of the major findings, the data in the Table need to be reviewed carefully.

Response: Thank you very much for finding the critical errors. There are some errors in

Table S1 as pointed out. We carefully checked Table S1 and corrected errors, and also revised associated Figures.

15. Both the Figs. 3a and 3e show growth of pthts strain at 39 °C. The dilutions are similar in both and the pictures are taken after 24 h. Why then the pthts strain is growing better in 3e than 3a? If the growth times are different, they need to be corrected.

Response: As pointed out, Fig. 3a and Fig. 3e have different cultivation times. Fig. 3e is a plate cultured longer than that of Fig. 3a, because we wanted to compare different growth phenotypes between *frr^{ts}* and *pth^{ts}* strains especially at 37°C and 39°C with prolonged cultivation time. We put this information in the legend of the figure.

Reviewer #5 (Remarks to the Author):

In this paper, the authors attempted to reveal a new quality control mechanism for protein synthesis via the dissociation of peptide-linked tRNAs from the ribosome at the early stage of translation elongation. The authors established a highly sensitive method for direct profiling of nascent peptide-binding tRNAs using mass spectrometry, and identified thousands of peptides from those accumulated in *E. coli* pthts. They found that one-third of the peptides were miscoded peptides, and most of them had the wrong amino acid residues introduced by mismatched tRNAs at their C-termini. The authors have also confirmed the validity of this observation using a reporter construct.

Response: We really appreciate deep understanding our studies.

Although the authors' method for detecting peptide-linked tRNAs is novel and impressive, I believe that this paper is not appropriate for *Nat Commun* for the following three reasons.

1. The novelty of this paper is low.

The following facts, which constitute the important framework of this paper, have already been reported:

(i) In *E. coli*, the error frequency of codons close to the initiation codon has been

found to be significantly low (Parker J. Biochem Biophys Res Commun. 1984; Precup J. Mol Gen Genet. 1989), probably due to the rapid dissociation of erroneous short peptidyl tRNAs are easily dissociated before or during translation (Rodnina MV. Annu Rev Biochem. 2001).

Response: Please let us confirm this point. Parker et al. (BBRC, 1984) just measured basal level codon misreading in *E. coli*, and did not show low error rate near initiation. However, Parker and Precup (MGG, 1986) and Precup et al. (MGG, 1989) reported that low level Phe-to-Leu mistranslation at 3rd codon, but high level at 8th codon of *E. coli argI* gene upon Phe starvation. These reports are the first indication of high translational accuracy in the beginning of ORFs in *E. coli* cultured under starvation of specific amino acids. Then, Rodnina et al speculated in their review paper (Annu Rev Biochem. 2001) that erroneous short peptidyl-tRNAs might be dissociated to maintain the translational fidelity. However, this is just a mere speculation among several other possible mechanisms. Any experimental evidence has never been provided so far. In the present manuscript, we report for the first time that the drop-off is an active mechanism to reject miscoded pep-tRNAs in the early elongation under normal growth condition (not starved conditions). Anyway, we clearly mentioned early studies by citing these papers appropriately.

(ii) Codon-anticodon mismatch on the ribosome of mRNA and tRNA causes peptide drop-off (Zaher HS. Nature. 2009).

Response: This Green lab's paper (Zaher et al. Nature. 2009) does not describe anything about drop-off of miscoded pep-tRNAs, but biochemically showed that RF2 recognizes miscoded pep-tRNA at P-site and induces hydrolysis of peptide moiety from pep-tRNA. In the supplemental data, they measured non-enzymatic dissociation of such miscoded pep-tRNA from ribosome, and found it was slow enough to test the RF2-mediated peptide release from the pep-tRNA. This is a quite different from our observation.

(iii) It has already been shown that RF3 prematurely terminates peptides with mistakes during protein synthesis on bacterial ribosomes (Zaher HS. Cell. 2011).

The authors have proved the same facts by using MS, but there is nothing novel in these. Also, these reports are cited in the text, but the reviewer could not

understand what novelty this paper added to this field in addition to the previous reports.

Response: This paper (Zaher et al., Cell 2011) showed that RF3 promotes RF2-mediated peptide release from the longer pep-tRNA which is stalled in the middle stage of translation. In this case, the role of RF3 is completely different from that in pep-tRNA drop-off. As we wrote in our manuscript, involvement of RF3 in pep-tRNA drop off has been proposed by biochemical and genetic studies (Ref 43-46). We clearly demonstrated functional role of RF3 by directly analyzing cellular pep-tRNAs by our MS analysis. This experiment does not aim to claim the novelty of our study, but to show the reliability and performance of our experimental setup. In fact, other reviewers highly appreciate this experiment.

2. Strange handling of peptides

After purifying pep-tRNA from *E. coli*, the authors recovered the peptide by treating it with 0.3% ammonia (according to my calculations, this is a solution of about pH 11) at 50°C for 30 minutes. Under these conditions, ammonia-induced beta-elimination must be occurring. Cysteine is the most susceptible to beta-elimination, and data supporting the occurrence of beta-elimination in the peptides can be found in Table S1. The peptides listed in Table S1 have a total of 5115 amino acid residues, of which only three are cysteine. Since proteins usually contain one cysteine residue for every 200 amino acid residues, the proportion shown in Table S1 is very small. Therefore, this is evidence that beta-elimination is occurring. The residues after beta-elimination are subjected to addition reactions by various substances in the solution. Therefore, these peptides would either be misidentified as other sequences or recognized as non-cognate peptides. Under these conditions, in addition to cysteine, serine and threonine may have also been subjected to beta-elimination as well. The resulting peptide sequences are likely to contain many errors and serious biases. It would be inappropriate to make conclusions based on these peptide sequences.

In order to prove that there is little mistake or bias in the peptide sequence, a control experiment is needed to show that there is no beta-elimination in many peptides under these conditions. Alternatively, more mild methods could be applied, such as the use of peptidyl-tRNA hydrolase to cleave peptides from pep-tRNA or the use of nuclease P1 to analyze the sequence of peptides that

still have Ado bound. The reviewer believes that the authors should conduct such experiments and prepare a paper based on the new data.

Response: This reviewer concerns about β -elimination of the alkylated Cys (here shown as Cys*) of the peptides during alkaline treatment for deacylation of pep-tRNAs (Figure D). To test this possibility, we treated tryptic digests of alkylated BSA with 0.3% ammonia water at 50°C for 30 min (the same condition of deacylation), followed by LC/MS analysis to compare abundance of the Cys*-containing peptides with and without

alkaline treatment (Figure E). If β -elimination occurs during alkaline treatment, relative abundance of the Cys*-

containing peptides would be reduced, because Cys* is converted to dihydroalanine. However, we found little change in the relative abundance of the Cys*-containing peptides (Figure E), and failed to detect any dihydroalanine-containing peptides after the alkaline treatment. As suggested, we also checked all peptides containing Thr and Ser (Figure E), and found no indication of the β -elimination reaction.

We next investigated why small numbers of Cys-containing peptides were detected in our analysis. We noticed that Cys codons are less abundant (0.6%) compared to the other codons in the beginning of highly expressed genes in *E. coli* (2-10 codon position). This is one reason why we detected small numbers of Cys-containing peptides in our analyses. The other reason would be a chemically unstable nature of Cys thiol, as the thiol group is easily oxidized to take several

derivatives bearing different oxidation numbers, leading to low detection sensitivity. In this revision, we reanalyzed the nascent peptides derived from pep-tRNA^{Cys} by mass spectrometry and identified 10 additional Cys*-containing peptides that are assigned to *E. coli* ORF. We revised Table S1 by adding these peptides to the list, and revised the related text and figures.

This reviewer also suspects our peptide assignment. We are confident of assigning the peptides based on high quality CID spectra with accurate molecular mass of product ions.

The authors focused their analysis on acetylated pep-tRNA. In order to acetylate pep-tRNA in their method, the formyl group of newly synthesized pep-tRNA must be removed by the peptide deformylase after dissociation from the ribosome, and the amino group must be exposed at the peptide amino terminus. This amino group-exposed pep-tRNA should generally be an unstable material that exists only for a very short time in the course of its degradation. The reason is that these are intermediates that are generated during the degradation process after dissociation from the ribosome, in which the formyl group and methionine are removed by peptide deformylase and methionine aminopeptidase, respectively. In general, reproducibility is required for experiments on unstable substances. Therefore, all of the peptide sequences submitted as data must be either validated by multiple similar analyses, or by sequence analysis of samples of *E. coli* dysfunctional for peptide deformylase by actinonin inhibition, or by obtaining similar peptide sequences in the absence of acetylation.

Response: This reviewer concerns about reproducibility of pep-tRNA analysis, because amino group-exposed pep-tRNA should generally be unstable. We carefully extract aa-tRNAs and pep-tRNAs from cells by acidic phenol under low temperature. This is a well-established method to analyze aa-tRNAs in the cell. Pep-tRNAs are much more stable than aa-tRNAs, because the amino group of pep-tRNA is far from its acyl bond, when compared to aa-tRNAs. All data are reproducible and reliable.

3. We don't know if the same phenomenon occurs in wild-type *E. coli*. In this paper, the authors performed almost all of their experiments using mutants of the *pthts* as a background strain. Therefore, it remains to be verified whether the same happens in wild-type *E. coli*. The scale of the data obtained from wild-

type *E. coli* may be smaller than that obtained from *pthts* strains, but its pep-tRNA data is necessary, and its principle must be the same as that given by *pthts* strains. We need to know about universal phenomena, not *pthts* strain-specific phenomena. If such observations cannot be made, then mass spectrometry is still not sensitive enough to be used for analysis in this field.

Response: This reviewer asked us to verify whether the same phenomena happen in wild-type *E. coli*. Historically, all analyses of pep-tRNA drop-off have been performed using *E. coli pth^{ts}* strain incubated at 43°C for 30 min, because Pth activity is too strong to detect pep-tRNAs in WT cells. The *pth^{ts}* strain has a *pth* gene with only single Gly-to-Asp mutation at position 101. This is not a special strain among other numerous *E. coli* strains. In addition, we actually detected pep-tRNAs in the *pth^{ts}* strain cultured even at 37°C, strongly suggesting that pep-tRNA drop-off is not an accidental event that arises under heat stress, but rather an inevitable event that takes place frequently in any *E. coli* strains cultured under normal growth conditions.

Others

What is the origin of aa-tRNA? The authors have analyzed aa-tRNA as well as pep-tRNA (p6, line 11-). Is this aa-tRNA observing a newly aminoacylated tRNA, or is it observing a substance in the process of degradation of pep-tRNA? There is no mention of this, and it was difficult to read the paper because I could not predict where the story would go at the beginning of the paper. The reviewer thinks that this should be clearly stated in the paper.

Response: We appreciate this important question. Regarding the origin of aa-tRNAs in *E. coli pth^{ts}* strain, there are two possibilities: (1) aa-tRNAs newly synthesized by aminoacyl-tRNA synthetases (aaRSs), or (2) aa-tRNAs produced by degradation of pep-tRNAs which were dissociated from ribosome. We firmly believe that most of aa-tRNAs in this strain are newly synthesized by aaRSs, because most aa-tRNAs are rapidly reduced upon Pth inactivation (Figure 1g). If aa-tRNAs are mainly produced from pep-tRNAs accumulated in the cell, this phenomenon would not be observed. In addition, based on the average drop-off rate of pep-tRNAs (10.5%), deacylated tRNAs would be produced much more than pep-tRNAs dissociated from the ribosome during translation. Most of aa-tRNAs should be produced by aminoacylation of deacylated tRNAs ejected from the ribosome.

REVIEWER COMMENTS

Reviewer #1 (Remarks to the Author):

The authors have responded thoughtfully to my questions and suggestions raised in the first round of review and further strengthened the presentation of their story. I have no additional comments.

Reviewer #2 (Remarks to the Author):

From my point of view, the authors have satisfactorily addressed the reviewers' concerns and have appropriately amended the manuscript. However, I would encourage the authors to consider to further shorten the discussion. Other than that, I recommend the work to be published in Nature Communications.

Here are a few edits

- The last sentence of the abstract is extremely long. Authors should consider splitting into two (or three?) sentences.

-The newly added sentence in page 4 "More recently, in the study examining..." should be edited for clarity

-First sentence in page 5: "... N-terminal regions of template mRNAs." mRNAs do not have N-terminal regions.

-Newly added paragraph in page 11: "...tRNA read the correspond codon." Change to corresponding.

Reviewer #5 (Remarks to the Author):

This revised manuscript is much improved over the previous draft. The authors appear to have satisfactorily addressed some of the previous reviewers' comments. The reviewer is very appreciative. However, several issues remain, as described below.

1. The novelty....

The authors' arguments were understandable. However, the high similarity between what is stated in the second paragraph on p3 and what the authors are trying to prove makes it difficult to distinguish to the average reader of this journal. The reviewer would prefer a more resolved introduction.

2. Strange....

Thank you for answering my questions with additional experiments. The reviewer agreed that the conditions used by the authors do not cause beta-elimination in the side chain of cysteine derivative, Ser and Thr. Nevertheless, this condition is still not very common in proteomics. Although irrelevant to this paper, the bonds between Asn/Asp-Gly and Asn/Asp-Pro are easily cleaved under basic conditions and at high temperatures, and proteomics researchers would be hesitant to use this condition. The reviewer would appreciate a brief summary of the results of this experiment to be added to Method.

With regard to reproducibility, the authors claim to be confident. While careful extraction with acid phenol is certainly necessary, it does not ensure actual reproducibility; the experimental proof is needed to confirm that the analytical system is reproducible by repeating the global experiment such as the one shown in Figure 4. The reviewer also note the following comment in the previous section. "This experiment does not aim to claim the novelty of our study, but to show the reliability and

performance of our experimental setup." The actual proof of reproducibility would be necessary to demonstrate the reliability and performance of this experimental setup.

The authors use Q sepharose to purify peptides from tRNAs. Negatively charged peptides may bind to quaternary amine residues of the resin. In addition, highly hydrophobic peptides such as signal peptides should cause nonspecific adsorption to the resin. The reviewer anticipates that this approach will make it difficult to recover peptides without bias. What methods do the authors use to avoid the bias in peptide types caused by the Q sepharose separation? In particular, the reviewer would like to hear comments on the experiment in Figure 4.

3. We don't know....

The reviewers agreed the authors' comment that the many experiments showing drop-off have been done with pthts strains and that there is little drop-off in wild strains. However, the reviewer believes that pthts is a special strain of E. coli. Of course, generalizations can be made if several different strains were studied, including the special strain, but all of the authors' experiments were with strains with the pthts background. The reviewer recommends that this claim be toned down, as the data presented are insufficient to argue that this occurs frequently in E. coli cultured under normal growth conditions.

4. The reviewer understands that the peptides do not have beta de-elimination, and would like to make a new comment below.

On p6, third paragraph.

The authors have normalized the quantitative data from many experiments to the level of f-Met-Ado. The reviewers are concerned about this. This is because it is only an assumption that there would be no effect of the pth knockout strain, since this is not a substrate of pth. The authors do not comment on changes in f-Met-Ado levels that depend on changes in growth temperature or other conditions. The reviewer recommends that the authors measure the actual cellular content of f-Met-Ado.

On p7, third paragraph.

While the authors begin the paragraph by stating that "all nascent peptides of the pep-tRNAs detected in this study were acetylated", they also state that "a majority of the pep-tRNAs detected in this study were acetylated at the N-terminus" at the end of the paragraph. Please correct this discrepancy. In addition, all dipeptides other than Met-Arg-Ado should also have been acetylated at the N-terminus by actinonin. Please add a note on this point.

On p10, first paragraph.

The authors note here that " We successfully detected and profiled the nascent peptides of the pep-tRNAs obtained from the pthts strain (Figure 4b), which clearly accumulated over time after shifting to 43°C." Please provide a supplementary table for the identification of these peptides in addition to the figures. The abstract states that 2,700 nascent peptides and 900 species of miscoded peptides were detected. These are thought to be the 3852 species of peptides described here. The representative experimental facts in the abstract need to be presented with sufficient transparency through data. The table should show the sequence, peptide score, pepExpect score, Δ peptide mass, etc (<https://www.mcponline.org/mass-spec-guidelines>).

Currently, it only shows that the mass error is less than 5ppm, which is insufficient. This alone does not tell us whether the "peak" shown here is a peptide or some other chemical with the same elemental composition. Even if it is a peptide, we do not know if the order of the amino acids is correct. Even if there is a flaw in the purification or identification method of this peptide, or even if there is a bias because of it, the reader cannot verify it. Again, the reviewer has a strong opinion that the data should include an identification table of the 3852 peptides.

On p12, second paragraph.

Are all 713 peptides identified here included in the 3852 peptides shown on p10? How much overlap is there? The reviewer would like to ask in terms of the robustness and reproducibility of this method.

On p13, first paragraph.

The author states that "the miscoding event induces dissociation of non-cognate pep-tRNAs from ribosomes. It is difficult to conclude that the drop-off event is the cause of the miscoding by the data presented in this paragraph. Could it simply be that the drop-off event is more likely to occur immediately after the miscoding event?"

On p13, second paragraph.

The "the cumulative fraction" should be calculated by summing the number of molecules of each peptide. The reviewer does not understand what the sum of the number of peptide types dropped-off from the ribosome indicates.

In Figure 5h, the correlation appears to be greatly reduced if the AAA codon is excluded from the plot. Pseudo-correlation is suspected. Please comment on this point to the authors.

On p15, second paragraph.

In the WT in Figure 5C, the cognate peptides of MWGI and MWGIV are drop-off in larger amounts than the noncognate peptides of MWGM and MWGMV. To conclude that this is inducible drop-off, the authors need to compare the drop-off rates ($=\text{drop-off}/(\text{drop-off}+\text{read-through})$) between cognate and non-cognate. Since the read-through data are not presented in this paper, the authors will need to measure read-through or build on existing data and reports. Authors are encouraged to discuss drop-off in pthts and $\Delta\text{tmcA}/\text{pthts}$ cells by this drop-off rate. The constitution of tetrapeptides and pentapeptides is not a surrogate indicator.

The peptides in Table S1 that contain Y, F, or W at positions other than the C-terminus seem to be less likely to be non-cognate. The reviewer recommends that the authors analyze the sequence of drop-off peptides. These pep-tRNAs are probably sorted by the same mechanism as the peptide bulkiness described by Chadani et al (Chadani et al, 2021).

5. Others

The "normal growth conditions" and "normal culture conditions" indicated in the manuscript needs to be defined. In general, one would consider a strain cultured in a rich medium at 37 degrees Celsius. Please define it clearly or use different words for the reader to understand.

First, we thank the reviewers for their careful and helpful review of our manuscript, and for providing valuable suggestions for its improvement. Our point-by-point responses to each of the reviewers' comments are shown below. Changes to the main text are marked in light blue.

Response to Reviewer #2's comments

From my point of view, the authors have satisfactorily addressed the reviewers' concerns and have appropriately amended the manuscript. However, I would encourage the authors to consider to further shorten the discussion. Other than that, I recommend the work to be published in Nature Communications.

We really appreciate the reviewer's constructive comments and recommendation for publication. As suggested, we have removed a few sentences from the discussion.

Here are a few edits

- The last sentence of the abstract is extremely long. Authors should consider splitting into two (or three?) sentences.

Thank you for the comment. We split the last part of the abstract into two sentences.

-The newly added sentence in page 4 "More recently, in the study examining..." should be edited for clarity

We edited it for clarity.

-First sentence in page 5: "... N-terminal regions of template mRNAs." mRNAs do not have N-terminal regions.

Thank you for pointing it out. It has been corrected to "N-terminal regions of template ORFs".

-Newly added paragraph in page 11: "...tRNA read the correspond codon." Change to corresponding.

Thank you for letting us know it. It was corrected.

Response to Reviewer #5's comments

This revised manuscript is much improved over the previous draft. The authors appear to have satisfactorily addressed some of the previous reviewers' comments. The reviewer is very appreciative.

We appreciate the positive comments.

However, several issues remain, as described below.

1. The novelty....

The authors' arguments were understandable. However, the high similarity between what is stated in the second paragraph on p3 and what the authors are trying to prove makes it difficult to distinguish to the average reader of this journal. The reviewer would prefer a more resolved introduction.

We appreciate this reviewer's constructive comments. We tried to improve the introduction that is readable for non-specialists.

2. Strange....

Thank you for answering my questions with additional experiments. The reviewer agreed that the conditions used by the authors do not cause beta-elimination in the side chain of cysteine derivative, Ser and Thr. Nevertheless, this condition is still not very common in proteomics. Although irrelevant to this paper, the bonds between Asn/Asp-Gly and Asn/Asp-Pro are easily cleaved under basic conditions and at high temperatures, and proteomics researchers would be hesitant to use this condition. The reviewer would appreciate a brief summary of the results of this experiment to be added to Method.

Thank you very much for evaluating our additional data and suggestion for explanation of our results. The brief summary of the results of this experiment has been added to the Method section.

With regard to reproducibility, the authors claim to be confident. While careful extraction with acid phenol is certainly necessary, it does not ensure actual reproducibility; the experimental proof is needed to confirm that the analytical system is reproducible by repeating the global experiment such as the one shown in Figure 4. The reviewer also note the following comment in the previous section. "This experiment does not aim to claim the novelty of our study, but to show the reliability and performance of our experimental setup." The actual proof of reproducibility would be necessary to demonstrate the reliability and performance of this experimental setup.

To verify the reproducibility of the pep-tRNA profiling in Fig. 4, we prepared pep-tRNAs from three independent samples of *pth*^{ts} strain incubated at 43°C for 30 min, and analyzed the nascent peptides by our new LC/MS set up (Figure. I). In this new analysis, we have detected much more peptides than those in the previous analysis; 4,413-4,666 cognate peptides derived from N-terminal regions of *E. coli* ORFs and 1,121-1,218 non-cognate peptides with single amino acid substitution. Three biological replicates showed excellent reproducibility with our new LC/MS setup. The median values of molecular weights for the cognate and non-cognate peptides were 750.9 ± 6.4 and 646.0 ± 9.5 , respectively, suggesting that non-cognate peptides with single substitution tend to have a smaller molecular weight than the cognate peptides. We have replaced Figure 4 with our new data set, and added triplicate data set in new Figure S5.

The authors use Q sepharose to purify peptides from tRNAs. Negatively charged peptides may bind to quaternary amine residues of the resin. In addition, highly hydrophobic peptides such as signal peptides should cause nonspecific adsorption to the resin. The reviewer anticipates that this approach will make it difficult to recover peptides without bias. What methods do the authors use to avoid the bias in peptide types caused by the Q sepharose separation? In particular, the reviewer would like to hear comments on the experiment in Figure 4.

Regarding potential bias in peptide types caused by the Q sepharose separation, in this revision, we separated peptides from tRNAs by Q sepharose using a buffer containing 250 mM NaCl to obtain negatively-charged peptides as much as possible. The subsequent desalting of the peptides was performed using the C18 tip, which is commonly used in proteome analysis. Of course, highly hydrophobic peptides might be lost by non-specific binding to any resin. But, Sepharose will be much better than C18 in general. Detailed procedure for

our sample preparation has been described in Method section.

3. We don't know....

The reviewers agreed the authors' comment that the many experiments showing drop-off have been done with pthts strains and that there is little drop-off in wild strains. However, the reviewer believes that pthts is a special strain of *E. coli*. Of course, generalizations can be made if several different strains were studied, including the special strain, but all of the authors' experiments were with strains with the pthts background. The reviewer recommends that this claim be toned down, as the data presented are insufficient to argue that this occurs frequently in *E. coli* cultured under normal growth conditions.

As suggested, we toned down the sentences and wording in the DISCUSSION regarding the claim that pep-tRNA drop-off occurs frequently under normal growth conditions. Please understand that the drop-off frequency is not affected by Pth mutation.

4. The reviewer understands that the peptides do not have beta de-elimination, and would like to make a new comment below.

On p6, third paragraph.

The authors have normalized the quantitative data from many experiments to the level of f-Met-Ado. The reviewers are concerned about this. This is because it is only an assumption that there would be no effect of the pth knockout strain, since this is not a substrate of pth. The authors do not comment on changes in f-Met-Ado levels that depend on changes in growth temperature or other conditions. The reviewer recommends that the authors measure the actual cellular content of f-Met-Ado.

Thank you for the comment. We also realized importance of measuring cellular level of f-Met-Ado. As shown in **Figure II**, the peak intensities of f-Met-Ado, Ac-Ala-Ado and Ac-Met-Ala-Ado are normalized by those of dihydrouridine (D) which is a ubiquitous and stable tRNA modification, because it is unlikely that this tRNA modification is affected during 30 min incubation at 43°C. As expected, Ac-Ala-Ado is drastically reduced, whereas Ac-Met-Ala-Ado is markedly accumulated. In contrast, f-Met-Ado does not change. This additional data clearly demonstrates that cellular level of f-Met-Ado is unchanged in *pth^{ts}* cells incubated at 43°C for 30 min. This new data has been added as a **new Figure S2a**. All relevant data are included in the source data file.

Figure II. Cellular level of f-Met-Ado and Ac-dipep-Ado
Relative intensity of Ac-Ala-Ado, Ac-Met-Ala-Ado and f-Met-Ado normalized by the intensity of dihydrouridine (D) in *pth^{ts}* cells at 0 min (light gray) and 30 min (gray) after shifting to 43°C.

On p7, third paragraph.

While the authors begin the paragraph by stating that "all nascent peptides of the pep-tRNAs detected in this study were acetylated", they also state that "a majority of the pep-tRNAs detected in this study were acetylated at the N-terminus" at the end of the paragraph. Please correct this discrepancy. In addition, all dipeptides other than Met-Arg-Ado should also have been acetylated at the N-terminus by actinonin. Please add a note on this point.

We have changed "all nascent peptides" to "most nascent peptides". In addition to Met-Arg-Ado, a variety of dipep-Ados are also detected as N-terminal formylated forms by actinonin treatment. As a representative, the data for Met-Arg-tRNA is shown in **Figure S2b**. We changed the annotation to "formylated-dipep-Ado species (f-Met-Arg-Ado as a representative example) appeared~".

On p10, first paragraph.

The authors note here that " We successfully detected and profiled the nascent peptides of the pep-tRNAs obtained from the *pth^{ts}* strain (Figure 4b), which clearly accumulated over time after shifting to 43°C." Please provide a supplementary table for the identification of these peptides in addition to the figures. The abstract states that 2,700 nascent peptides and 900 species of miscoded peptides were detected. These are thought to be the 3852 species of peptides described here. The representative experimental facts in the abstract need to be presented with sufficient transparency through data. The table should show the sequence, peptide score, pepExpect score, Δ peptide mass, etc (<https://www.mcponline.org/mass-spec-guidelines>).

Currently, it only shows that the mass error is less than 5ppm, which is insufficient. This alone does not tell us whether the "peak" shown here is a peptide or some other chemical with the same elemental composition. Even if it is a peptide, we do not know if the order of the amino acids is correct. Even if there is a flaw in the purification or identification method of this peptide, or even if there is a bias because of it, the reader cannot verify it. Again, the reviewer has a strong opinion that the data should include an identification table of the 3852 peptides.

We realized that our descriptions have caused some confusion for the reviewer to misunderstand our pep-tRNA profiling. As we mentioned in this result section, we just analyzed molecular mass of nascent peptides (3-14 aa) from pep-tRNAs. Due to the large variation in their sequences and molecular mass, it was difficult to determine the sequence of each peptide individually, as each was present in only a small quantity. At this stage, we did not perform CID to sequence each peptide. Based on exact molecular mass of each peptide, we were able to classify these peptides into cognate peptides and non-cognate peptides with single amino acid substitution. Therefore, we have rewritten the related sentences so that the reader would not misunderstand the pep-tRNA profiling in **Figure 4**. In addition, we revised the abstract, because the number of the detected peptides has massively increased in this revision.

To verify the pep-tRNA profiling, we here compare mass chromatograms (TIC) of nascent peptides from pep-tRNAs in *pth^{ts}* cells incubated at 43°C for 0 min and 30 min (**Figure III**). TICs in mass range covering 3-14mer peptides clearly popped up at 30 min, suggesting that most of detected molecules in this study can be assigned to nascent peptides of pep-tRNAs. As pointed out by the reviewer, we also detected a number of non-peptide molecules even at 0 min (**Figure III**). We removed those molecules as chemical noise from the mass list of the 30 min sample in this analysis. This process has been described in Methods section. Later, in **Figure 5**, we precisely sequenced each peptide by isolation of individual pep-tRNAs, and confirmed both cognate and non-cognate peptides by CID analyses.

Regarding mass error range of 5-ppm, many papers published in Molecular & Cellular Proteomics cited by the reviewer and other journals actually use mass error range of 5 ppm or higher as a threshold for peptide analysis by LC/MS or LC/MS/MS (For examples, Shen et al. Mol Cell Proteomics. 2009, Kleifeld et al. Nat Biotechnol. 2010, Chick et al. Nat Biotechnol. 2015, Zhong et al. Nat Commun. 2020, Lischnig et

al. Mol Cell Proteomics. 2022, Skowronek et al. Mol Cell Proteomics. 2022). We believe the 5-ppm mass error is an acceptable condition for our peptide analysis. As the reviewer pointed out, we added peptide score, pepExpect and Δ peptide mass of each nascent peptide attached to the isolated pep-tRNAs in this study to Table S1.

On p12, second paragraph.

Are all 713 peptides identified here included in the 3852 peptides shown on p10? How much overlap is there? The reviewer would like to ask in terms of the robustness and reproducibility of this method.

Thank you for commenting on this. Based on the results of the experiments performed in this revision, we checked that point. In this revision, we obtained new data set for pep-tRNA profiling (5631-5787 peptides). About 68% of the identified peptides were included in the mass list obtained by our new pep-tRNA profiling, whereas rest 32% of them were not included due to their low abundance. Peptide analyses of the isolated pep-tRNAs (Figure 5) are more sensitive than pep-tRNA profiling (Figure 4), because lowly abundant peptides attached to tRNAs can be concentrated by pep-tRNA isolation. In fact, we isolated each individual pep-tRNA from approximately 40 mg of total RNA, and performed peptide identification, whereas we only used 4 μ g of total RNA for pep-tRNA profiling. To identify nascent peptides obtained from individual pep-tRNAs isolated from *E. coli* cells, we chose mass spectra with sufficient intensity by data-dependent scan for CID to obtain product ions with good quality. In addition, if multiple genes are hit by MASCOT search, those peptides are removed from our list. Only peptides uniquely matched to single genes are included in our list. They are the reasons why we identified limited number of peptides in Figure 5.

On p13, first paragraph.

The author states that "the miscoding event induces dissociation of non-cognate pep-tRNAs from ribosomes. It is difficult to conclude that the drop-off event is the cause of the miscoding by the data presented in this paragraph. Could it simply be that the drop-off event is more likely to occur immediately after the miscoding event?"

Thank you for this comment. We rephrased it.

On p13, second paragraph.

The "the cumulative fraction" should be calculated by summing the number of molecules of each peptide. The reviewer does not understand what the sum of the number of peptide types dropped-off from the ribosome indicates.

Each peptide type represents a cognate, C₀X, C₋₁X, C₋₂X and C₋₃X type peptide of pep-tRNA. We classified them by the codon position at which they dropped off and counted those peptides at each codon position, and then made the cumulative curve along the codon position. By doing so, we could see a tendency for C₀X pep-tRNAs to dissociate from ribosome at codons closer to the start codon than cognate pep-tRNAs.

In Figure 5h, the correlation appears to be greatly reduced if the AAA codon is excluded from the plot. Pseudo-correlation is suspected. Please comment on this point to the authors.

Since the frequency of AAA codons is relatively high in the beginning of *E. coli* ORFs, AAA codons are more prominent than the other codons in Figure 5h. Because this analysis is not arbitrary, AAA codon should not be excluded from this analysis. This data is important in considering the mechanism of C₀X pep-tRNA drop-off.

On p15, second paragraph. In the WT in Figure 5C, the cognate peptides of MWGI and MWGIV are drop-off in larger amounts than the noncognate peptides of MWGM and MWGMV. To conclude that this is inducible drop-off, the authors need to compare the drop-off rates (=drop-off/(drop-off+read-through)) between cognate and non-cognate. Since the read-through data are not presented in this paper, the authors will need to measure read-through or build on existing data and reports. Authors are encouraged to discuss drop-off in pthts and $\Delta tmcA$ /pthts cells by this drop-off rate. The constitution of tetrapeptides and pentapeptides is not a surrogate indicator.

We appreciate this great suggestion. We calculated the drop-off rates (=drop-off/(drop-off+read-through)) of both cognate MWGI-tRNA and non-cognate MWGM-tRNA at the 4th codon of the reporter gene. Here, we used the amount of penta-pep-tRNAs dissociated at the 5th codon as read-through peptides. The results showed that non-cognate MWGM-tRNA is dissociated from the ribosome twice more frequently than the cognate MWGI-tRNA in both WT and $\Delta tmcA$ strains (Figure IV), suggesting that miscoding induces pep-tRNA drop-off. We added it as a new Figure S12b.

The peptides in Table S1 that contain Y, F, or W at positions other than the C-terminus seem to be less likely to be non-cognate. The reviewer recommends that the authors analyze the sequence of drop-off peptides. These pep-tRNAs are probably sorted by the same mechanism as the peptide bulkiness described by Chadani et al (Chadani et al, 2021).

We'd like to thank the reviewer for bringing this point to our attention. The intrinsic ribosome destabilization (termed IRD) discovered by Chadani et al. occurs by a mechanism distinct from the pep-tRNA drop-off, because ribosomes are disassembled without the action of RF3 and RRF. According to their paper (Chadani et al. 2021), aromatic amino acids, F, Y and W, in the nascent peptides counteract IRD because they stabilize 70S ribosome during translation. In this context, we compared amino acid composition of N-terminal peptides (2-10th codons) of *E. coli* ORFs with those of nascent peptides of pep-tRNAs dissociated from the ribosome (Figure V). We found that F, Y and W are specifically eliminated from the nascent peptides of pep-tRNAs, compared to amino acid composition of N-terminal

peptides (2-10th codons) of *E. coli* ORFs, indicating that aromatic nascent peptides might stabilize the elongating ribosome to prevent pep-tRNA drop-off. However, we have to consider some bias of peptide analysis by LC/MS, because those aromatic peptides might be lost in hydrophobic chromatography. We have to take it account this into our consideration. We added this point in DISCUSSION part.

5. Others

The "normal growth conditions" and "normal culture conditions" indicated in the manuscript needs to be defined. In general, one would consider a strain cultured in a rich medium at 37 degrees Celsius. Please define it clearly or use different words for the reader to understand.

Thank you for the comment. We edited the text to make it easier for readers to understand the words pointed out by the reviewers.

REVIEWER COMMENTS

Reviewer #5 (Remarks to the Author):

This revised manuscript is a significant improvement over the previous draft. The authors appear to have satisfactorily addressed some of the previous reviewers' comments. The reviewer is very grateful. However, several issues remain, as described below.

1. The novelty....

We appreciate this reviewer's constructive comments. We tried to improve the introduction that is readable for non-specialists.

-Unfortunately, the second paragraph of p3, starting with "Fidelity of translation is..." which I pointed out, did not seem to be changed.

2. Strange....

2-1. Thank you very much for evaluating our additional data and suggestion for explanation of our results. The brief summary of the results of this experiment has been added to the Method section.

-The reviewer was satisfied with your response. Thank you very much.

2-2. To verify the reproducibility of the pep-tRNA profiling in Fig. 4.....

- The reviewer recognized that the authors are claiming that these MS signals are detected as peptides solely because their mass values match those in the database. The reviewer would like to know if the evidence that these are peptides is really just the mass value. Although not stated in the text, the reviewer presumes that one of the requirements for this material to be a peptide is that these MS signals are detected as doublets 2 Da apart because they are cleaved from tRNA in 50% H₂¹⁸O and therefore contain 18^{OH} and 16^{OH} 1:1 at the carboxy terminus. If this assumption is correct, the authors should provide more details in the Methods section. The Methods section should also include the threshold for extracting a significant MS signal from background noise.

In the opinion of the reviewers, the above is not sufficient. At least three additional requirements must be met. The first is to show a control experiment in which the bare peptide is shown to be free of OH substitutions under the same conditions. The second is to measure MS/MS for some very strong signals and identify them as peptides. Third, it would also be necessary to create a decoy database and measure its FDR for detection on MS signals only. Appropriate FDR values would also need to be set. The reviewer suggests that these should be described in the Methods and Results sections to maintain transparency. The reviewer believes that once all these issues have been addressed, the reproducibility of the pep-tRNA profiling should be discussed.

2-3. Regarding potential bias in peptide types caused by the Q Sepharose separation.....

- The authors are inconsistent in stating that the recovery of peptides on Sepharose is better than that on C18, while enriching with C18 tips. The authors could not deny the bias, so they should accept the bias of the detected peptides. After accepting it, for example, the authors should present the hydrophobicity and pI ranges of the detectable peptides as a limitation of this method. This is because these physicochemical enrichment and selection methods depend on the hydrophobicity and charge of the peptides. The reviewer recommends that the authors discuss in the text which peptides would be detectable by comparing the hydrophobicity and pI of the peptides in the N-terminal sequence database with those actually detected.

3. As suggested, we toned down the sentences and wording in the DISCUSSION regarding the claim that pep-tRNA drop-off occurs frequently under normal growth conditions. Please understand that the drop-off frequency is not affected by Pth mutation.

- The reviewer thanks the authors for softening the wording in the DISCUSSION. As the authors said, this paper is a kind of methodological paper. Therefore, it is important to inform the reader about the limitation. In this case, the author has clearly presented the detection limits of this method in their

previous responses. The reviewer suggests that the authors should include in the text as a limitation of this method that drop-off peptides cannot be detected in wild-type E. coli.

4. The reviewer understands that the peptides do not have beta.....

4-1. Thank you for the comment. We also realized importance of measuring cellular level of f-Met-Ado. As shown in Figure II.....

- The reviewer is grateful for the new data processing. In this case, the literature showing that dihydrouridine is constant must be mentioned. The authors should provide references either in the figure legend or where appropriate.

4-2. We have changed "all nascent peptides" to "most nascent peptides".....

- Thanks for the change. The reviewer agreed.

4-3. We realized that our descriptions have caused some confusion for the reviewer to misunderstand our pep-tRNA profiling....

- All right, the reviewer misunderstood. Now that these are properly understood, the reviewer has written the three requirements as described above (answer 2). If these are met, the MS signals will be distinguished as peptide origin rather than chemical noise origin. Note that the increase in MS signals at 30 minutes is a necessary condition but not a sufficient condition. This is because the chemical noise may have only increased due to a heat shock.

For Table S1, the reviewer was satisfied because you added peptide score, pepExpect and peptide mass. As long as there is MS/MS as a piece of evidence, a mass error of 5 ppm is fine.

4-4. Thank you for commenting on this. Based on the results of the experiments

-The reviewer does not agree with the authors' comment that peptide analysis of pep-tRNA is more sensitive than pep-tRNA profiling. The authors' logic that a method that identifies 700 peptides from 40 mg of total RNA is more sensitive than a method that detects >5500 peptides from 4 µg of total RNA is contradictory. So, what are the 5000 peptides, excluding about 500 (68% of 700) identified peptides from the 5500 peptides? The reviewer thinks that these "5000 detected peptides" are consistent with the misidentification of chemical noise. Therefore, in order for the pep-tRNA profiling data to be accepted, the three requirements in the answer 2 must be met and resolved the "5000 peptides" paradox. Alternatively, the reviewer strongly recommends that the authors withdraw the pep-tRNA profiling data and discussion and reconstruct the paper with other data.

4-5. Thank you for this comment. We rephrased it.

- The reviewer agreed.

4-6. Each peptide type represents a cognate, C0X, C-1X, C-2X and C-3X.....

-The reviewer agreed.

4-7. Since the frequency of AAA codons is relatively high in the beginning.....

- Figure 5h should not have been plotted on a frequency basis. Each axis of the graph should be derived from the absolute quantitative value of each peptide, but the graph was plotted on a frequency basis, presumably the authors were unable to obtain such granular data. Furthermore, AAA is clearly an outlier and without this value the argument made in this paragraph is not valid. The authors should accept that this argument is speculative, rewrite it, and move the paragraph beginning "The number of codons at drop-off sites..." on p. 27 to the Discussion section.

4-8. We appreciate this great suggestion. We calculated the drop-off rates ($=\text{drop-off}/(\text{drop-off}+\text{read-through})$) of both.....

- Thank you for acknowledging the reviewer's comments. I appreciate the creation of a new figure, but it should be noted that this figure treats the pentapeptide dissociated at the fifth codon as read-through. The reviewer considers the read-through in this case to be according to the following

equation

read-through=5 mer+6 mer+7 mer+.... +full length

The author needs to prove either in the literature or by experiment that the full length from 6 mer is a negligible amount to prove that 5 mer is equal to the above read-through.

4-9. We'd like to thank the reviewer for bringing this point to our attention. The intrinsic ribosome destabilization (termed IRD)....

- Thank you for accepting the reviewer's comments; according to Figure V, the drop-off peptide also does not seem to contain much acidic amino acids or methionine. The reviewer thinks that the author should also comment on this in the discussion section.

5. Thank you for the comment. We edited the text to make it easier for readers to understand.....

- Thank you. The reviewer agreed.

We thank reviewer #5 for careful review of our manuscript. Our point-by-point responses to reviewer #5's comments are shown below. Changes to the main text are marked in yellow.

Reviewer #5 (Remarks to the Author)

This revised manuscript is a significant improvement over the previous draft. The authors appear to have satisfactorily addressed some of the previous reviewers' comments. The reviewer is very grateful. However, several issues remain, as described below.

1. The novelty....

We appreciate this reviewer's constructive comments. We tried to improve the introduction that is readable for non-specialists.

-Unfortunately, the second paragraph of p3, starting with "Fidelity of translation is..." which I pointed out, did not seem to be changed.

2. Strange....

2-1. Thank you very much for evaluating our additional data and suggestion for explanation of our results. The brief summary of the results of this experiment has been added to the Method section.

-The reviewer was satisfied with your response. Thank you very much.

Thank you very much.

2-2. To verify the reproducibility of the pep-tRNA profiling in Fig.4.....

-

The reviewer recognized that the authors are claiming that these MS signals are detected as peptides solely because their mass values match those in the database. The reviewer would like to know if the evidence that these are peptides is really just the mass value. Although not stated in the text, the reviewer presumes that one of the requirements for this material to be a peptide is that these MS signals are detected as doublets 2 Da apart because they are cleaved from tRNA in 50% H₂¹⁸O and therefore contain 18^{OH} and 16^{OH} 1:1 at the carboxy terminus. If this assumption is correct, the authors should provide more details in the Methods section. The Methods section should also include the threshold for extracting a significant MS signal from background noise.

In the opinion of the reviewers, the above is not sufficient. At least three additional requirements must be met. The first is to show a control experiment in which the bare peptide is shown to be free of OH substitutions under the same conditions. The second is to measure MS/MS for some very strong signals and identify them as peptides. Third, it would also be necessary to create a decoy database and measure its FDR for detection on MS signals only. Appropriate FDR values would also need to be set. The reviewer suggests that these should be described in the Methods and Results sections to maintain transparency. The reviewer

believes that once all these issues have been addressed, the reproducibility of the pep-tRNA profiling should be discussed.

Regarding the first requirement, the OH group of the carboxyl moiety of bare peptide is chemically stable, and it is impossible that the OH is spontaneously replaced by the surrounding water-derived OH under the conditions we used. Therefore, we do not think this control experiment is necessary.

To address the second and third requirements, we have performed MS/MS-based sequence analyses of abundant peptides with strong signals in the pep-tRNA profiling. To calculate the FDR, we used the decoy database, which consists of peptides with the reverse sequence of the reference peptides of the N-terminal sequences of the *E. coli* ORFs. We here show several examples of peptides identified in this experiment by setting 1.72% FDR threshold (Figure I). These peptides were also identified in the isolated pep-tRNAs (Table S1), clearly demonstrating that pep-tRNA profiling can detect nascent peptides of pep-tRNAs.

We added the sentence in the main text to clearly mention that the peptides found in pep-tRNA profiling are detected based on their accurate masses without CID data. In addition, we included the data of MS/MS-based peptide sequence (shown in Figure I) in Figure S1f.

2-3. Regarding potential bias in peptide types caused by the Q Sepharose separation.....

-

The authors are inconsistent in stating that the recovery of peptides on Sepharose is better than that on C18, while enriching with C18 tips. The authors could not deny the bias, so they should accept the bias of the detected peptides. After accepting it, for example, the authors should present the hydrophobicity and pI ranges of the detectable peptides as a limitation of this method. This is because these physicochemical enrichment and selection methods depend on the hydrophobicity and charge of the peptides. The reviewer recommends that the authors discuss in the text which peptides would be detectable by comparing the hydrophobicity and pI of the peptides in the N-terminal sequence database with those actually detected.

Regarding this comment, we do not use Q-sepharose to recover peptides but to trap tRNAs for their removal

after deacylation of pep-tRNAs, i.e., separation of peptides and tRNAs, because injection of a large amount of tRNAs would affect peptide separation by LC/MS. So, we do not state in the text that the recovery of peptides on Q-Sepharose is better than that on C18. In general, unspecific binding to Sepharose resin must be smaller than that to C18. Here, we removed tRNAs by Q-sepharose with a buffer containing 250 mM NaCl. Most of anionic peptides should not be trapped but elute in flowthrough fraction in this condition. In the desalting step after the tRNA removal, most of peptides are enriched by C18 tip. However, it is common knowledge that the highly hydrophobic peptides cannot be recovered efficiently in this process. We added comments on this in METHODS.

3. As suggested, we toned down the sentences and wording in the DISCUSSION regarding the claim that pep-tRNA drop-off occurs frequently under normal growth conditions. Please understand that the drop-off frequency is not affected by Pth mutation.

-

The reviewer thanks the authors for softening the wording in the DISCUSSION. As the authors said, this paper is a kind of methodological paper. Therefore, it is important to inform the reader about the limitation. In this case, the author has clearly presented the detection limits of this method in their previous responses. The reviewer suggests that the authors should include in the text as a limitation of this method that drop-off peptides cannot be detected in wild-type *E. coli*.

The difficulty in detecting pep-tRNAs in WT cells (PTH positive) is due to very rapid degradation of pep-tRNAs in the cells, not to the limitation of our method described here. Therefore, we do not discuss the limitation of our method in relation to the inability to detect pep-tRNAs in WT cells. To help the reader understand this point, we have added the reason why we constructed the temperature-sensitive strain for PTH.

4. The reviewer understands that the peptides do not have beta.....

Thank you very much for understanding our results.

4-1. Thank you for the comment. We also realized importance of measuring cellular level of f-Met-Ado. As shown in Figure II....

- The reviewer is grateful for the new data processing. In this case, the literature showing that dihydrouridine is constant must be mentioned. The authors should provide references either in the figure legend or where appropriate.

There is no report about alteration of dihydrouridine (D) in such short period of time under such specific condition. In general, tRNA modifications are structural components and do not change dynamically in such short period of time. Here we normalized Ac-Ala-Ado, Ac-Met-Ala-Ado (Ac-MA-Ado), f-Met-Ado and dihydrouridine (D) by another tRNA modification (pseudouridine:Ψ, upper panel) and adenosine (A, lower panel) (Figure II). This additional analysis clearly shows that the amounts of f-Met-tRNA and D were almost constant in such short period of time.

4-2. We have changed "all nascent peptides" to "most nascent peptides".....

- Thanks for the change. The reviewer agreed.

Thank you very much.

4-3. We realized that our descriptions have caused some confusion for the reviewer to misunderstand our pep-tRNA profiling....

- All right, the reviewer misunderstood. Now that these are properly understood, the reviewer has written the three requirements as described above (answer 2). If these are met, the MS signals will be distinguished as peptide origin rather than chemical noise origin. Note that the increase in MS signals at 30 minutes is a necessary condition but not a sufficient condition. This is because the chemical noise may have only increased due to a heat shock.

To avoid such misunderstanding, we have added a detailed explanation of the purpose of performing the isolated pep-tRNA analyses to the main text to clarify the difference between pep-tRNA profiling and isolated pep-tRNA analyses. The validity of pep-tRNA profiling has been answered in other responses.

4-4. Thank you for commenting on this. Based on the results of the experiments

-

The reviewer does not agree with the authors' comment that peptide analysis of pep-tRNA is more sensitive than pep-tRNA profiling. The authors' logic that a method that identifies 700 peptides from 40 mg of total RNA is more sensitive than a method that detects >5500 peptides from 4 µg of total RNA is contradictory. So, what are the 5000 peptides, excluding about 500 (68% of 700) identified peptides from the 5500 peptides? The reviewer thinks that these "5000 detected peptides" are consistent with the misidentification of chemical noise. Therefore, in order for the pep-tRNA profiling data to be accepted, the three requirements in the answer 2 must be met and resolved the "5000 peptides" paradox. Alternatively, the reviewer strongly recommends that the authors withdraw the pep-tRNA profiling data and discussion and reconstruct the paper with other data.

We first prepared pep-tRNAs from the cell. During this preparation, most of chemicals and compounds should be removed from the tRNA fraction. As shown in Figure I, we confirmed several abundant peptides by CID. As shown in Figure S4, we also confirmed ¹⁸O-labeled peptides when pep-tRNAs are deacylated in ¹⁸O-water. This is concrete evidence showing 5000 peptides are actually derived from pep-tRNAs.

As pointed out, it is true that peptide analysis of individual pep-tRNA is more sensitive than pep-tRNA profiling, because 32% of 700 peptides are not detected in pep-tRNA profiling. But, please notice that we only showed a limited numbers of peptides with high quality and confident CID data in the analyses on isolated pep-tRNAs. The 5000 peptides in the pep-tRNA profiling were assigned based on their accurate mass without CID data, because we know that they precisely match the N-terminal peptides of *E. coli* ORFs. But they do not always match unique ORFs. In fact, a single peptide often hits multiple ORFs, especially for peptides shorter than 5-mer, it is difficult to attribute them to a single gene. Therefore, to discuss detailed mechanism of pep-tRNA drop-off, we isolated individual pep-tRNAs and carefully sequenced their nascent peptides. In fact, we only picked up peptides that have high quality CID and are surely assigned to N-terminal sequence of specific ORF. Please notice that we discarded a number of peptides with low quality CID. Peptides that hit more than one ORF are also excluded. Thus, we collected not only abundant peptides, but also minor peptides derived from individual pep-tRNAs. In addition, about 20% of pep-tRNAs were deacylated during isolation of individual pep-tRNA by RCC (Figure S6a). They are the reasons why we showed 700 peptides and only 68% of them are detected in pep-tRNA profiling.

4-5. Thank you for this comment. We rephrased it.

- The reviewer agreed.

Thank you very much.

4-6. Each peptide type represents a cognate, COX, C-1X, C-2X and C-3X.....

-The reviewer agreed.

Thank you very much.

4-7. Since the frequency of AAA codons is relatively high in the beginning.....

- Figure 5h should not have been plotted on a frequency basis. Each axis of the graph should be derived from the absolute quantitative value of each peptide, but the graph was plotted on a frequency basis, presumably the authors were unable to obtain such granular data. Furthermore, AAA is clearly an outlier and without this value the argument made in this paragraph is not valid. The authors should accept that this argument is speculative, rewrite it, and move the paragraph beginning "The number of codons at drop-off sites..." on p. 27 to the Discussion section.

The discussion here is based on the number of pep-tRNAs dissociated from each codon, not on their abundance (peak intensity of MS signal for every peptide). Therefore, frequency should be appropriate for plotting. While all codons should be included in our analyses to avoid any bias, we described in the figure legend the coefficient of determination in the case of excluding AAA codon.

4-8. We appreciate this great suggestion. We calculated the drop-off rates (=drop-off/(drop-off+read-through)) of both.....

- Thank you for acknowledging the reviewer's comments. I appreciate the creation of a new figure, but it should be noted that this figure treats the pentapeptide dissociated at the fifth codon as read-through. The reviewer considers the read-through in this case to be according to the following equation read-through=5 mer+6 mer+7 mer+.... +full length. The author needs to prove either in the literature or by experiment that the full length from 6 mer is a negligible amount to prove that 5 mer is equal to the above read-through.

Since the drop-off efficiency strongly depends on the codon-anticodon pairing at a given codon position, the drop-off and readthrough efficiencies at the next codon and subsequent positions should be the same regardless of the miscoding event at the given position. In addition, we observed the same amounts of pep-tRNAs bearing 3-mer, 4-mer, and 5-mer cognate peptides in both WT and *ΔtmcA* strains (main **Figure 6b**). Thus, to calculate the drop-off rates for the 4-mer cognate (MWGI-tRNA) and noncognate pep-tRNAs (MWGM-tRNA), we use the 5-mer pep-tRNA (MWGIV-tRNA and MWGMV-tRNA) as products of readthrough at the fourth AUA codon. The formula, 4-mer pep-tRNA/(4-mer pep-tRNA + 5-mer pep-tRNA), was applied to calculate the drop-off rate in this study. We added this explanation in METHODS.

4-9. We'd like to thank the reviewer for bringing this point to our attention. The intrinsic ribosome destabilization (termed IRD)....

- Thank you for accepting the reviewer's comments; according to Figure V, the drop-off peptide also

does not seem to contain much acidic amino acids or methionine. The reviewer thinks that the author should also comment on this in the discussion section.

Thank you for suggestion. We mentioned this in the legend of Table S3.

5. Thank you for the comment. We edited the text to make it easier for readers to understand.....

- Thank you. The reviewer agreed.

Thank you very much.